# New fossil remains of *Homo naledi* from the Lesedi Chamber, South Africa

John Hawks[1,2], Marina Elliott[1], Peter Schmid[1,3], Steven E Churchill[1,4], Darryl J de Ruiter[1,5], Eric M Roberts[6], Hannah Hilbert-Wolf[6], Heather M Garvin[1,7,8], Scott A Williams[1,9,10], Lucas K Delezene[1,11], Elen M Feuerriegel[1,12], Patrick Randolph-Quinney[1,13,14], Tracy L Kivell[1,15,16], Myra F Laird[1,17], Gaokgatlhe Tawane[1], Jeremy M DeSilva[1,18], Shara E Bailey[9,10], Juliet K Brophy[1,19], Marc R Meyer[20], Matthew M Skinner[1,15,16], Matthew W Tocheri[21,22], Caroline VanSickle[1,2,23], Christopher S Walker[1,4,24], Timothy L Campbell[5], Brian Kuhn[25], Ashley Kruger[1,26], Steven Tucker[1], Alia Gurtov[1,2], Nompumelelo Hlophe[1], Rick Hunter[1], Hannah Morris[1,27], Becca Peixotto[1,28], Maropeng Ramalepa[1], Dirk van Rooyen[1], Mathabela Tsikoane[1], Pedro Boshoff[1], Paul HGM Dirks[6], Lee R Berger[1]*

[1]Evolutionary Studies Institute, University of the Witwatersrand, Wits, South Africa; [2]Department of Anthropology, University of Wisconsin, Madison, United States; [3]Anthropological Institute and Museum, University of Zürich, Winterthurerstr, Zürich, Switzerland; [4]Department of Evolutionary Anthropology, Duke University, Durham, United States; [5]Department of Anthropology, Texas A&M University, College Station, United States; [6]Geosciences, College of Science and Engineering, James Cook University, Townsville, Australia; [7]Department of Anthropology/Archaeology, Mercyhurst University, Erie, United States; [8]Department of Applied Forensic Sciences, Mercyhurst University, Erie, United States; [9]Center for the Study of Human Origins, Department of Anthropology, New York University, New York, United States; [10]New York Consortium in Evolutionary Primatology, New York, United States; [11]Department of Anthropology, University of Arkansas, Fayetteville, United States; [12]Department of Anthropology, University of Washington, Seattle, United States; [13]School of Anatomical Sciences, University of the Witwatersrand Medical School, Johannesburg, South Africa; [14]School of Forensic and Applied Sciences, University of Central Lancashire, Preston, United Kingdom; [15]School of Anthropology and Conservation, University of Kent, Canterbury, United Kingdom; [16]Department of Human Evolution, Max Planck Institute for Evolutionary Anthropology, Leipzig, Germany; [17]Department of Organismal Biology and Anatomy, University of Chicago, Chicago, United States; [18]Department of Anthropology, Dartmouth College, Hanover, United States; [19]Department of Geography and Anthropology, Louisiana State University, Baton Rouge, United States; [20]Department of Anthropology, Chaffey College, Rancho Cucamonga, United States; [21]Department of Anthropology, Lakehead University, Thunder Bay, Canada; [22]Human Origins Program, Department of Anthropology, National Museum of Natural History, Smithsonian Institution, Washington, United States; [23]Department of Anthropology, Bryn Mawr College, Bryn Mawr, United States; [24]Department of Molecular Biomedical Sciences, College of Veterinary Medicine, North Carolina State University, Raleigh, United States; [25]Department of Geology, University of Johannesburg, Johannesburg, South Africa; [26]School of Geosciences, University of the Witwatersrand, Johannesburg, South Africa; [27]Department of

*For correspondence: Lee.Berger@wits.ac.za

Competing interests: The authors declare that no competing interests exist.

Forestry and Natural Resources, University of Georgia, Athens, United States; [28]Department of Anthropology, American University, Washington, United States

**Abstract** The Rising Star cave system has produced abundant fossil hominin remains within the Dinaledi Chamber, representing a minimum of 15 individuals attributed to *Homo naledi*. Further exploration led to the discovery of hominin material, now comprising 131 hominin specimens, within a second chamber, the Lesedi Chamber. The Lesedi Chamber is far separated from the Dinaledi Chamber within the Rising Star cave system, and represents a second depositional context for hominin remains. In each of three collection areas within the Lesedi Chamber, diagnostic skeletal material allows a clear attribution to *H. naledi*. Both adult and immature material is present. The hominin remains represent at least three individuals based upon duplication of elements, but more individuals are likely present based upon the spatial context. The most significant specimen is the near-complete cranium of a large individual, designated LES1, with an endocranial volume of approximately 610 ml and associated postcranial remains. The Lesedi Chamber skeletal sample extends our knowledge of the morphology and variation of *H. naledi*, and evidence of *H. naledi* from both recovery localities shows a consistent pattern of differentiation from other hominin species.

## Introduction

The Rising Star cave system (26°1′13′′ S; 27°42′43′′ E, *Figure 1*) in the Cradle of Humankind World Heritage Site, Gauteng Province, South Africa, is known for the discovery in 2013 of more than 1,550 fossils representing a novel hominin species, *Homo naledi* (*Berger et al., 2015*; *Dirks et al., 2015*). These remains, representing at least 15 individuals of various ages at death, were recovered from a deep chamber (30 m below ground surface), named the Dinaledi Chamber.

Additional fossil hominin material was subsequently discovered in the Lesedi Chamber of the cave system in November 2013 by Rick Hunter and Steven Tucker. The deposition of sediment and skeletal remains in the Lesedi Chamber has no direct geological connection to the Dinaledi Chamber. In the time following the first discovery of hominin material in the Lesedi Chamber, excavators have recovered 131 hominin specimens within three discrete collection areas. The sedimentary context of the three collection areas is broadly similar, but we have not yet established whether the fossil material resulted from a single depositional episode or from multiple distinct events.

We approached the hominin skeletal remains from the Lesedi Chamber with the aim of identifying elements, assessing the number of individuals represented by the material, and determining the taxonomic identity of the sample. Preliminary examination of the hominin remains suggested that they are morphologically consistent with *H. naledi*. To test this hypothesis, we carried out systematic comparisons, employing the taxonomic diagnosis of this species (*Berger et al., 2015*) and focusing upon those characters that distinguish *H. naledi* from other hominin taxa. We also present essential contextual information to place the specimens within the Lesedi Chamber and provide descriptions of the hominin specimens, focusing upon those features that contribute to the taxonomic diagnosis of the sample. All identifiable hominin fragments, including those that do not present information useful to taxonomic diagnosis, are listed in *Table 1*.

## Results

### Name of the chamber

Following the University of the Witwatersrand's fossil-numbering system (*Zipfel and Berger, 2009*), this second *H. naledi* locality has been designated U.W. 102. The chamber itself has been named the Lesedi Chamber, a word meaning 'light' in Setswana. By contrast, the Dinaledi Chamber was numbered site U.W. 101. Excavations in the Lesedi Chamber have been carried out in three areas, designated U.W. 102a, U.W. 102b, and U.W. 102c.

**eLife digest** Species of ancient humans and the extinct relatives of our ancestors are typically described from a limited number of fossils. However, this was not the case with *Homo naledi*. More than 1500 fossils representing at least 15 individuals of this species were unearthed from the Rising Star cave system in South Africa between 2013 and 2014. Found deep underground in the Dinaledi Chamber, the *H. naledi* fossils are the largest collection of a single species of an ancient human-relative discovered in Africa.

After the discovery was reported, a number of questions still remained. These questions included: why were so many fossils from a single species found at the one site, and how did they come to rest so far into the cave system? Possible explanations such as *H. naledi* living in the cave or being washed in by a flood were considered but ruled out. Instead, the evidence was largely consistent with intact bodies being deliberately disposed of in the cave and then decomposing.

Now, Hawks et al. – who include many of the researchers who were involved in the discovery of *H. naledi* – report that yet more *H. naledi* fossils have been unearthed from a second chamber in the Rising Star cave system, the Lesedi Chamber. The chamber is 30 meters below the surface and there is no direct route between it and the Dinaledi Chamber. Again, the evidence is most consistent with the bodies arriving intact into the chamber, and there were no signs that the remains had been exposed to the surface environment.

Also like the Dinaledi Chamber, no remains of other ancient humans or their relatives were found in the Lesedi Chamber. In total, 133 fossils of *H. naledi* have been found in this second chamber representing at least three individuals: two adults and a juvenile. However, and as Hawks et al. point out, only a small volume of the chamber has been excavated so far, and so there are likely more fossils still to be found.

The fossils in the Lesedi Chamber are similar to those found before but include intact examples of bones, like the collarbone, that were previously known only from fragments. Perhaps the most impressive among the new fossils is a relatively complete skull that is part of a partial skeleton. The skull could have housed a brain that was 9% larger than the maximum estimate calculated from the previous *H. naledi* fossils.

Though these new fossils provide us with yet more information about *H. naledi*, some questions still remain unanswered – the material from the Lesedi Chamber is undated, for example. However, a related study by Dirks et al. does give an estimate for the age of the fossils from the Dinaledi Chamber, while Berger et al. provide an explanation for why this date might be much younger than was previously predicted.

## Location of the Lesedi Chamber

The Lesedi Chamber is in the central sector of the Rising Star system (*Figure 2*), at a depth of ~30 m from the surface directly above the chamber. All measurements reported here are approximate. The first fossil deposit to be recognized (U.W. 102a) is located just off the southwest corner of the North-South Fracture Passage, a northern arm of the Lesedi Chamber. This fossil deposit is approximately 60 m NNE in a straight line from the Dinaledi Chamber. There is no straight-line route between the Dinaledi and Lesedi Chambers, and the shortest traversable route between the two areas is almost 145 m. There are currently four access routes from the surface to the Lesedi Chamber. The most accessible of these currently follows an 86 m downward-sloping path with several narrow passages and short climbs, but only one squeeze and no significant crawls. This has been the main access route for excavators. The other three routes are each substantially more challenging.

## Location of skeletal material within the Lesedi Chamber

In addition to the first fossil deposit to be recognized in the chamber, two additional concentrations of skeletal material have been identified to date (*Figure 3*), and we have designated these as areas 102a, 102b, and 102c. We began investigating each of these areas because team members noticed hominin fossil material exposed on sediment surfaces. The discovery of 102a by Rick Hunter and Steven Tucker led to the initial scientific investigation of the chamber; discoveries of both 102b and

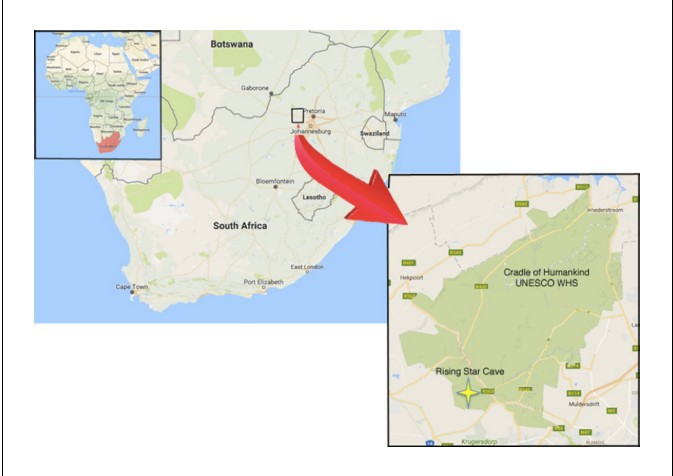

**Figure 1.** Geographical location of the Rising Star cave in the Cradle of Humankind UNESCO World Heritage Site.

102c were made by Hannah Hilbert-Wolf during the course of geological sampling of the chamber. These three areas do not represent a systematic sampling of the chamber's contents and we have excavated only a very small sediment volume, less than 200 L (<0.2 m$^3$) in total from all three areas. The chamber contains a much greater volume of sediment and we do not know what density of fossil bone it may contain beyond our samples.

U.W. 102a is located at the entrance of a 20–50-cm-wide blind tunnel, which is 1.8 m long in total. The blind tunnel leads off of the southwest corner of the North-South Fracture Passage (*Figure 3*). Fossil material was exposed on the surface within this blind tunnel at the time of discovery. We have excavated the proximal 1.5 m of this blind tunnel, which has a tapering width of less than 50 cm in our excavation unit. The depth of excavation in this area is a maximum of 40 cm. The deposit in this area is a weakly stratified, unlithified mud-clast breccia. Most hominin material has been recovered from an approximately 10-cm-thick horizon of fine-grained mud-clast breccia, beneath a surface layer of ~2 cm of lighter brown-colored mudstone. This deposit is the source of at least some of the sediments that slope from the blind tunnel into the Antechamber. Fossil material attributed to 102a has also been recovered from the surface within the North-South Fracture Passage.

U.W. 102b is a sediment deposit on a horizontal chert shelf 80 cm above the cave floor along the western wall of the Antechamber. It is also dominated by unlithified mud-clast breccia. The 102b deposit is located ~3.8 m to the south and 1.8 m below the 102a deposit. After the discovery of hominin fossil material on the surface here, we undertook limited excavations, with a total volume of ~20 L.

U.W. 102c is a small unlithified sediment deposit within an irregular dissolution cavity on the north wall of the east–west-running Cake-Icing Fracture. This deposit is 1.3 m above the current cave floor. It is 11.6 m from U.W. 102a, and 0.3 m below the level of the 102a fossils. We have excavated this small sediment pocket in its entirely, with a total volume of approximately 2 L.

Geological work to characterize the Lesedi Chamber depositional history is underway. The stratigraphy is complex, with some hominin and faunal material concentrated in deposits of poorly consolidated mud-clast breccia, generally similar to the facies in the Dinaledi Chamber (*Dirks et al., 2015*). Notably, the fossil material in the Lesedi Chamber is concentrated in minor side fractures, dissolution cavities, or on chert shelves well above the current chamber floor. Our working hypothesis is that the chamber once held a greater volume of sediment than is present today, and when sediment eroded from the chamber, erosional remnants remained in protected fractures, wall cavities, and on chert shelves along the chamber walls. This and other indications of reworking of the deposits make it uncertain how much of the hominin assemblage may remain in its primary depositional context.

**Table 1.** Hominin fossil material from the Lesedi Chamber. All diagnostic hominin specimens are listed, with attribution to element. Specimens that have been refitted are not listed separately. Most Locality 102a cranial fragments are presumed to be part of LES1 and are not listed separately.

| Specimen number | Element | Notes |
|---|---|---|
| **LOCALITY 102a** | | |
| LES1 | cranium | constituted of 57 specimens, not listed separately |
| U.W. 102a-001 | proximal right femur | |
| U.W. 102a-002 | proximal right humerus | |
| U.W. 102a-003 | proximal left femur | |
| U.W. 102a-004 | distal left femur | |
| U.W. 102a-010 | right scapula fragment | acromion |
| U.W. 102a-013 | humeral head fragments | |
| U.W. 102a-015 | right proximal ulna | |
| U.W. 102a-018 | long bone fragment | immature |
| U.W. 102a-019 | partial rib | |
| U.W. 102a-020 | right ulna fragment | |
| U.W. 102a-021 | right clavicle | |
| U.W. 102a-025 | right radius shaft fragment | |
| U.W. 102a-028 | right fourth metacarpal | |
| U.W. 102a-036 | T10 vertebra | |
| U.W. 102a-039 | rib fragments | |
| U.W. 102a-040 | long bone shaft fragment | |
| U.W. 102a-117 | right scaphoid | |
| U.W. 102a-138 | right ilium fragments | immature |
| U.W. 102a-139 | L5 vertebra fragments | |
| U.W. 102a-148 | sternum fragment | |
| U.W. 102a-151 | T11 vertebra | |
| U.W. 102a-152 | rib fragments | |
| U.W. 102a-154 | T12 and L1 vertebrae | found in articulation |
| U.W. 102a-155 | mid-thoracic vertebral body | |
| U.W. 102a-171 | atlas fragment | |
| U.W. 102a-172 | atlas fragment | |
| U.W. 102a-189 | rib fragment | |
| U.W. 102a-195 | rib fragment | |
| U.W. 102a-206 | left clavicle fragment | |
| U.W. 102a-207 | rib fragment | |
| U.W. 102a-210 | sacral element | immature, possibly S1 |
| U.W. 102a-231 | rib fragment | |
| U.W. 102a-232 | rib fragment | |
| U.W. 102a-236 | humerus head fragment | |
| U.W. 102a-239 | left clavicle fragment | |
| U.W. 102a-247 | right scapula fragment | coracoid process |
| U.W. 102a-250 | right first rib | |
| U.W. 102a-252 | rib fragment | |
| U.W. 102a-256 | left scapula fragment | portion of body, spine, and acromion |
| U.W. 102a-257 | left proximal humerus | |

*Table 1 continued on next page*

*Table 1 continued*

| Specimen number | Element | Notes |
|---|---|---|
| U.W. 102a-279 | left scapula fragment | partial glenoid fossa |
| U.W. 102a-280 | rib fragment | |
| U.W. 102a-300 | vertebral fragment | |
| U.W. 102a-306 | L4 vertebra body | |
| U.W. 102a-322 | L2 vertebra body | |
| U.W. 102a-337 | vertebral fragment | neural arch |
| U.W. 102a-348 | right pubic ramus fragment | |
| U.W. 102a-349 | vertebral fragment | neural arch |
| U.W. 102a-358 | rib fragments | |
| U.W. 102a-360 | vertebral fragment | |
| U.W. 102a-455 | ulna shaft fragment | |
| U.W. 102a-456 | ulna shaft fragment | |
| U.W. 102a-470 | rib fragments | |
| U.W. 102a-471 | right distal radius fragment | |
| U.W. 102a-474 | long bone fragment | immature |
| U.W. 102a-476 | right capitate | |
| U.W. 102a-477 | partial right lunate | |
| U.W. 102a-479 | rib fragment | |
| **LOCALITY 102b** | | |
| U.W. 102b-178 | $LI_2$ | |
| U.W. 102b-437 | $rdm_2$ | |
| U.W. 102b-438 | right mandibular corpus fragment | immature, $RP_4$ in crypt |
| U.W. 102b-502 | cranial fragments | |
| U.W. 102b-503 | $RP_4$ crown | |
| U.W. 102b-506 | cranial fragment | |
| U.W. 102b-507 | cranial fragment | |
| U.W. 102b-509 | cranial fragment | |
| U.W. 102b-511 | $LC_1$ crown | |
| U.W. 102b-514 | cranial fragment | |
| U.W. 102b-515 | $LI^2$ | |
| U.W. 102b-516 | cranial fragment | |
| **LOCALITY 102c** | | |
| U.W. 102 c-589 | left mandibular fragment | $LM_1$ and $LM_2$ in place |

## Hominin material from 102a

Hominin material from the 102a area includes 118 identifiable specimens (*Table 1*; *Figure 4*). Fifty-seven of these are cranial and dental specimens that either refit directly or are morphologically compatible with a nearly complete fossil cranium, designated LES1 (*Figure 5*). Hominin postcranial remains from locality 102a include 61 identified specimens that represent a minimum of 31 postcranial elements, not counting ribs. These include a minimum of two partial femora, two partial humeri, one complete clavicle and two clavicular fragments, two partial ulnae, several fragments of scapula and radius, many rib fragments, a near-complete first rib, a partial sternum, four hand and wrist elements, an immature ilium and sacrum fragment, and a partial thoracic and lumbar vertebral column. Every anatomical region of the skeleton is represented with the notable exceptions of tibia, fibula and pedal remains.

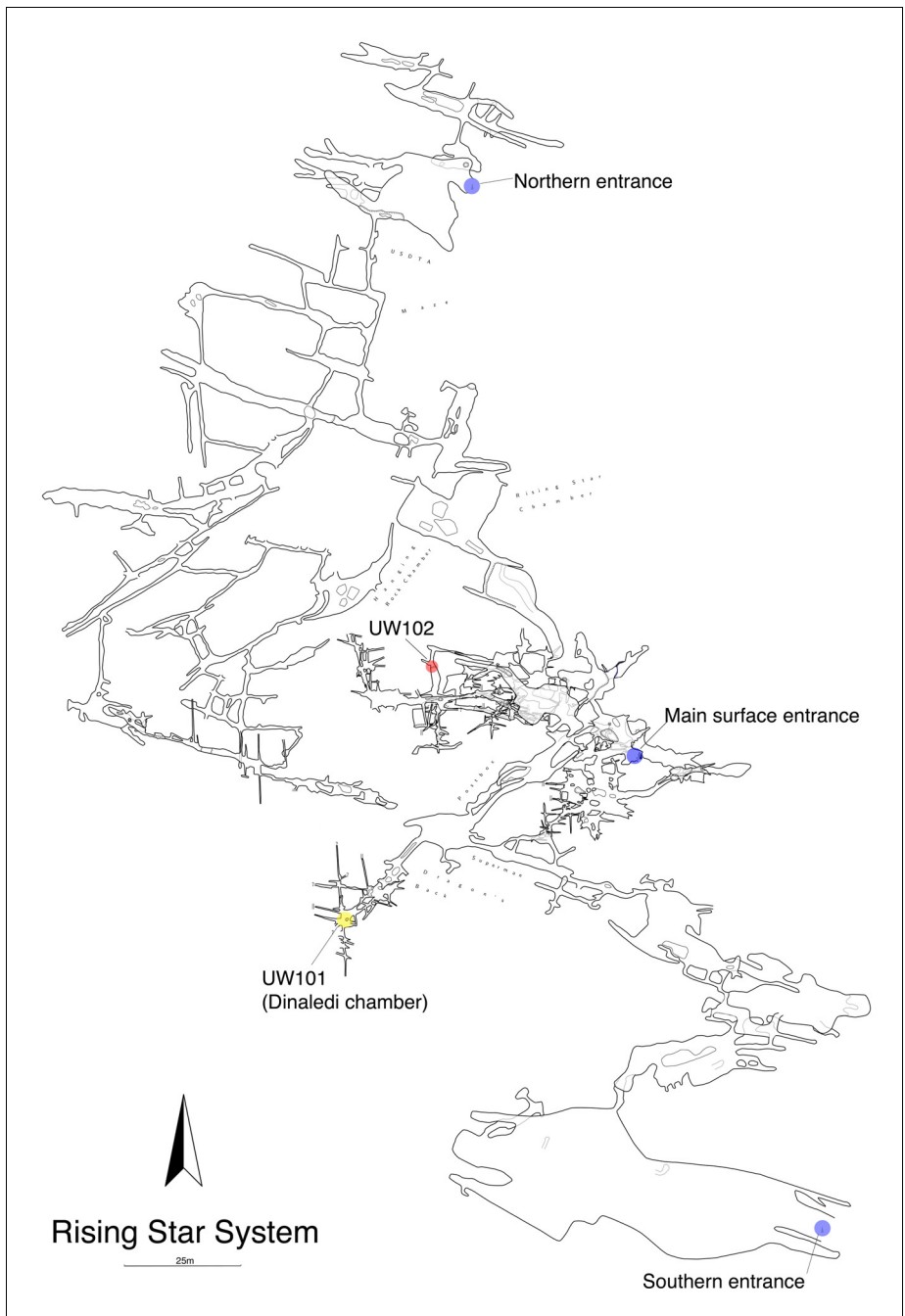

**Figure 2.** Location of the Lesedi Chamber (U.W.102) in the Rising Star system (red circle). The Dinaledi Chamber (U.W. 101) is marked by a yellow circle, while three surface entrances into the system are marked by blue circles.

## LES1

The LES1 cranium is fragmented but is represented by most of the vault and part of the face (*Figure 5*). To date, we have successfully refitted the near-complete mandible, the near-complete right maxilla, a partial palate and a partial left maxillary dental row, and a partial vault including the near-complete frontal, left and right nasal and left lacrimal bones, near-complete left parietal and temporal, partial right parietal, and a portion of left occipital. LES1 has a complete adult dentition except for the crowns of the lower left central and lateral incisors. The face is reconstructed from the partial right maxillary bone, including the frontal process, which refits to the right nasal bone and frontal.

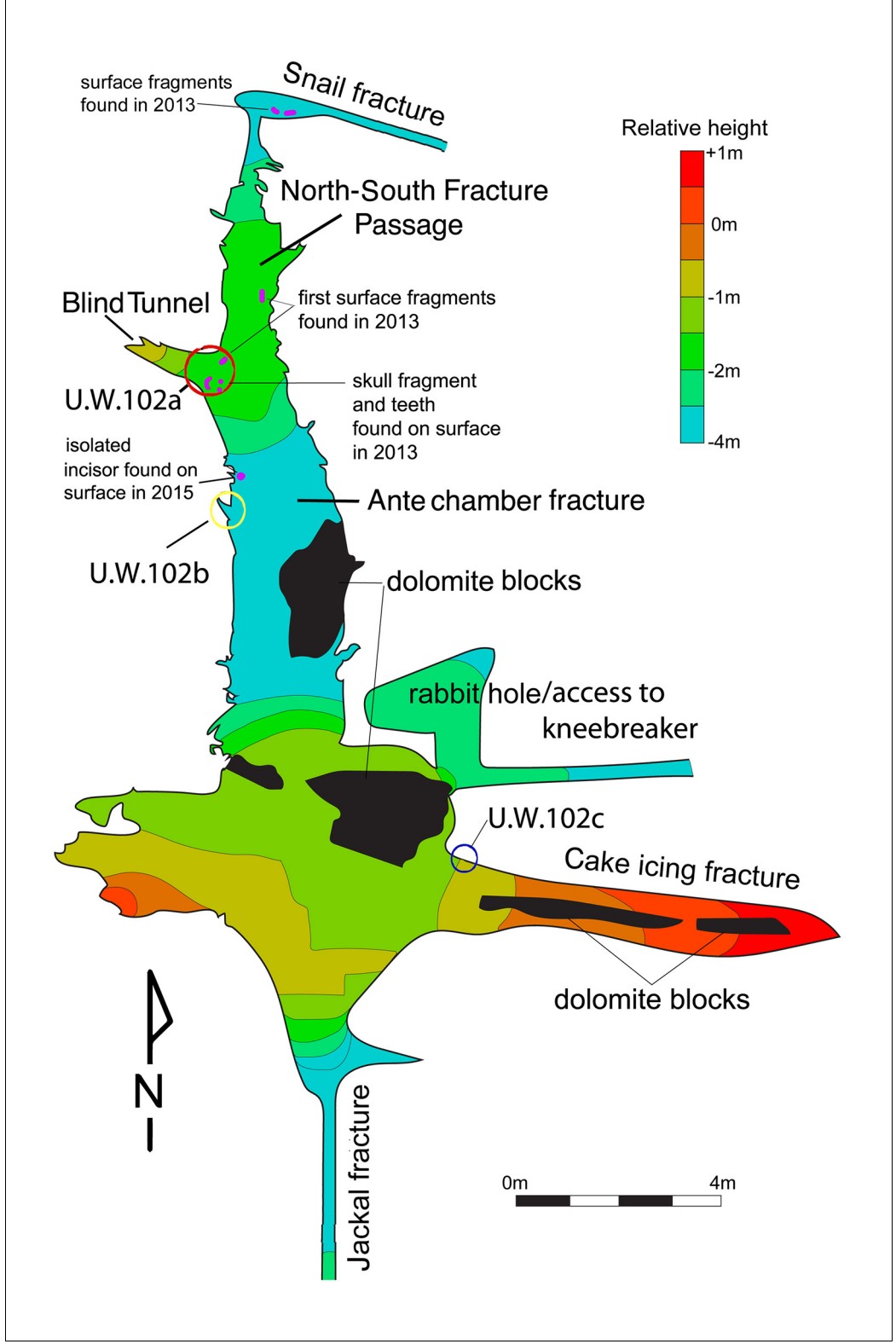

**Figure 3.** Schematic of the Lesedi Chamber, showing the three hominin-bearing collection areas: U.W.102a, 102b, and 102c.

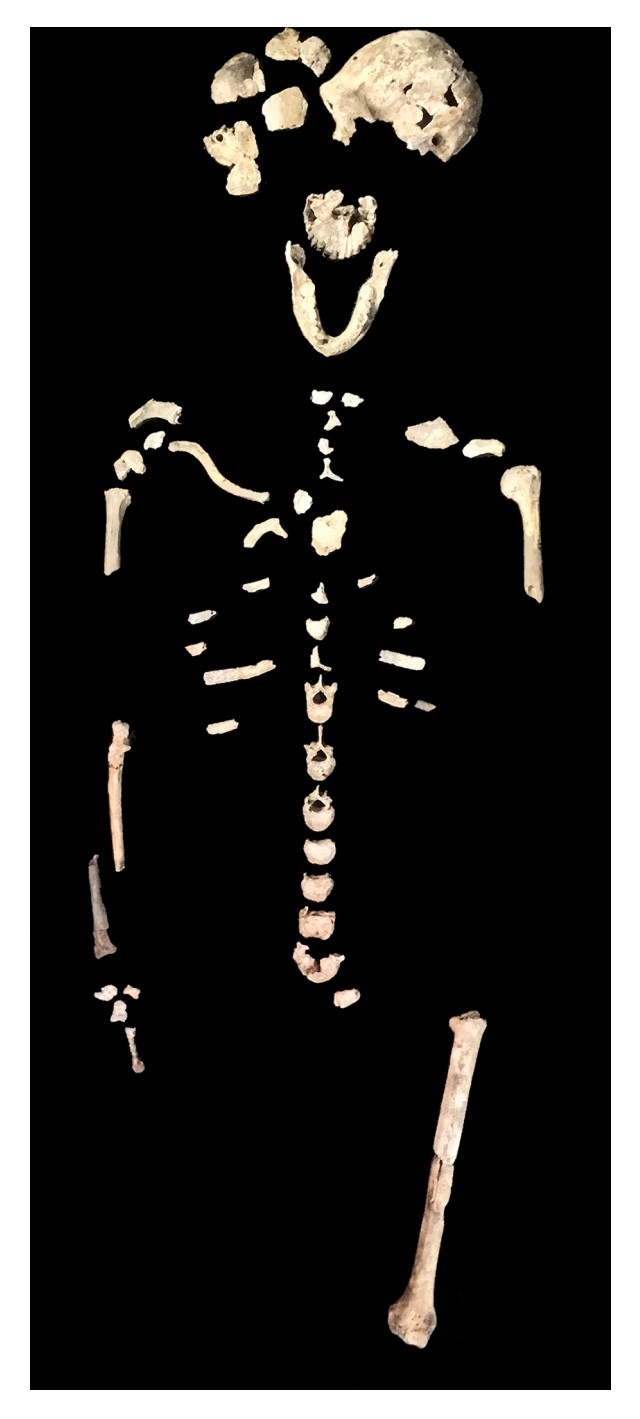

**Figure 4.** Skeletal material from locality 102a provisionally assigned to the LES1 skeleton. The adult cranial material from 102a all belongs to a single cranium; most of the adult postcranial material probably belongs to the same individual. The adult cranial and postcranial material is shown here, except for the U.W. 102a-001 femur. The possibility that the femora represent two adult individuals makes it unclear which femur may be attributable to the skeleton; for the purposes of illustration, the U.W. 102a-003/U.W. 102a-004 femur is included in this photograph.

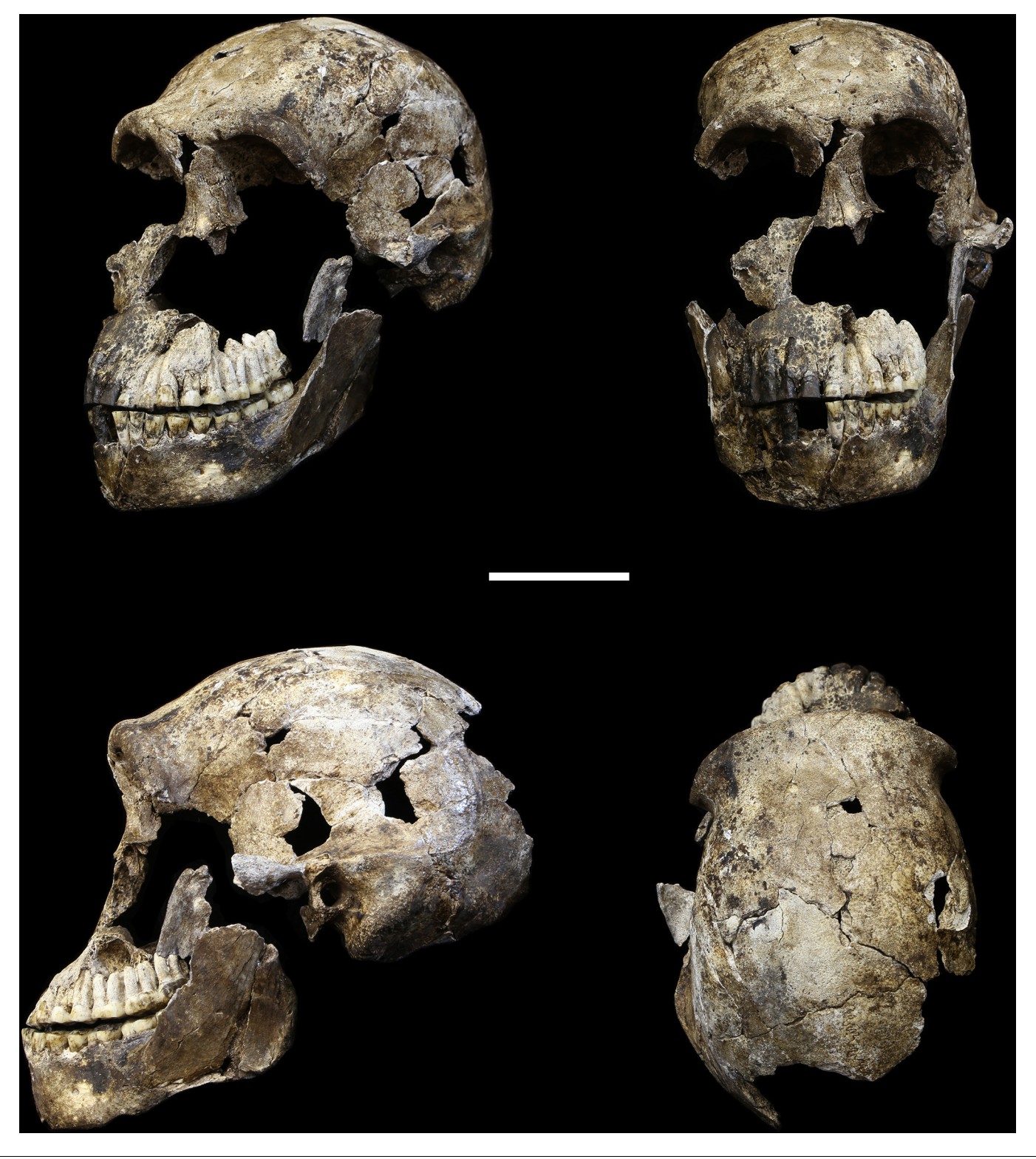

**Figure 5.** LES1 cranium. Clockwise from upper left: three-quarter, frontal, superior and left lateral views. Fragments of the right temporal, the parietal and the occipital have also been recovered (not pictured), but without conjoins to the reconstructed vault or face. Scale bar = 5 cm.

The left mandibular ramus is well-enough preserved to allow a rough estimation of the condyle position, enabling an approximation of the midsagittal contour of the face (*Figure 5*).

All additional cranial fragments in the present 102a collection are non-duplicative with this refitted vault and face, and where they represent the opposite side of the vault, they match in morphological detail. However, many of the fragments lack clear refits with the existing vault or maxillary portions. Further physical reconstruction of the cranium will await fragments that may emerge from excavation in the future. The refitted vault, with the application of virtual mirror reconstruction, is sufficient to allow an estimate of endocranial volume of approximately 610 ml (*Figure 6*).

Most of the features of the LES1 vault are characteristic of *H. naledi* from the Dinaledi Chamber (*Supplementary file 1*; *Figure 7*). The LES1 vault is relatively short anteroposteriorly, without the elongation and sharp occipital angulation found in *H. erectus*. LES1 exhibits mild frontal and parietal bossing, similar to *H. naledi* DH3. Other features on the vault that are consistent with *H. naledi* include prelambdoidal flattening, limited postorbital constriction, widely spaced temporal lines, a continuous supraorbital torus with a supratoral sulcus, an occipital torus, and a marked angular torus. In the temporal region, LES1 has an anteroinferiorly oriented root of the zygomatic process of the temporal, a medially positioned mandibular fossa, a small and obliquely oriented external auditory meatus, a projecting Eustachian process, a small vaginal process, a weak crista petrosa, a triangular-shaped mastoid process, and a small suprameatal spine. Each of these traits characterizes the Dinaledi *H. naledi* sample (*Berger et al., 2015*; *Laird et al., 2017*). Some of these traits occur individually in other species, including *H. erectus*, *H. habilis*, *H. rudolfensis*, and *Australopithecus sediba*, but they have never been found in combination except in *H. naledi* (*Figure 7*).

The maxilla and mandible of LES1 are also consistent with the Dinaledi *H. naledi* sample (*Supplementary file 1*; *Figures 8*, *9*, *10* and *11*). The maxilla has a mediolaterally flattened subnasal region, a parabolic dental arcade, and an anteriorly shallow palate. The mandible of LES1 has a gracile mandibular corpus, a vertical mandibular symphysis with weak mentum osseum, a steeply inclined lingual alveolar plane, weak inferior and absent superior transverse tori, continuous and deeply excavated anterior and posterior subalveolar fossae, mental foramina positioned above mid-corpus height, well defined ectoangular and endoangular tuberosities, and a root of the ascending ramus that originates at the mesial border of the $M_2$. Again, many of these traits can be found individually in other hominin species, but in combination, they are uniquely found in *H. naledi*.

The teeth exhibit moderate occlusal wear on the second and third molars, trending toward near-complete dentine exposure on the occlusal surfaces of first molars and substantial removal of occlusal detail of the anterior dentition. The dental morphology of LES1 is entirely consistent with the

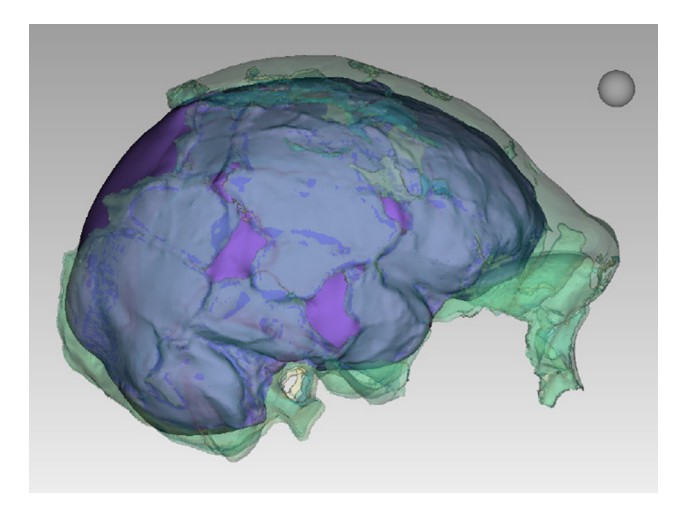

**Figure 6.** Digital reconstruction of endocranial volume in LES1. The refitted calvaria was mirrored and filled, resulting in a volume estimate of 610 ml. Scale sphere = 10 mm.

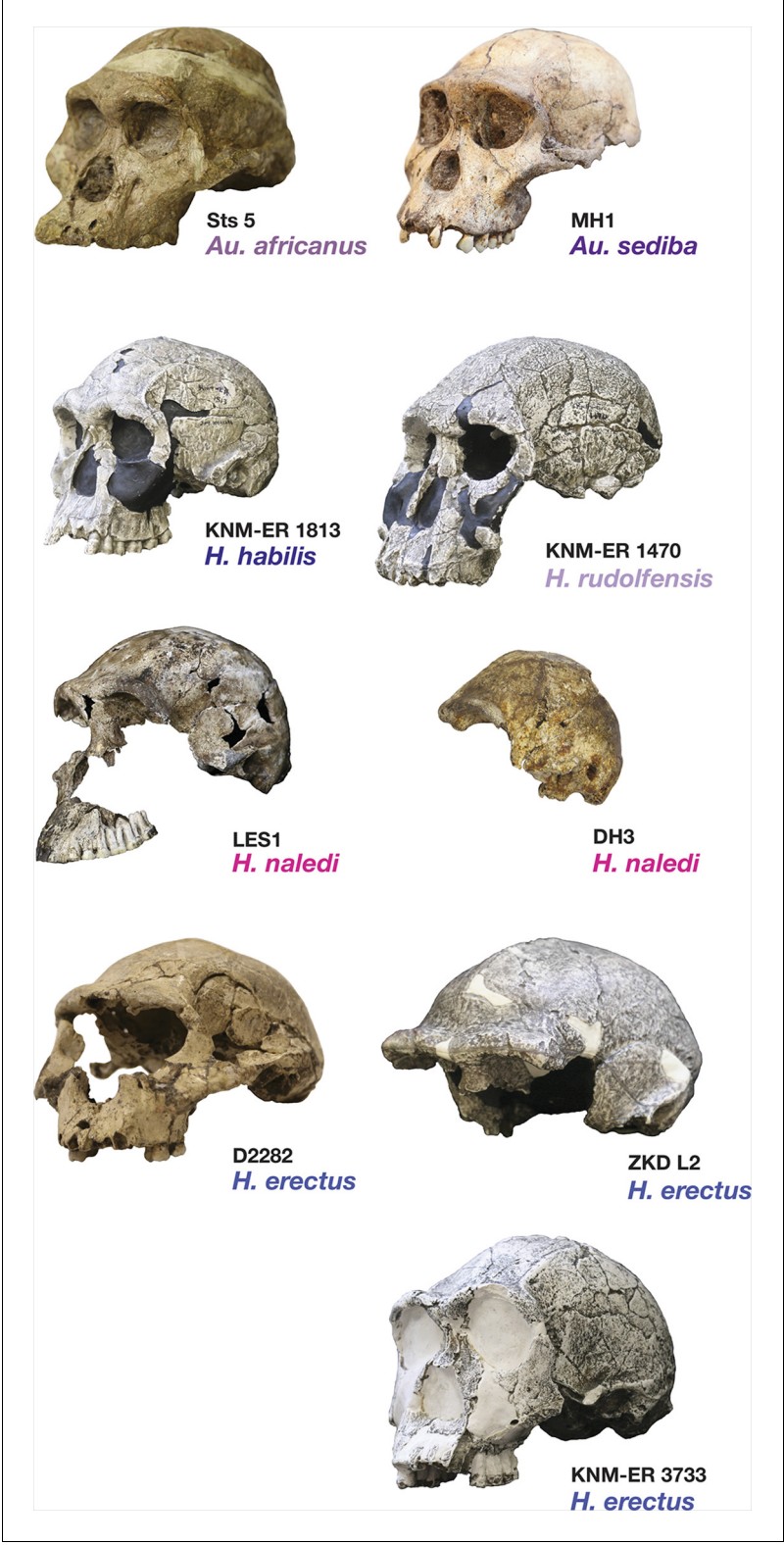

**Figure 7.** Frontal and vault morphology in *H. naledi* compared to that in other hominin species. Several of the crania pictured here are similar to *H. naledi* in endocranial volume, including Sts 5, MH1, KNM-ER 1813, and D2282, representing four different species. However, these skulls contrast strongly in other features. *H. erectus* is highly variable in size, as illustrated here by D2282 from Dmanisi, Georgia, one of the smallest and earliest *H. erectus* crania, and the L2 cranium from Zhoukoudian, China, one of the largest and latest *H. erectus* specimens. *Figure 7 continued on next page*

*Figure 7 continued*

The relatively early KNM-ER 3733 has a size and endocranial volume close to the mean for *H. erectus*. Cranial remains that are attributed to *H. erectus* share a combination of anatomical features despite their diversity in size. Many such features of *H. erectus* are also shared with *H. naledi*, *H. habilis*, or *Au. sediba*, and notably, the differences in the frontal and vault between KNM-ER 1813 (*H. habilis*) and KNM-ER 1470 (*H. rudolfensis*) are mostly features that the smaller KNM-ER 1813 shares with *H. naledi*, *H. erectus*, and *Au. sediba*. The *H. naledi* skulls share some aspects of frontal morphology with *Au. sediba*, *H. habilis* and *H. erectus* that are not found in *Au. africanus* or *H. rudolfensis*, including frontal bossing and a supratoral sulcus. Two additional traits of the *H. naledi* anterior vault are shared with *Au. sediba* and *H. erectus*: slight postorbital construction and a posterior position of the temporal crest on the supraorbital torus. More posteriorly on the vault, *H. naledi* further shares an angular torus with *H. erectus*, and some individuals also have sagittal keeling. Both of these traits are also present in some archaic humans. Some *H. naledi* crania, such as DH3, are substantially smaller than any *H. erectus* cranium, and the small size and thin vault bone of even the largest *H. naledi* skull, LES1, are outliers compared to *H. erectus*, matched only by some Dmanisi crania. The facial morphology of *H. naledi* is more distinct from those of *H. erectus* and *H. habilis*. The nasal bones of LES1 do not project markedly anteriorly, although like many specimens of *H. erectus*, LES1 has a projecting nasal spine. LES1 has a relatively flat lower face, with a transversely concave clivus and incisors that project only slightly past the canines. This morphology is similar but less extreme than that found in KNM-ER 1470 of *H. rudolfensis*, and is not shared with the other species pictured here. *H. naledi* has several distinctive features of the temporal bone that are absent from or found in only a few specimens of the other species pictured, including a laterally inflated mastoid process (comparable to some specimens of *Au. afarensis*), a weak or absent crista petrosa (comparable to *Au. afarensis*), and a small external auditory meatus (comparable to KNM-WT 40000 of *Kenyanthropus platyops* [**Leakey et al., 2001**]). In this illustration, KNM-ER 1813, KNM-ER 1470, KNM-ER 3733, and ZKD L2 are represented by casts. Because these images are in a nonstandard orientation, scale is approximate.

Dinaledi sample of *H. naledi* (**Figures 10** and **12**). The mesiodistal and buccolingual (or labiolingual) crown dimensions of all the LES1 teeth fall within the range of the Dinaledi dental sample, except

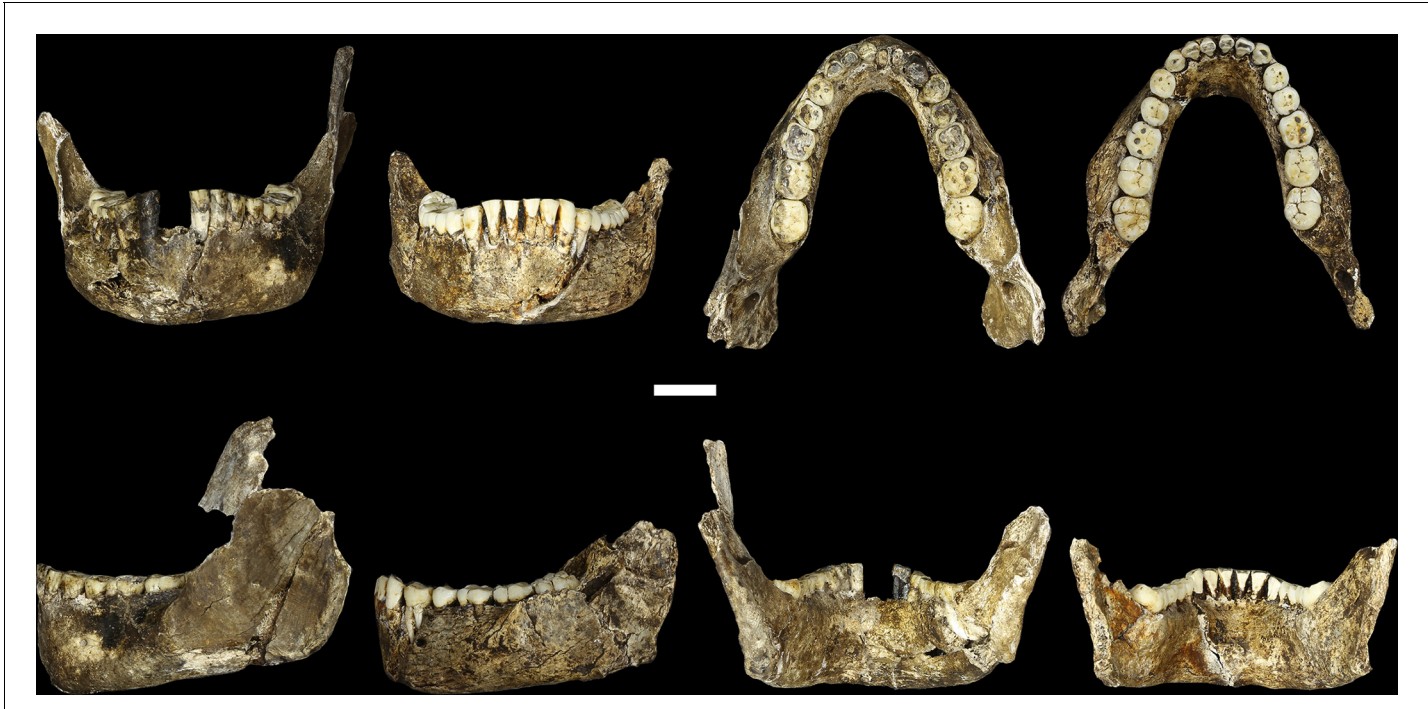

**Figure 8.** LES1 mandible compared to the DH1 holotype mandible of *H. naledi*. In each pair, LES1 is on the left and DH1 on the right. Top left: anterior view. Top right: occlusal view. Bottom left: left lateral view. Bottom right: posterior view. Scale bar = 2 cm.

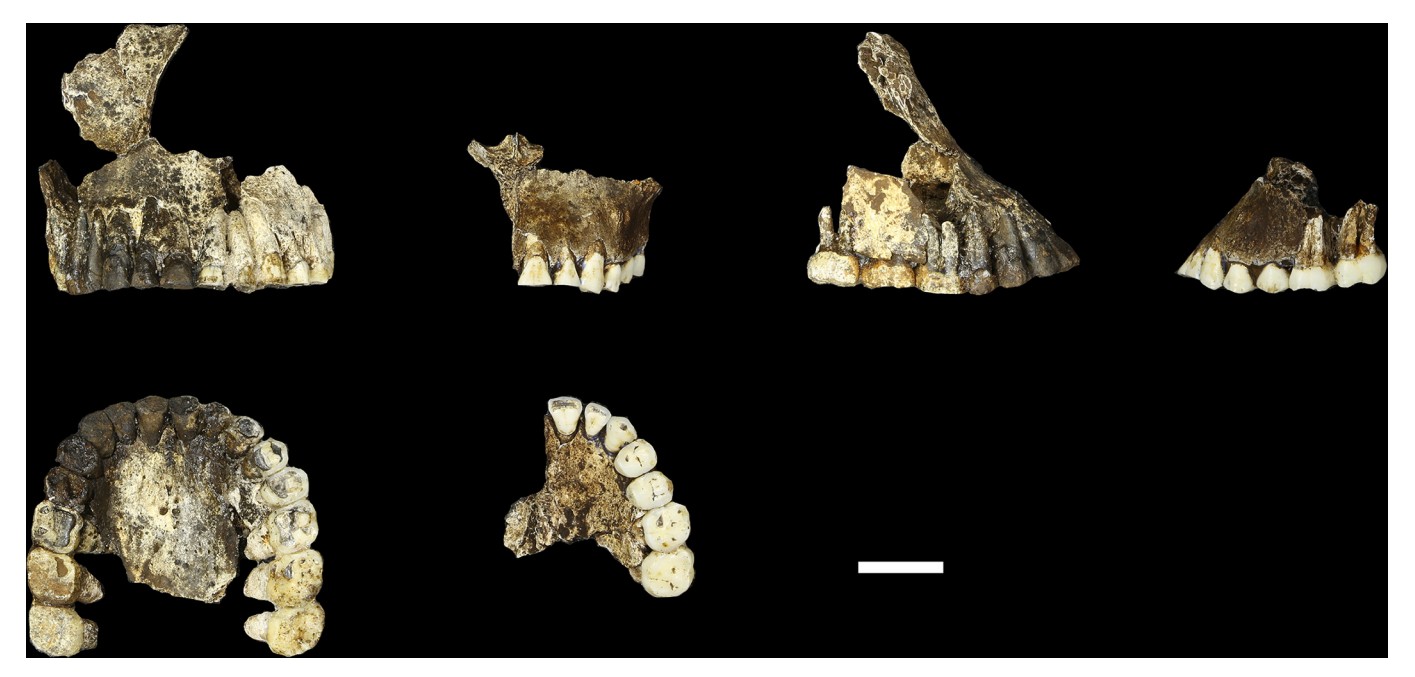

**Figure 9.** Comparison of LES1 maxilla to the DH1 holotype maxilla of *H. naledi*. In each pair, LES1 is on the left and DH1 on the right. Top left: anterior view. Top right: right (LES1) and left (DH1) lateral view. Bottom: occlusal view. Scale bar = 2 cm.

for those teeth where interproximal wear has clearly reduced the mesiodistal dimension (*Table 2*; *Figure 13*). The $P_3$ crowns are worn, but they are roughly symmetrical about their mesiodistal axis in occlusal view; they are fully bicuspid and multirooted, with a smaller circular mesiobuccal root and larger, more platelike, distal root. This configuration is repeated throughout the Dinaledi dental assemblage. The shared overall $P_3$ morphology of LES1 and the Dinaledi sample is distinctive in *H. naledi* and not observed in other species of hominins (*Figure 12*; *Berger et al., 2015*). The $P_3$ and $P_4$ are both three-rooted, with two ovoid roots present buccally and a larger ovoid root present lingually. The roots are not widely splayed as in some other multi-rooted hominins, and especially for the $P_4$, the buccal roots are closely packed in buccal view. This root configuration is seen in the *H. naledi* type specimen, U.W. 101–1277. The mandibular canine crowns have asymmetrically placed crown shoulders, with the mesial more apically placed than the distal. Further, the distal shoulder is formed by an accessory cuspule. These features are strongly distinctive in *H. naledi* (*Berger et al., 2015*), with only a few specimens of *H. erectus* approaching this canine configuration. None of the molars exhibit any evidence of supernumerary cusps, and cingular features, such as the protostylid and Carabelli's feature, are either absent or weakly developed and are expressed independently of the grooves of the crown. The molar size gradient in the LES1 mandible is $M_1 < M_2 < M_3$ as in the Dinaledi Chamber sample of *H. naledi* (*Figure 10*). The Dinaledi Chamber includes no maxillary dentition with all three molars in place, but U.W. 101–1269 is a $LM_3$ that exhibits a mesial interproximal facet that matches the distal facet of the $LM_2$ present in the U.W. 101–1277 (DH1) maxilla. If these specimens do represent a single individual, then the maxillary molar gradient for this specimen would be $M_1 < M_3 < M_2$, which is also seen in the LES1 maxilla. In total, these dental features are within the known range for *H. naledi* in every instance and distinguish LES1 clearly from all other hominin species.

The LES1 cranium does exhibit some traits that differ from comparable examples in the Dinaledi Chamber. The cranium is slightly larger overall, with an estimated endocranial volume of approximately 610 ml, and this larger size is reflected in the external vault measurements. Previously, the largest known *H. naledi* endocranium was DH1 at approximately 560 ml (*Berger et al., 2015*). LES1 contrasts in morphological features with the small DH3 cranium in ways that have been observed

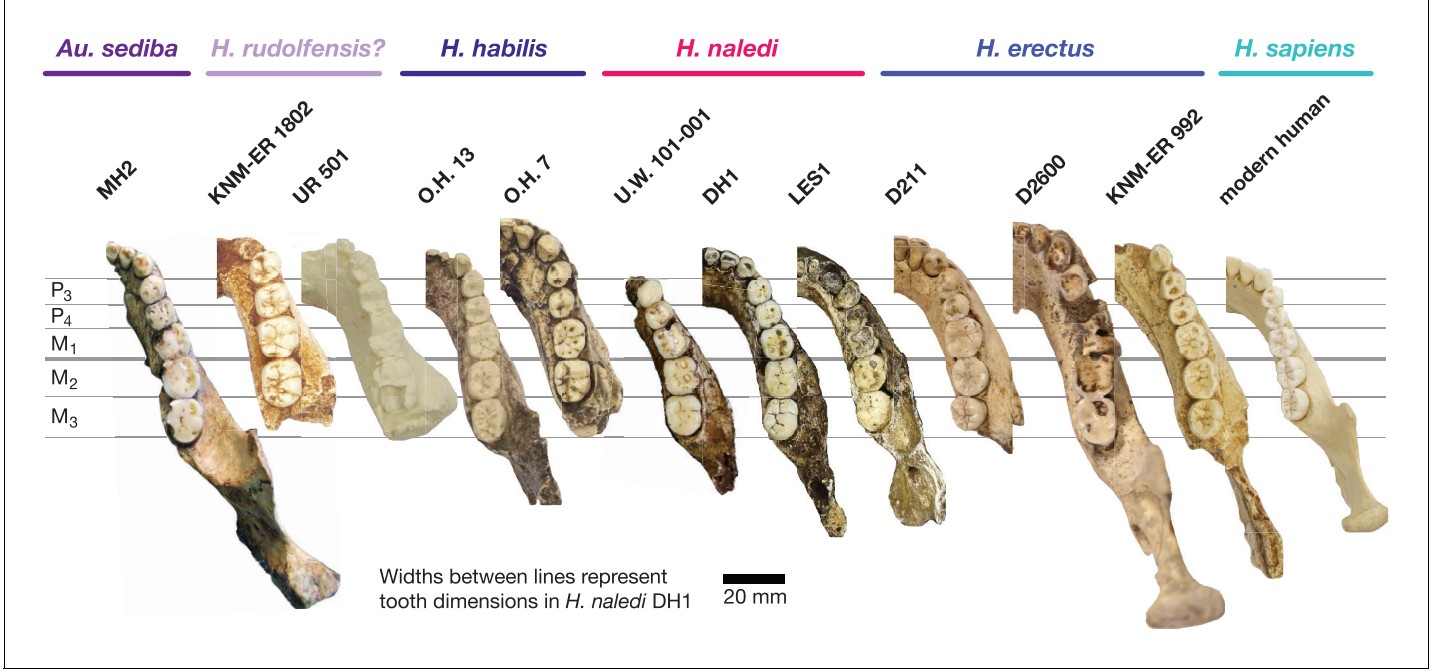

**Figure 10.** Mandibular and dental anatomy in *H. naledi* compared to other species of *Homo*. Right demi-mandibles attributed to *H. rudolfensis*, *H. habilis*, *H. naledi*, *H. erectus*, and *H. sapiens* are pictured. All mandibles are aligned using the line marking the distal edge of the first molar. Each of the six horizontal lines corresponds to the edges of teeth in the DH1 mandible, the holotype specimen of *H. naledi*, with corresponding teeth labeled to the left. Using these lines, it is apparent which specimens have longer premolars and first molars, and which have longer second and third molars compared to DH1. The dentition of the LES1 mandible has been affected by interproximal wear, resulting in shorter mesiodistal measurements. Mandibular morphology and dental proportions vary slightly among most species of *Homo*, particularly in comparison with the large differences in dental proportions among species of *Australopithecus* and *Paranthropus*. Still, *H. naledi* is clearly distinguishable from other species of *Homo* (*Berger et al., 2015*; *Laird et al., 2017*). Fossils of *H. rudolfensis*, *H. habilis*, and *H. erectus* differ from *H. naledi* in the proportions of different parts of the postcanine tooth row and in features of the mandibular corpus. **H. erectus.** While fossils attributed to *H. erectus* vary in dental proportions, the early African and Georgian fossil specimens (here represented by KNM-ER 992, D211 and D2600) have larger first molars than *H. naledi*, comparable premolar sizes, and highly variable second and third molar sizes. The mandibles attributed to *H. erectus* mostly have greater corpus height than *H. naledi* mandibles and are highly diverse in corpus breadth, symphyseal thickness, and robusticity. Many have a strong post-incisive planum, most obvious in D2600 (shown). All three also differ from *H. naledi* in the crown complexity of their molars and premolar morphology, as illustrated in more detail in *Figure 12*. Some specialists would attribute these three mandibles of *H. erectus* to three different species. **H. habilis.** The two Olduvai mandibles of *H. habilis* are themselves quite different from each other in size. Both have similar dental proportions to *H. naledi* with bigger teeth across the postcanine dentition. *Tobias (1967)* viewed O.H. 13 as being similar to *H. erectus* and described it as an 'evolved *H. habilis*'. Its occlusal morphology and dental proportions do resemble KNM-ER 992 (*Wood, 1991*), although the mandibular corpus is thinner and shallower, with a curved base in lateral profile. A strong post-incisive planum is evident in both mandibles. **H. rudolfensis.** The KNM-ER 1802 and Uraha (UR 501) mandibles have often been attributed to *H. rudolfensis*, although both attributions may be doubtful (*Leakey et al., 2012*). However, both lack any special similarities with contemporary australopiths and represent a megadont early *Homo* morphology with corpus size and robusticity much greater than those of *H. naledi*. **Au. sediba.** Molar sizes in the MH2 mandible are around 1 mm larger than the average for *H. naledi*, but the proportions are very similar to those of *H. naledi*, and like *H. naledi*, MH2 has a weak post-incisive planum and a small symphysis area. **H. sapiens.** The modern human mandible shown here, from a recent South African individual, has similar first molar size to the *H. naledi* mandibles, but much smaller premolars and second and third molars. The crown complexity in this individual, which is not unusual for African population samples, is substantially greater than evidenced in *H. naledi*. The mandibular corpus is smaller and much less robust than *H. naledi*. KNM-ER 1802, UR 501, O.H. 13, O.H. 7, and KNM-ER 992 are illustrated here with casts; the remainder are original specimens. The left side of O.H. 7 is shown here mirrored.

when comparing male individuals with female individuals in other hominin species. The supraorbital torus of LES1 is more pronounced than that in the small DH3 individual. LES1 has a stronger supra-mastoid/suprameatal crest, a larger mastoid process, and a more marked pterygoid insertion when compared to the U.W. 101–361 mandible. Although LES1 is outside of the endocranial volume range of specimens presently attributed to *H. naledi*, the larger size and more robust features of LES1 are consistent with the combination of cranial and mandibular characters in *H. naledi*.

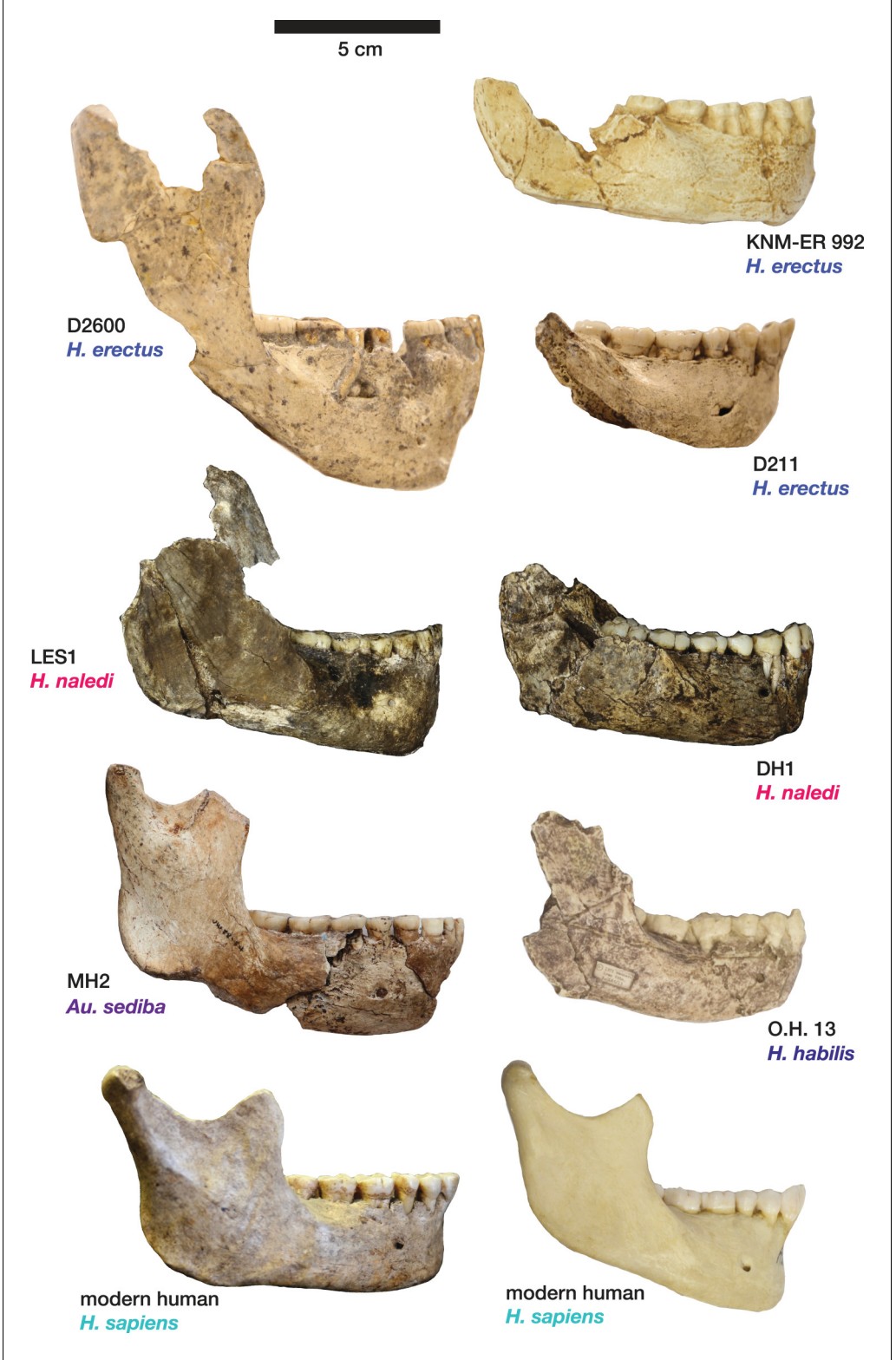

**Figure 11.** Comparison of *H. naledi* mandibles to other hominin species, from lateral view.  The DH1 holotype mandible and the LES1 mandible of *H. naledi* have a moderately deep mandibular corpus compared to other species of *Homo*; the LES1 mandible has a slightly greater corpus height anteriorly (at P$_3$) than posteriorly (at M$_2$). LES1 has rather a high coronoid process; the height of the condyle was probably lower than this. The mental foramen is at the midpoint or slightly higher in both *H. naledi* mandibles, and in both, the symphysis is nearly

*Figure 11 continued*

vertical. These features vary substantially within *Homo*. Modern humans (bottom) typically have a chin, but otherwise vary substantially in corpus height, whether the base of the corpus is parallel with the alveolar portion or with the occlusal surfaces of the teeth. Here that variability is illustrated with two modern human mandibles of male individuals, one from island Melanesia (left), and one from southern Africa (right). *H. erectus* exhibits very extensive variation in corpus height and thickness. D2600 (shown) is extremely thick and robust, but is not an outlier; other *H. erectus* mandibles approach or equal its corpus dimensions. The position of the mental foramen also varies, as does the relative anterior versus posterior corpus height and the symphyseal profile, from more sloping to near vertical (as illustrated by KNM-ER 992, although this specimen is damaged at the symphysis). MH2 (*Au. sediba*) has comparable corpus height and robusticity to the *H. naledi* mandibles, with a more sloping symphysis. O.H. 13 is a more gracile mandible than the *H. naledi* specimens in many respects. It has a curved base and a sloping symphysis. The more complete left side of LES1 is shown here and mirrored for comparison to other specimens. KNM-ER 992 and O.H. 13 are represented here by casts.

## Comparative cranial anatomy

The comparative anatomy of the *H. naledi* cranial remains from the Dinaledi Chamber was presented in detail by *Laird et al. (2017)*, and morphometric comparative analyses of that collection and of other hominin samples were presented by *Schroeder et al., 2017*. Additionally, *Rightmire et al., 2017* addressed the morphological features of *H. naledi* in comparison with the Dmanisi fossil crania, in particular the robust D4500 cranium. The anatomy of the LES1 cranium reinforces the conclusions of those studies in most respects (*Supplementary file 1*, *2*; *Figure 7*).

Crania of *H. naledi* are most similar in cranial vault shape to other *Homo* or *Homo*-like australopith crania with small endocranial volume, including D2700, D2280, MH1, KNM-ER 1470, and KNM-ER 1813 (*Schroeder et al., 2017*). These shape similarities do not reflect small size alone: for example, *H. naledi* cranial material is quite distinct in shape from LB1, and the small DH3 calvaria of *H. naledi* is also notably different from KNM-ER 42700 in shape analyses (*Schroeder et al., 2017*). Additional differences between *H. naledi* and other small specimens of *Homo* are evident among the morphological characters of the skull (*Laird et al., 2017*). Compared to specimens attributed to *H. erectus*, *H. habilis* or *H. rudolfensis*, the temporal bone of *H. naledi* exhibits a small external auditory meatus, a shallow and relatively narrow mandibular fossa, a small postglenoid process, and a laterally inflated mastoid process. The features of the occiput that distinguish *H. naledi* from *H. erectus* (*Laird et al., 2017*) are not part of the preserved sections of LES1. However, the maxilla of LES1 is better preserved than DH1, and like the latter specimen, presents a transversely flat nasoalveolar clivus, similar to *H. rudolfensis* but not *H. erectus* or *H. habilis*, a shallow anterior palate, unlike *H. erectus* or *H. habilis*, and an anteriorly projecting anterior nasal spine, comparable to *H. erectus* but not present in *H. habilis* or *H. rudolfensis*. The LES1 cranium is similar to the Dinaledi *H. naledi* sample in its morphological differences from the *H. floresiensis* LB1 cranium (*Berger et al., 2015*). All *H. naledi* crania are estimated to have been larger than LB1 in endocranial volume, while LES1 and the other *H. naledi* cranial remains lack the reduced cranial height, marked frontal keel, canine juga and anterior pillars of the LB1 cranium. The cranial anatomiesof *H. naledi* and LES1 share a unique set of traits that otherwise distinguish *Homo* species from each other.

The LES1 mandible shares very similar overall dimensions, shape, and morphological features with DH1 from the Dinaledi Chamber (*Supplementary file 1*; *Figures 8*, *10* and *11*). Comparative analysis of overall mandibular shape places *H. naledi* closer to australopith mandibles such as Sts 36 and Sts 52b than to any specimens of *H. habilis*, *H. rudolfensis*, or *H. erectus*, despite the large range of shape variation among *H. erectus* mandibular specimens (*Schroeder et al., 2017*). Dmanisi mandibular specimens, including D2735 and D2600, are different from the DH1 mandible despite the similarity in vault shape between D2700 and D2280 and *H. naledi* DH3, and these mandibular differences are likewise reflected in the LES1 mandible. The morphological features of the LES1 mandible align it clearly with DH1 and other partial *H. naledi* mandibles from the Dinaledi Chamber, setting it apart from other species of *Homo*, including those with similar-sized molars (*Figures 10* and *12*). Unlike the *H. floresiensis* mandibles LB1 and LB2, the mandibular symphyses of LES1 and DH1 have steeply inclined posterior faces that lack any post-incisive planum or superior transverse torus. The mandibular molar gradient of *H. naledi* and the morphology of the mandibular premolars also

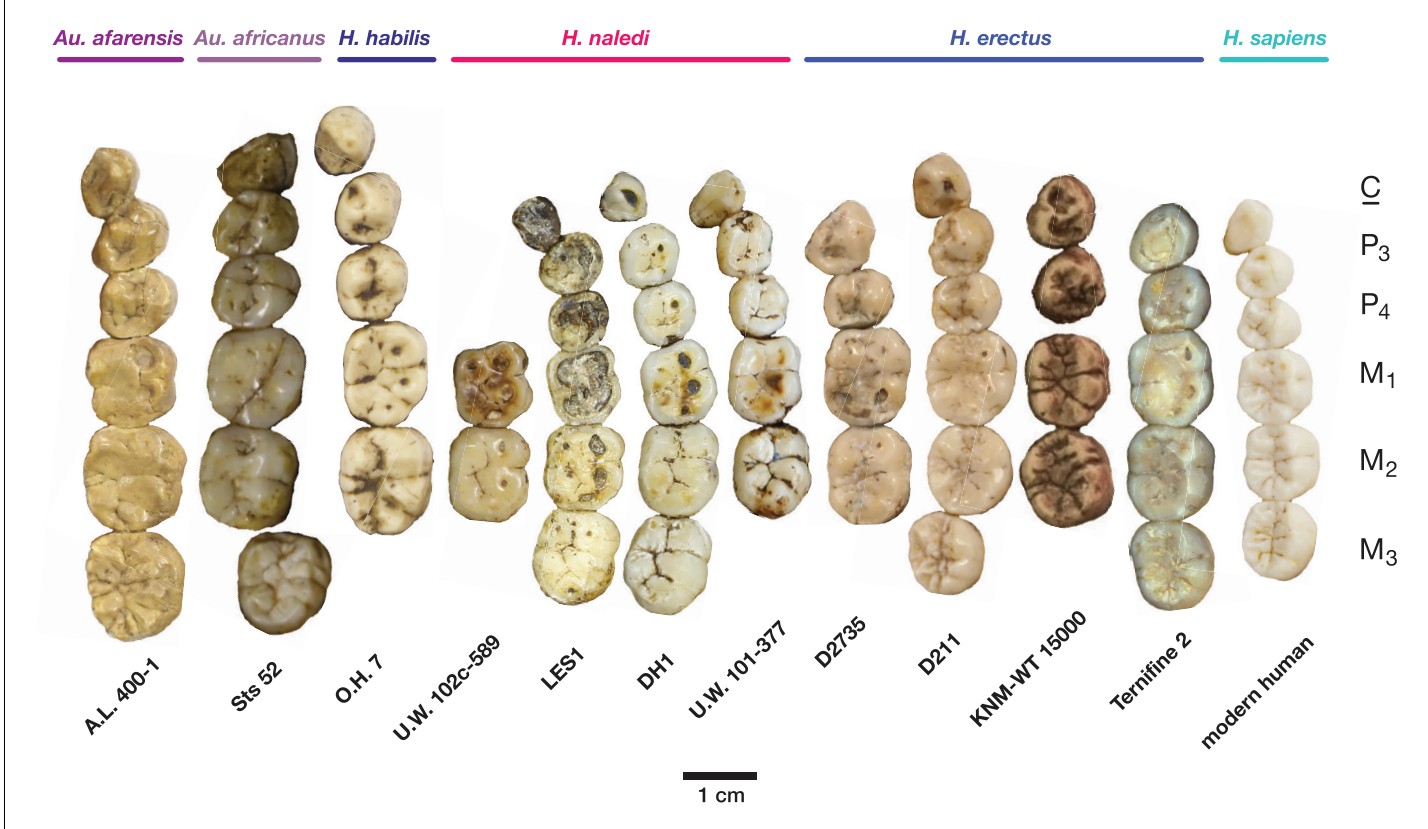

**Figure 12.** Occlusal view of *H. naledi* mandibular teeth compared to those of other hominins. Teeth from the canine to the third molar are shown, if present, in the orientation in which they are found within the mandible. All individuals are aligned vertically by the distal margin of the first molar. Mandibles from the Lesedi Chamber, U.W. 102 c-589 and LES1 are shown next to DH1 and U.W. 101–377 (*Berger et al., 2015*). The mandibles illustrated from *H. erectus* have relatively little occlusal wear, so their morphology can be seen more clearly than that of worn mandibles. The immature U.W. 101–377 (*H. naledi*) is comparable in developmental age and wear to O.H. 7 (*H. habilis*), as well as to D2735 and KNM-WT 15000 (*H. erectus*). When compared to *H. habilis*, *H. erectus*, and australopiths, *H. naledi* is notable for its relatively small first molars, its relatively small canines, and its lack of supernumerary cusps and crenulation on the molars. The complexity of molar cusp and groove patterns is especially evident in the chronologically early *H. erectus* specimens from Africa and Georgia shown here. For example, the unworn $M_2$ of the immature U.W. 101–377 mandible of *H. naledi* has a relatively simple crown anatomy with very little wrinkling or crenulation. By comparison, the $M_2$ of D2735, D211, and KNM-WT 15000, all with minimal occlusal wear, show extensive crenulation and supernumerary cusps. Canine size and molar crown complexity vary substantially among modern human populations, but the southern African individual illustrated on the right is not atypical for its population, and has greater molar crown complexity and larger canine dimensions than any of the *H. naledi* mandibular dentitions. The morphology of the third premolar varies extensively among these hominin species and within *H. erectus*. The *H. naledi* $P_3$ anatomy can be seen clearly in the immature U.W. 101–377 individual. It is characterized by roughly equally prominent lingual and buccal cusps and an expanded talonid. In *H. naledi*, this tooth is broadly similar in morphology and size to the $P_4$. This configuration of the $P_3$ is not present in the other species, with only KNM-WT 15000 exhibiting some expansion of the lingual cusp in what remains an asymmetrical and rounded $P_3$. A.L. 400–1, O.H. 7 and KNM-WT 15000 are represented by casts; The left dentition of U.W. 102 c-589 and O.H. 7 have been mirrored to compare to right mandibles. Images have been scaled by measured first molar dimensions.

distinguish it from *H. floresiensis* (*Brown and Maeda, 2009*; *Kaifu et al., 2011*, *2015*). The symphyseal morphology of *H. naledi* likewise distinguishes LES1 and DH1 from mandibular remains attributed to *H. habilis* and *H. rudolfensis,* such as OH 7, OH 13, KNM-ER 60000 and KNM-ER 1802.

## Claviculae

**U.W. 102a-021** is a nearly complete right clavicle, missing only the articular surface of the sternal end, where trabecular bone is exposed over the entire articular area, including a small bit of the anterior surface (*Figure 14*). The shaft is broken into two pieces near the midshaft but the two pieces conjoin cleanly. There is also a small bit of damage to the acromial end. On the posterior surface, the medial part of the crest for the conoid tubercle is broken off. The specimen exhibits a dark

**Table 2.** Dental measurements for Lesedi Chamber specimens.

| Specimen | Mesiodistal diameter | Buccolingual (or labiolingual) diameter |
|---|---|---|
| U.W. 102b-437 ldm$_2$ | 10.7 | 8.7 |
| U.W. 102b-503 RP$_4$ | 8.4 | 10.9 |
| U.W. 102b-515 LI$_2$ | 6.8 | 6.5[†] |
| U.W. 102b-178 LI$_2$ | 5.6 | 5.9 |
| U.W. 102b-511 LC$_1$ | 6.8 | 6.8[†] |
| U.W. 102 c-589 LM$_1$ | 11.4 | 10.6 |
| U.W. 102 c-589 LM$_2$ | 13.1 | 11.3 |
| LES1 maxillary | | |
| RI$_1$ | 7.6* | 6.9 |
| RI$_2$ | 6.8* | 7.0 |
| RC$_1$ | 7.5 | 8.7 |
| RP$_3$ | 8.1 | 10.8 |
| RP$_4$ | 8.1 | 11.3 |
| RM$_1$ | 10.6* | 11.8 |
| RM$_2$ | 11.7 | 12.7 |
| RM$_3$ | 11.4 | 12.7 |
| LI$_1$ | 7.4* | 6.9 |
| LI$_2$ | 6.1* | 6.8 |
| LC$_1$ | 7.4 | 8.7 |
| LP$_3$ | 8.0 | 10.9 |
| LP$_4$ | 8.1 | 11.3 |
| LM$_1$ | 10.7* | 11.9 |
| LM$_2$ | 12.1 | 12.8 |
| LM$_3$ | 11.4 | 13.6 |
| LES1 mandibular | | |
| RI$_1$ | 5.8* | 6.3 |
| RI$_2$ | 5.4* | 6.1 |
| RC$_1$ | 7.1 | 7.7 |
| RP$_3$ | 8.4 | 9.3 |
| RP$_4$ | 8.2 | 9.1 |
| RM$_1$ | 10.8* | 10.6 |
| RM$_2$ | 12.3 | 11.5 |
| RM$_3$ | 13.3 | 11.7 |
| LC$_1$ | 7.8 | 7.5 † |
| LP$_3$ | 8.4 | 9.3 |
| LP$_4$ | 8.2 | 9.1 |
| LM$_1$ | 11.2* | 10.6 |
| LM$_2$ | 12.3 | 11.5 |
| LM$_3$ | 13.3 | 11.7 |

*Denotes measurements where the tooth is extremely worn, and mesiodistal diameter reported here has not been corrected for the degree of wear.

†Denotes instances where we report a minimum value for labiolingual measurements because the crown is not complete or is broken.

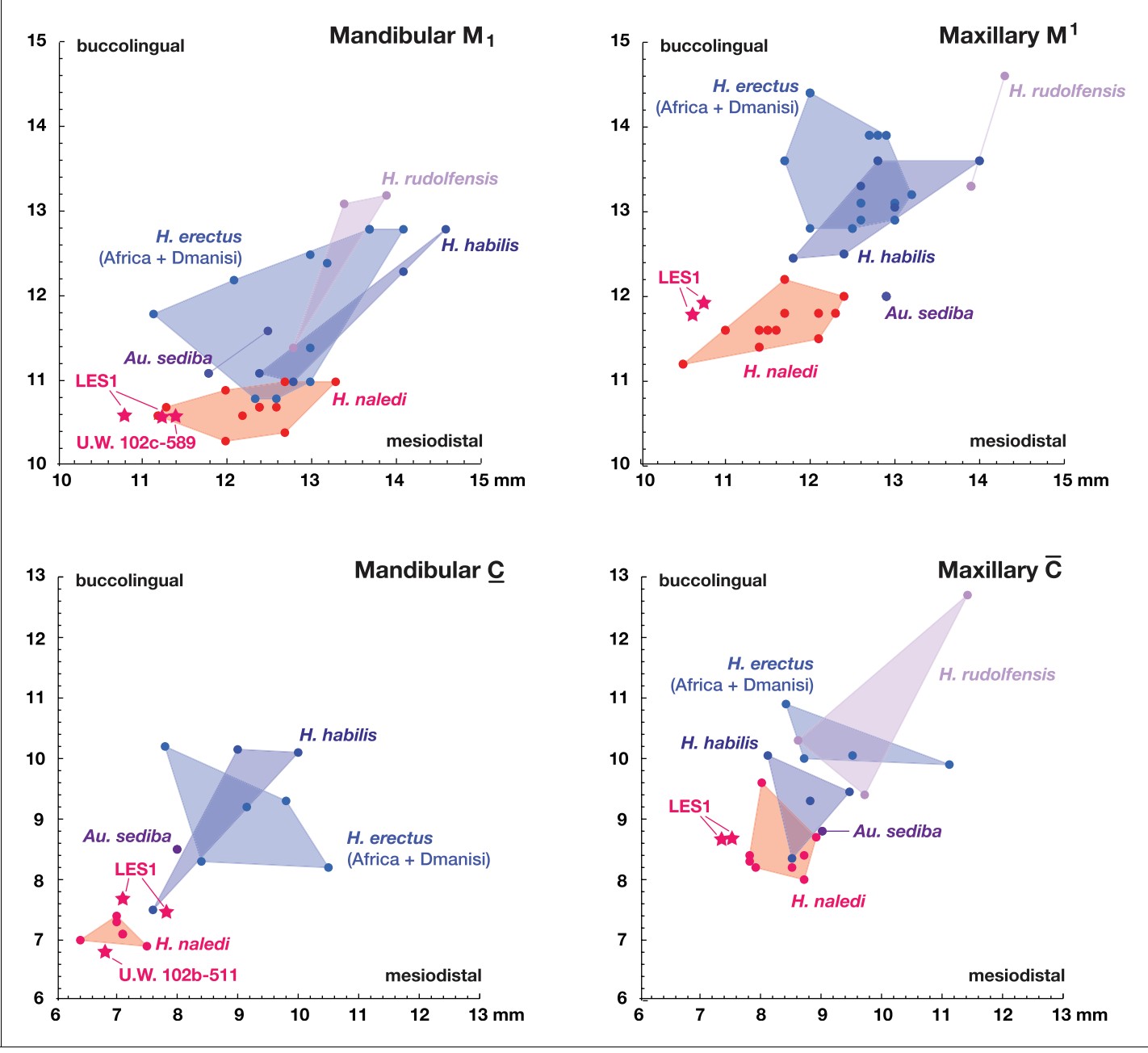

**Figure 13.** Metric comparisons of the Lesedi Chamber dental material. *H. naledi* is clearly differentiated in first molar and canine dimensions from other species with broadly similar cranial and dental morphology, including *Au. sediba, H. habilis, H. rudolfensis,* and early *H. erectus* samples from Africa and Georgia. The material from the Lesedi Chamber is within the range of or similar to *H. naledi* in these dimensions and well differentiated from the other samples. Top left: mandibular first molar dimensions. Top right: maxillary first molar dimensions. Bottom left: mandibular canine dimensions. Bottom right: maxillary canine dimensions. The LES1 first molars and maxillary canines have a substantial degree of interproximal wear, and the values plotted here are not corrected for this wear, which shortened the mesiodistal dimension by as much as a millimeter. The values plotted here should thus be regarded as minimum values. The *H. erectus* sample here includes specimens from the Lake Turkana area, Konso, Tighenif (Ternifine), Thomas Quarry, and Dmanisi; Asian *H. erectus* specimens are omitted. Attributions of *H. habilis* and *H. rudolfensis* specimens are indicated in the Materials and methods.

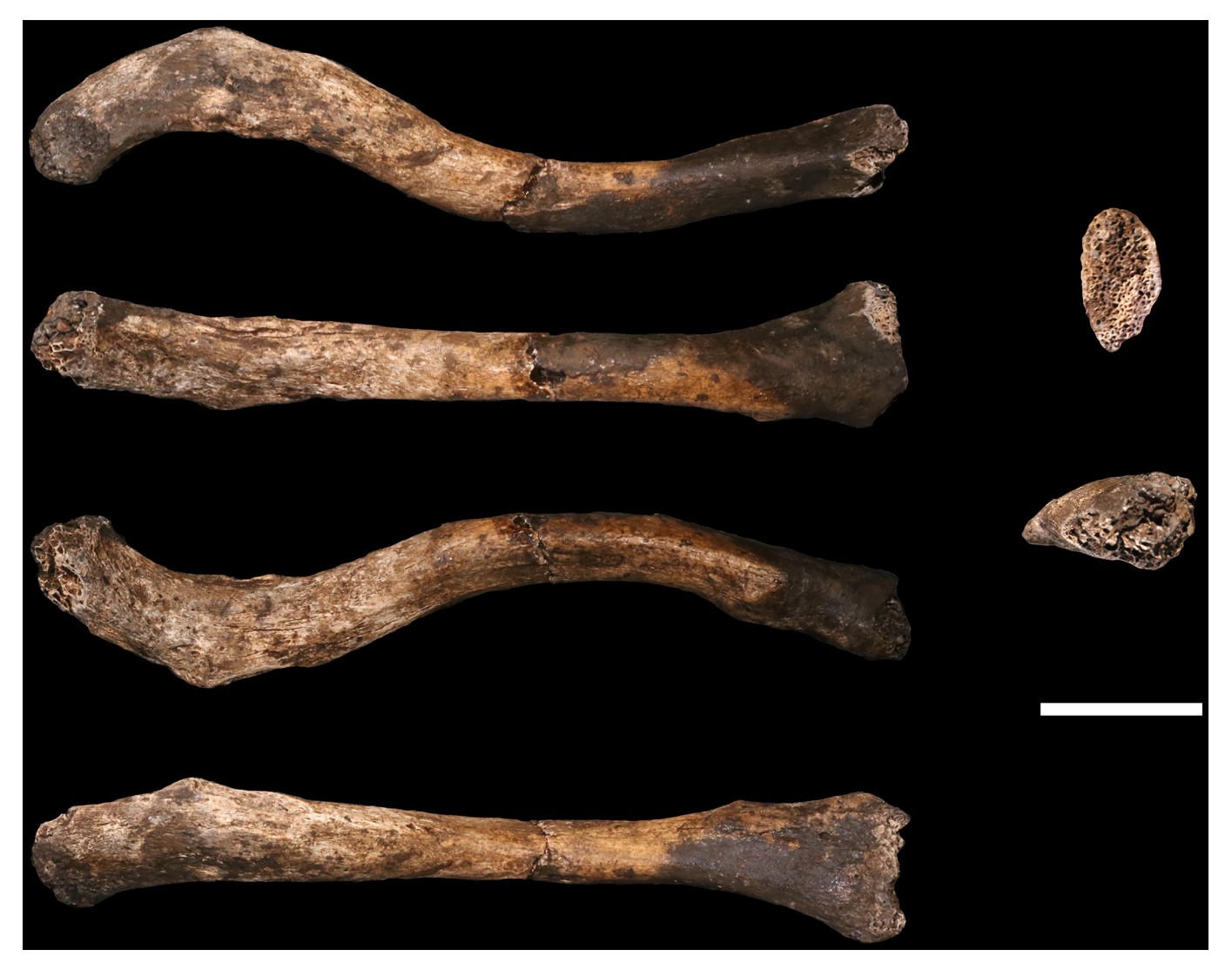

**Figure 14.** U.W. 102a-021 right clavicle from the Lesedi Chamber.  Left, from top: superior, anterior, inferior, posterior views. Right, from top: medial and lateral views. Scale bar = 2 cm.

surface coating on the anterior aspect of the sternal half and patchy areas of black staining on its acromial half. There are fine hairline longitudinal cracks on much of the acromial half of the bone.

U.W. **102a-206** is a ca. 41-mm-long fragment of left clavicular shaft, preserving the midshaft region (based on anatomical comparisons with U.W. 102a-021). The shaft anteroposterior (AP) and superoinferior (SI) dimensions are slightly smaller than those of the right side clavicle at this position. The fragment compares favorably to U.W. 102a-021 in overall size, curvature, and shaft morphology. **U.W. 102a-239** is the acromial end of a left clavicle, including the lateral 51.5 mm, preserving the conoid tubercle (but not its medial crest) and the articular surface for the acromion of the scapula. This is slightly larger than the acromial end of U.W. 102a-021, but otherwise fairly similar in morphology.

## Comparative clavicular anatomy

The clavicular anatomy of *H. naledi* is comparable to that present in *Au. sediba* (*Churchill et al., 2013*) and *H. habilis* (*Oxnard, 1969*; *Ohman, 1986*; *Voisin, 2001*), suggesting a superiorly positioned shoulder (*Feuerriegel et al., 2017*). The U.W. 102a-021 and U.W. 102a-206 claviculae are

consistent with the morphology noted in the clavicular material from the Dinaledi Chamber. As with clavicular specimens from U.W. 101, the overall appearance of the clavicle is smooth (non-rugose), with only a few weakly developed entheses. The deltoid crest is present as a mildly rugose line on the anterior surface of the lateral curvature. The conoid tubercle appears well-developed and forms a posteriorly projecting flange, producing a pronounced border to a deep subclavian sulcus. However, unlike some specimens from the Dinaledi Chamber (such as U.W. 101–258), the conoid is not centrally positioned on the shaft, but rather occurs on the posterior margin.

## Humerus

**U.W. 102a-002** is a proximal shaft fragment of a right humerus (*Figure 15*). The head and greater tubercle are missing, as is all but the very distal base of the lesser tubercle. From this metaphyseal region, the fragment preserves approximately 50–60% of the shaft, with a total fragment length of 85 mm. **U.W. 102a-013** includes two fragments identified as humeral head, each with some articular subchondral bone, which may derive from the same element as U.W. 102a-002. They appear to be consistent in curvatures of the articular surface with U.W. 102a-257. The specimen is mostly coated with a brown to dark-brown mineral patina, the surface is unweathered with only slight surface removal on the distal end of the anterior surface. The breaks, both proximal and distal, are sharp.

U.W. **102a-257** is a fragment of left humerus, including the head and proximal shaft, and is largely complete from head to just around midshaft (*Figure 16*). There is corrosion to the superior aspect of the proximal articular surface (which precludes an accurate measurement of the superoinferior diameter of the head), and to the articular margin of the superolateral head. The surface of the specimen is otherwise very well preserved with no signs of weathering. A dark-brown to black patina covers much of the posterior surface of the shaft, wrapping around to the anterior surface on the most distal part. The proximal 40 mm or so of the lateral crest of the deltoid tuberosity is present. U.W. 102a-257 is consistent with the morphology and size of U.W. 102a-002, and they may represent left and right humeri of the same individual.

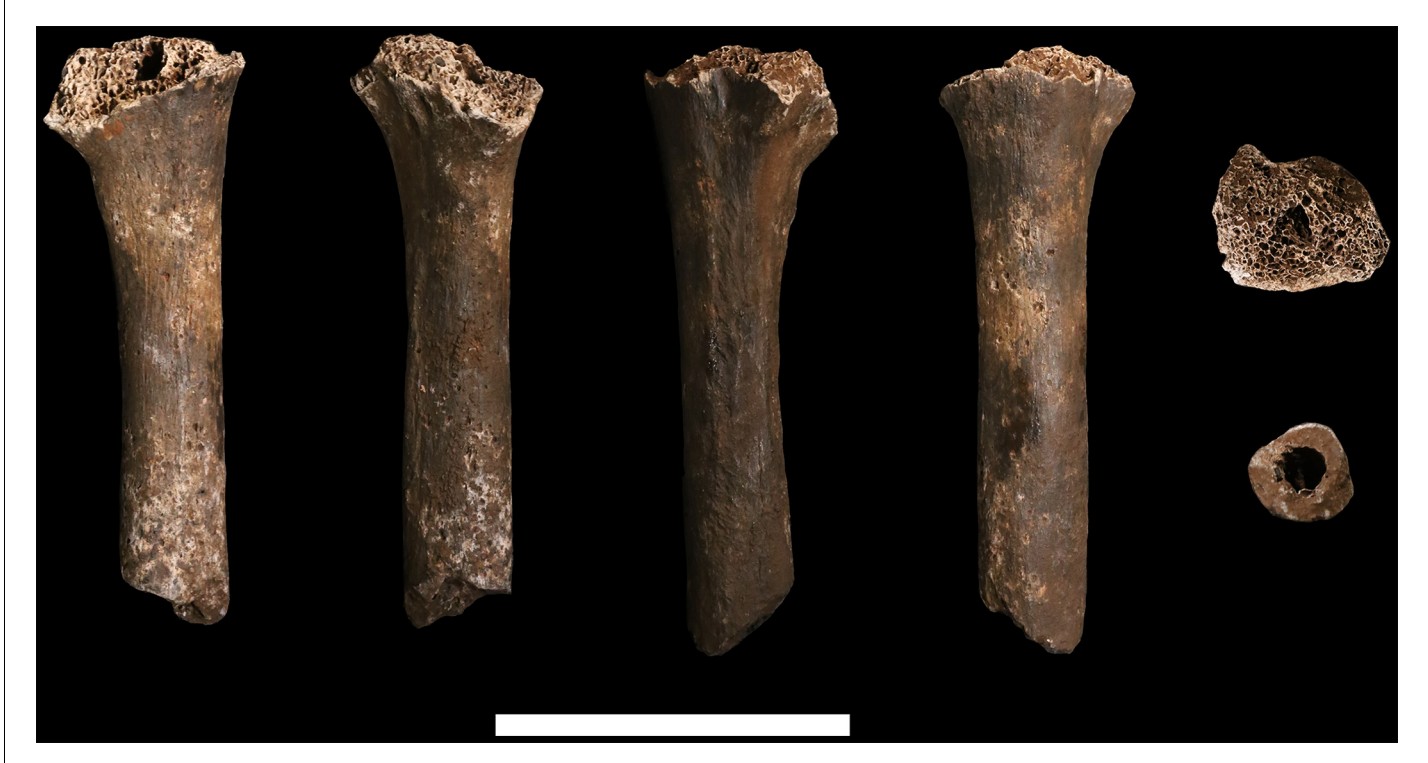

**Figure 15.** U.W. 102a-002 right humerus fragment. From left: posterior, medial, anterior and lateral views. Right, from top: Scale bar = 5 cm.

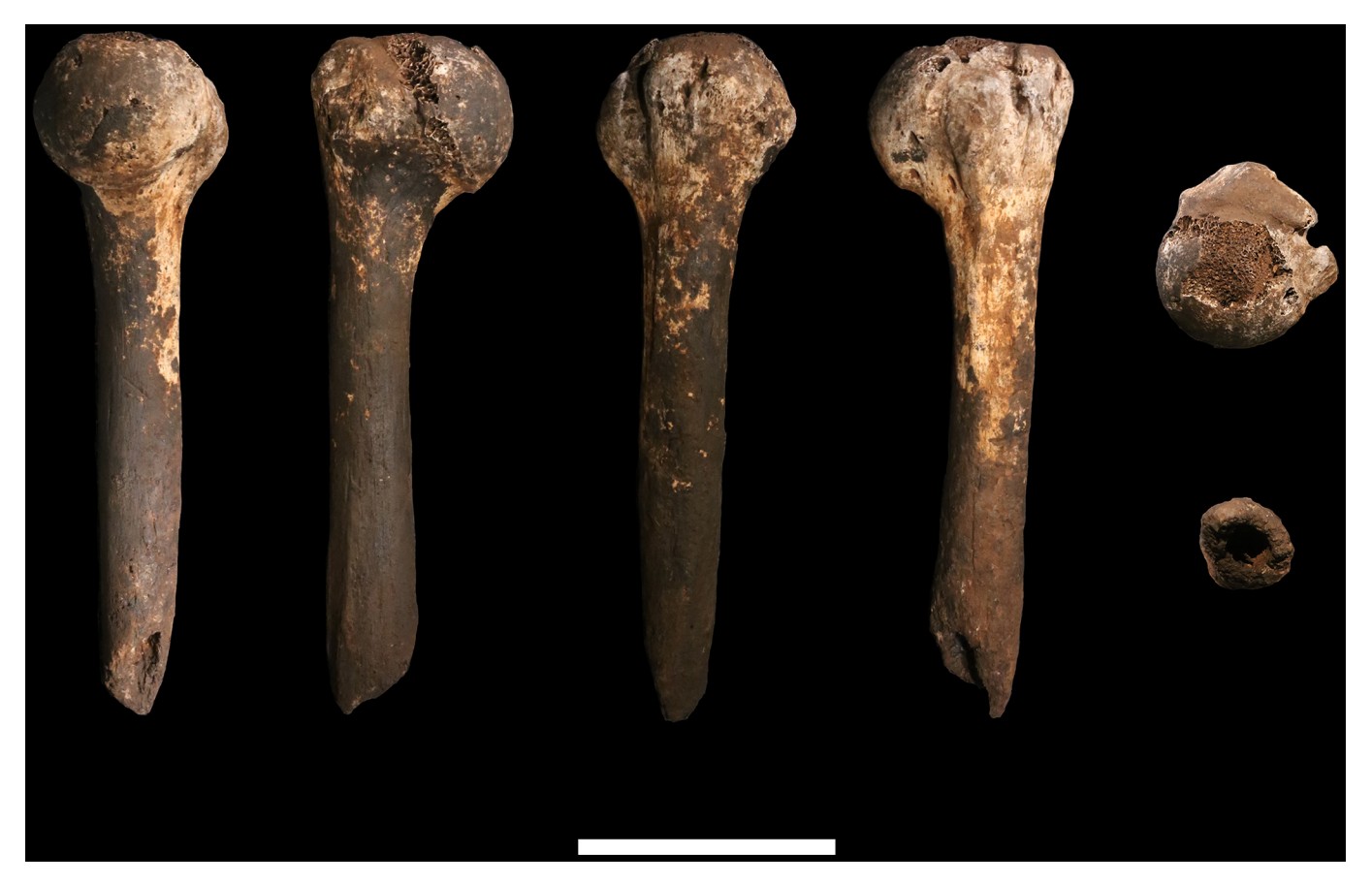

**Figure 16.** U.W. 102a-257 left proximal humerus fragment. From left: posterior, medial, anterior and lateral views. Top: proximal view. Bottom: distal view. Scale bar = 5 cm.

The proximal humerus fragments from the Lesedi Chamber have morphology consistent with the Dinaledi Chamber collection of *H. naledi*, both in the size of the head and in the shaft diameter. In both assemblages, the bicipital groove appears deep and narrow, and the lesser tubercle is projecting. The most distinctive aspect of the humerus material of *H. naledi* in comparison with other hominin species is the very low humeral torsion angle in the adult **U.W. 101–283** humerus, in which the head faces nearly directly posteriorly (*Feuerriegel et al., 2017*). This aspect cannot be assessed directly in the fragments from the Lesedi Chamber.

## Ulna

**U.W. 102a-015** is a right proximal ulna, on which much of the trochlear notch and olecranon process are preserved in addition to the proximal half of the diaphysis (*Figure 17*). There is erosion to the anterior tips and margins of the coronoid and olecranon process. The surface of the shaft is well-preserved and exhibits very slight hairline longitudinal cracks. The break to the distal end is sharp and cleanly transverse. The olecranon process is mediolaterally narrow and the trochlear notch appears to have opened anterosuperiorly, as in modern humans but not Neandertals. While there is only one fragmentary (and probably immature) proximal ulna from the Dinaledi Chamber, U.W. 102a-015 generally compares well with U.W. 101–560 in terms of morphology and gracility.

## Hand and wrist

Four adult hand and wrist elements have been recovered from the 102a locality (*Figure 18*). **U.W. 102a-028** is a right fourth metacarpal (RMc4), with a small base and a relatively radioulnarly broad

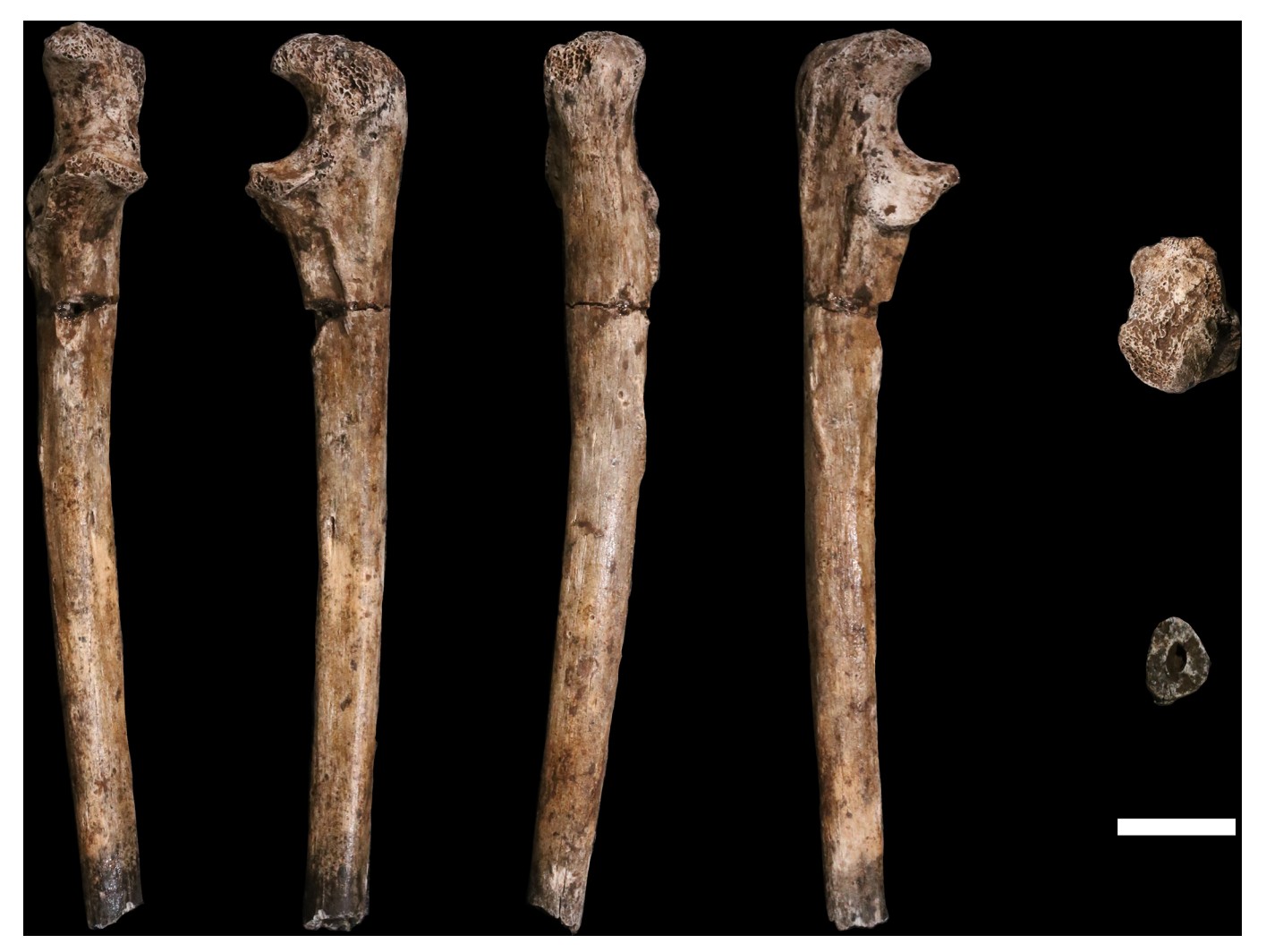

**Figure 17.** U.W. 102a-015 ulna fragment. From left: anterior, medial, posterior and lateral views. Right from top: proximal and distal views. Scale bar = 2 cm.

head (*Figure 15*). The metacarpal shaft shows substantial curvature and is relatively robust for its length, although it still falls within the upper range of variation found in modern humans (*Figure 16*). **U.W. 102a-117** is a complete right scaphoid; **U.W. 102a-476** is a complete right capitate; and **U.W. 102a-477** is a partial right lunate. The scaphoid, lunate, and capitate are consistent in size and appear to match each other when placed in anatomical articulation; the RMc4 is likewise a good match in size, with a lateral base matching in dorsopalmar contour the base of the capitate. Thus, these four bones may represent the right hand of one individual.

These four bones are qualitatively similar in overall shape to that described for *H. naledi*, but they are absolutely larger in most of their overall dimensions (*Kivell et al., 2015*). The lunate is missing a large portion of its articular surface for the radius and adjacent areas, precluding quantitative comparisons of its morphology. A canonical variates analysis of scaphoid and capitate comparative morphology in African apes and hominins demonstrates that the Dinaledi and Lesedi scaphoids and capitates fall together within a distinct space relative to other fossil and extant hominids. Along the first canonical axis, Dinaledi and Lesedi wrist remains cluster with modern humans and Neandertals because they all share derived features relative to those of African apes (*Figure 19*; *Supplementary file 3*). For instance, the scaphoid's trapezium facet extends further onto the

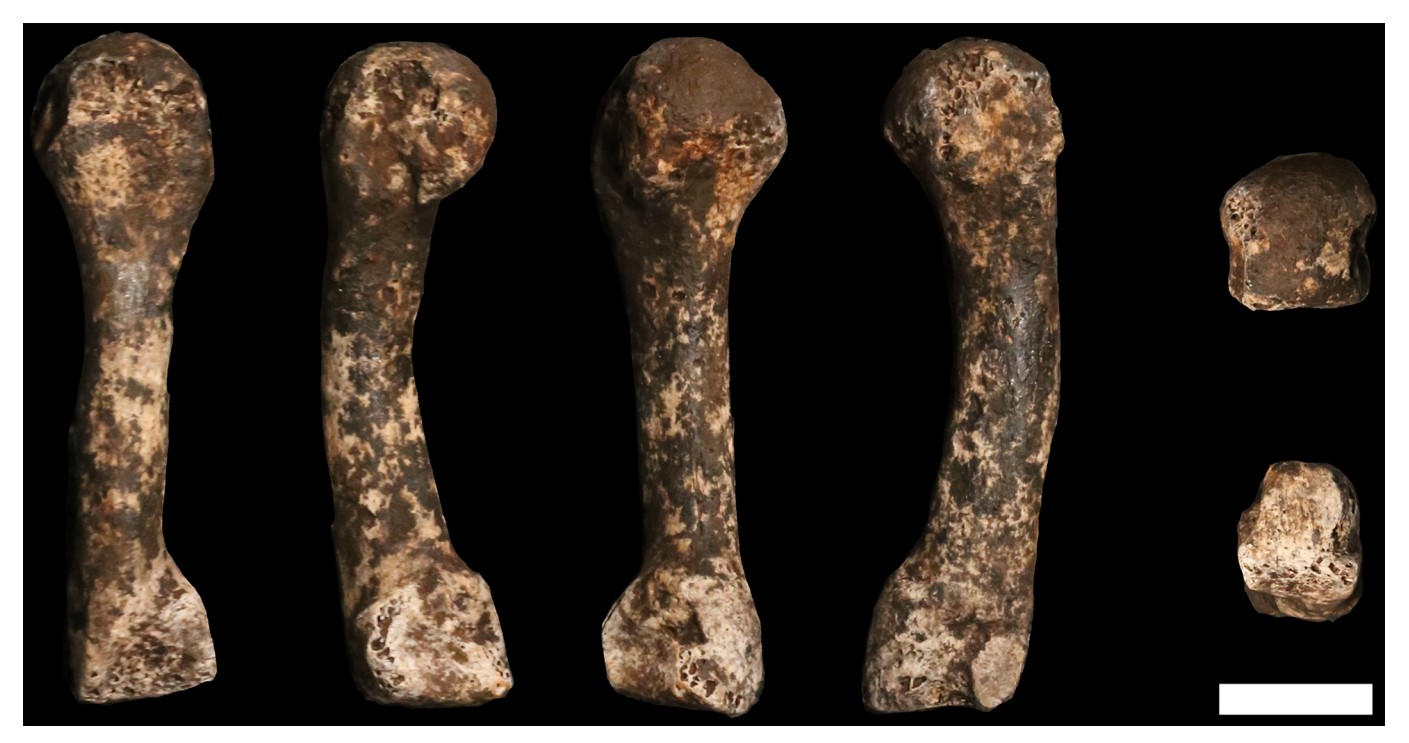

**Figure 18.** U.W. 102a-028 right fourth metacarpal. From left: dorsal, ulnar, palmar and radial views. Right from top: distal and proximal views. Scale bar = 1 cm.

tubercle, and together, the trapezium and trapezoid facets are relatively large, as in modern humans and Neandertals (*Supplementary file 3*). The Dinaledi and Lesedi scaphoid and capitate morphology are distinguished from those of modern humans and Neandertals on the second canonical axis because the Mc2 facet orientation in *H. naledi* is roughly intermediate between that of modern humans and Neandertals on the one hand and that of African apes on the other. In this respect, the *H. naledi* capitates are more similar to those of *H. floresiensis* and several australopiths.

## Vertebrae

Seven vertebrae have been recovered in the 102a assemblage, all from the lower thoracic and lumbar region of the spine. These vertebrae are roughly equivalent in preservation. The thin cortical bone of the vertebral bodies is eroded in large patches on these elements with exposure of underlying trabeculae. They have minimal surface staining or patination, and where the vertebral arches are present, the cortical surface is well-preserved.

**U.W. 102a-036** is a largely complete antepenultimate thoracic vertebra, inferred as T10, with limited erosion to the anterior surface of the vertebral body and some damage distally on the transverse processes, particularly on the right side, and to the posterior end of the spinous process (*Figure 20*). The vertebral body is ovoid to kidney-shaped and the ring apophyses are relatively thick, covering approximately three-quarters of the vertebral body surface. The spinal canal is ovoid in shape and about one-third the size of the vertebral body. Facets for the tenth rib are posteriorly positioned, almost entirely on the pedicles. The pedicles themselves are transversely thick, as are the transverse processes, which are strongly posteriorly oriented. Together, the transverse processes and pedicles form nearly continuous, robust lateral structures for anchoring epaxial muscles and for transmitting forces to and from the vertebral body, respectively.

**U.W. 102a-151** is a nearly complete penultimate thoracic vertebra, inferred as T11, with some erosion and loss to the left side of the vertebral body and missing the left-side transverse process (*Figure 21*). Portions of the superior vertebral body surface are eroded away, revealing trabeculae.

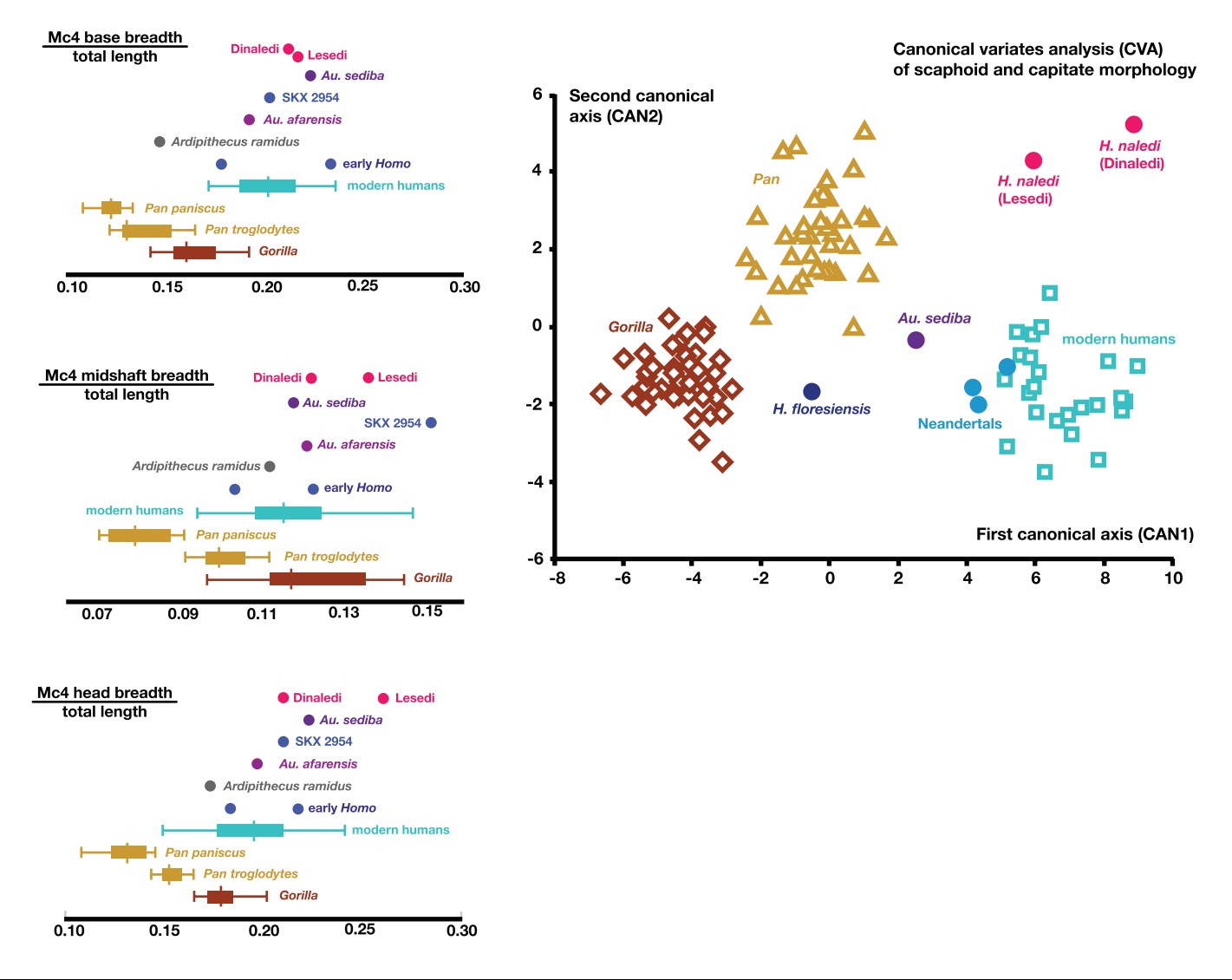

**Figure 19.** Quantitative comparisons of hand and wrist material from the Lesedi Chamber. Left: ratios of fourth metacarpal dimensions in *H. naledi* compared to those in other hominin and great ape samples. Right: canonical variates analysis of capitate and scaphoid morphology in humans, chimpanzees, gorillas, and fossil hominins. *H. naledi* from the Dinaledi Chamber occupies a unique position in scaphoid and capitate joint configurations, which is closely matched by the capitate and scaphoid from the Lesedi Chamber. In this analysis, no *a priori* groups are assumed; we also examined the scenario in which *Homo naledi* and other fossil specimens are included as *a priori* groups and the results are essentially identical.

The superior articular facets are planiform and posteriorly oriented. The inferior articular facets are asymmetrical – the right side is curved and anterolaterally oriented, whereas the left side is planiform and oriented anteriorly on the coronal plane, as in the transitional vertebra. Costal facets are large, extending from the posterior aspect of the body inferiorly and posteriorly onto the pedicle. The vertebral body is kidney- to heart-shaped and the spinal canal is ovoid, with a slightly triangular shape. The spinous process is relatively long and relatively horizontal in its orientation, with its major axis deflecting inferiorly at an angle of approximately 20° from the surface of the superior vertebral body.

**U.W. 102a-154a** is a nearly complete last thoracic vertebra, inferred as T12. The right inferior articular facet, distal spinous process, and anterior aspect of the inferior vertebral body are broken away. The anterior portion of the body is eroded on the right side, as are the lateral aspects of the superior vertebral body. The superior articular facets are asymmetrical, matching the inferior articular

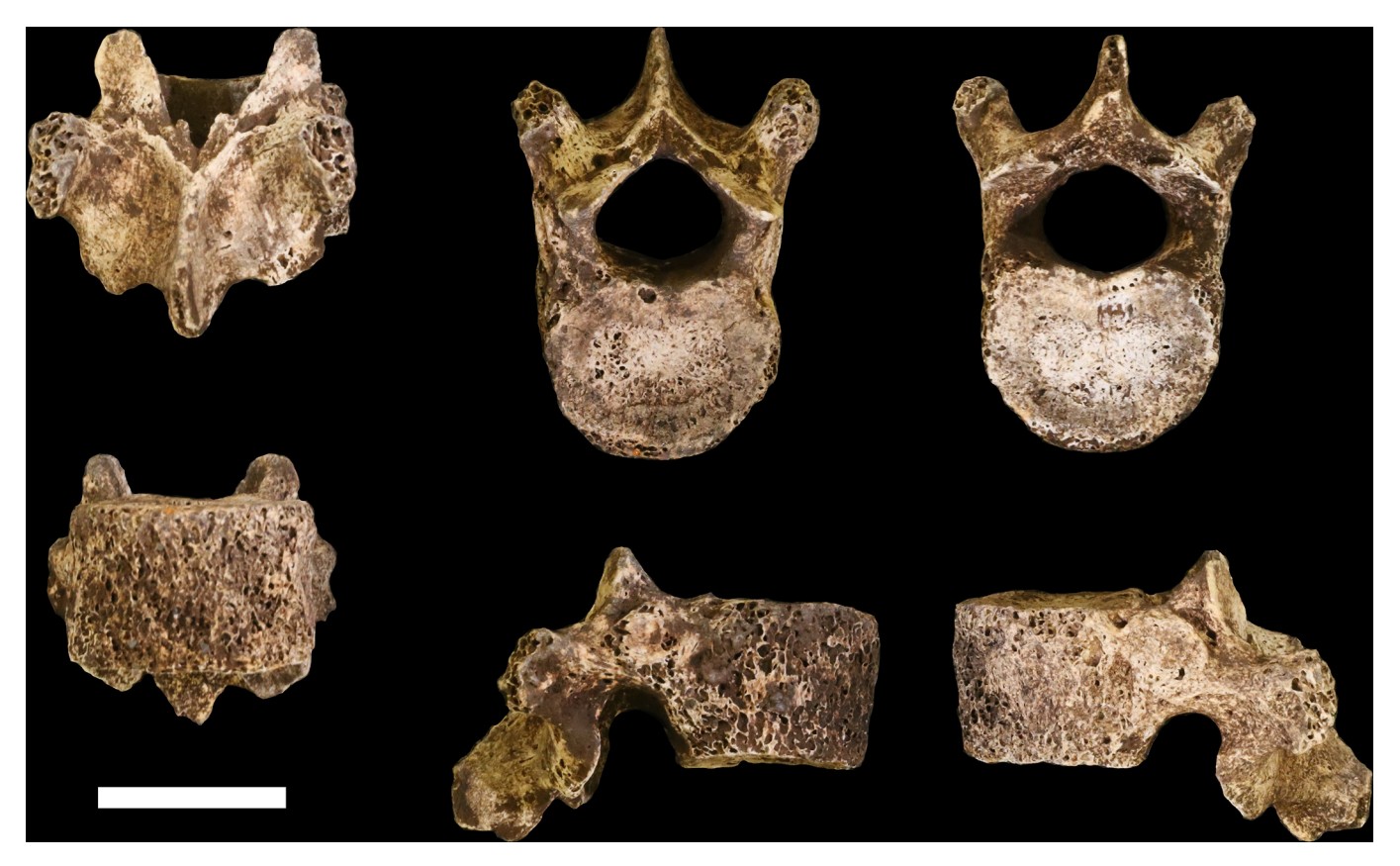

**Figure 20.** U.W. 102a-036 vertebra, T10. Clockwise from top left: posterior, superior, inferior, left, right, and anterior views. Scale bar = 2 cm.

facets of the superjacent vertebra (U.W. 102a-151): the left superior articular facet is planiform and posteriorly oriented on the coronal plane, whereas the right superior articular facet is curved and posterolaterally oriented. The right superior articular facet is comparatively diminutive in size, particularly in transverse dimension. The vertebral body is kidney- to heart-shaped and transversely wide. The costal facets are positioned at the body-pedicle border but are eroded on both sides; thus, their morphology cannot be fully appreciated. The pedicles themselves are anteroposteriorly short and contribute to a wide, ovoid spinal canal.

**U.W. 102a-154b, U.W. 102a-322**, and **U.W. 102a-306** are vertebral bodies associated with little or no vertebral arch structures. **U.W. 102a-139** is a lumbar vertebra preserving most aspects of the vertebral body and neural arch, but it is broken into five pieces that refit reasonably well, although the spinous process is missing. None of the bodies or preserved aspects of pedicles bear costal facets. U.W. 102a-154b nicely articulates with U.W. 102a-154a superiorly and U.W. 102a-322 inferiorly, and U.W. 102a-306 and U.W. 102a-139 articulate with each other; however, U.W. 102a-322 and U.W. 102a-306 do not articulate. The lumbar transverse processes of U.W. 102a-139 are anteroposteriorly wide, emerging anteriorly from the posterior aspect of the vertebral body along the pedicles and posteriorly to the bases of the superior articular processes. Its body is clearly posteriorly wedged in lateral view. Together, these features indicate that U.W. 102a-139 is the last lumbar vertebra. Therefore, the likely seriation is as follows: U.W. 102a-154b is L1, U.W. 102a-322 is L2, U.W. 102a-306 is L4, U.W. 102a-139 is L5, and L3 is missing.

## Comparative vertebral anatomy

The U.W. 102a-036 T10 and U.W. 102a-151 T11 vertebrae are directly comparable to the near-complete U.W. 101–855 T10 and U.W. 101–1733 T11 vertebrae from the Dinaledi Chamber. The Dinaledi

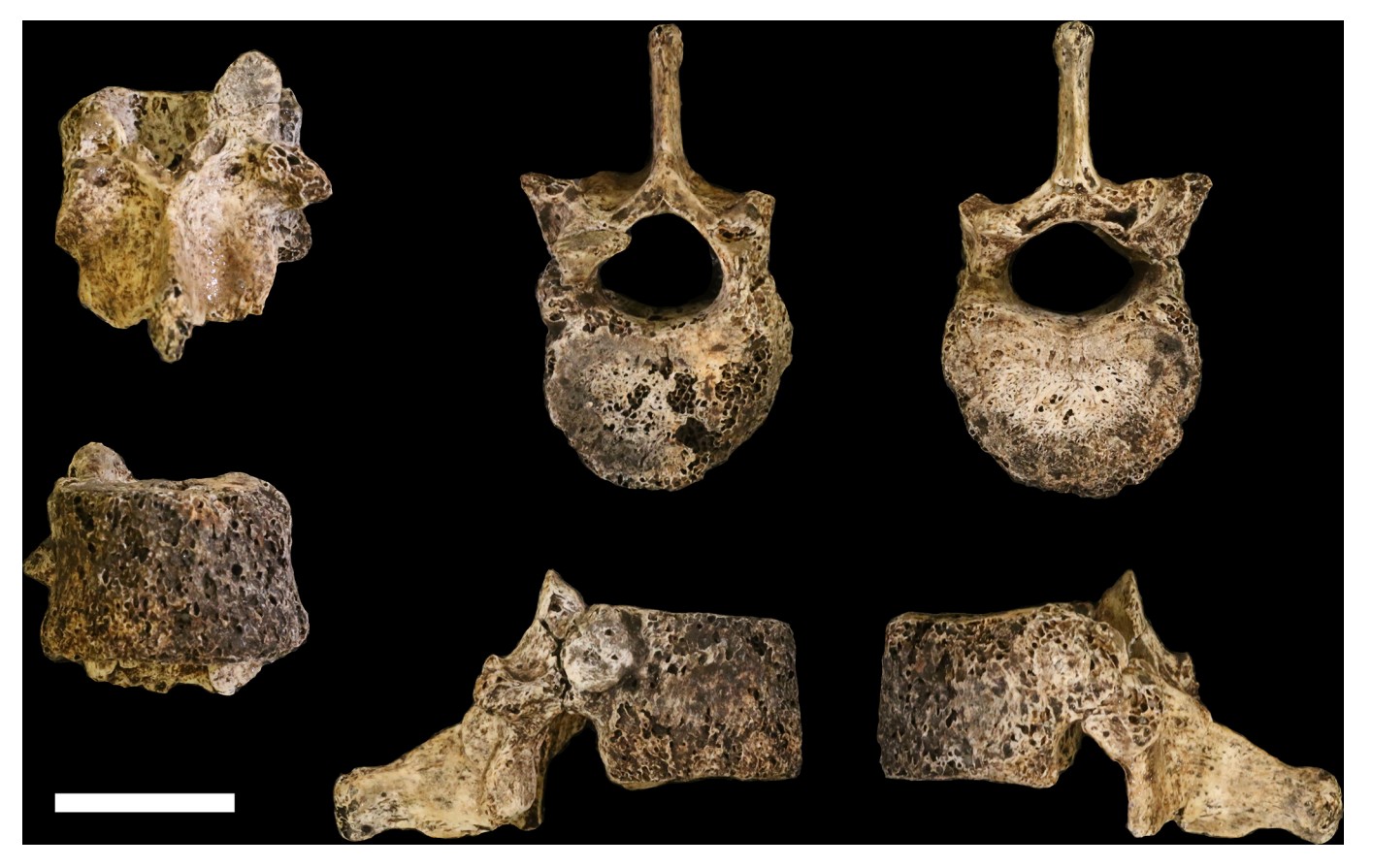

**Figure 21.** U.W. 102a-151 vertebra, T11.  Clockwise from top left: posterior, superior, inferior, left, right, and anterior views. Scale bar = 2 cm.

and Lesedi Chamber pairs are comparable in size, but the Lesedi vertebrae clearly belong to a larger, more robust (presumed male) individual. Although the Dinaledi transverse processes are broken at their bases, the preserved aspects are strongly posteriorly oriented, albeit to a lesser degree than those from the Lesedi Chamber (*Figure 22*). The lower thoracic transverse processes of *Au. afarensis*, *Au. africanus*, and *Au. sediba* possess more laterally oriented transverse processes. Only SKX-41692, a presumed *P. robustus* T10, possesses similarly posteriorly oriented transverse processes among australopiths. However, its relatively large vertebral body, small spinal canal, and overall shape contrast with the lower thoracic transverse processes of *H. naledi* (*Williams et al., 2017*). The combination of a relatively large vertebral body and spinal canal is present in both the Dinaledi and the Lesedi Chamber T10 vertebrae, but not in*Australopithecus* and *Paranthropus* specimens. The Dinaledi T11 bears planiform articular facets superiorly and inferiorly and is therefore not the transitional vertebra. In the Lesedi Chamber vertebral column, the change in articular facet orientation occurs asymmetrically across the T11 and T12 vertebrae, as occurs in <4% of modern humans (*Williams et al., 2017*). In all known *Australopithecus* and *H. erectus* specimens, the transitional vertebra is T11 (*Haeusler et al., 2002*, *2011*; *Williams et al., 2013*; *Meyer et al., 2015*).

## Costae

**U.W. 102a-250** is a nearly complete right first rib, with erosion and breakage to the head, tubercle, lateral border and distal end. The neck is flattened in its superior-inferior dimension and descends in the vertebro-inferior direction. The tubercle and the posterior angle coincide. The facet of the articular tubercle was damaged post-mortem.

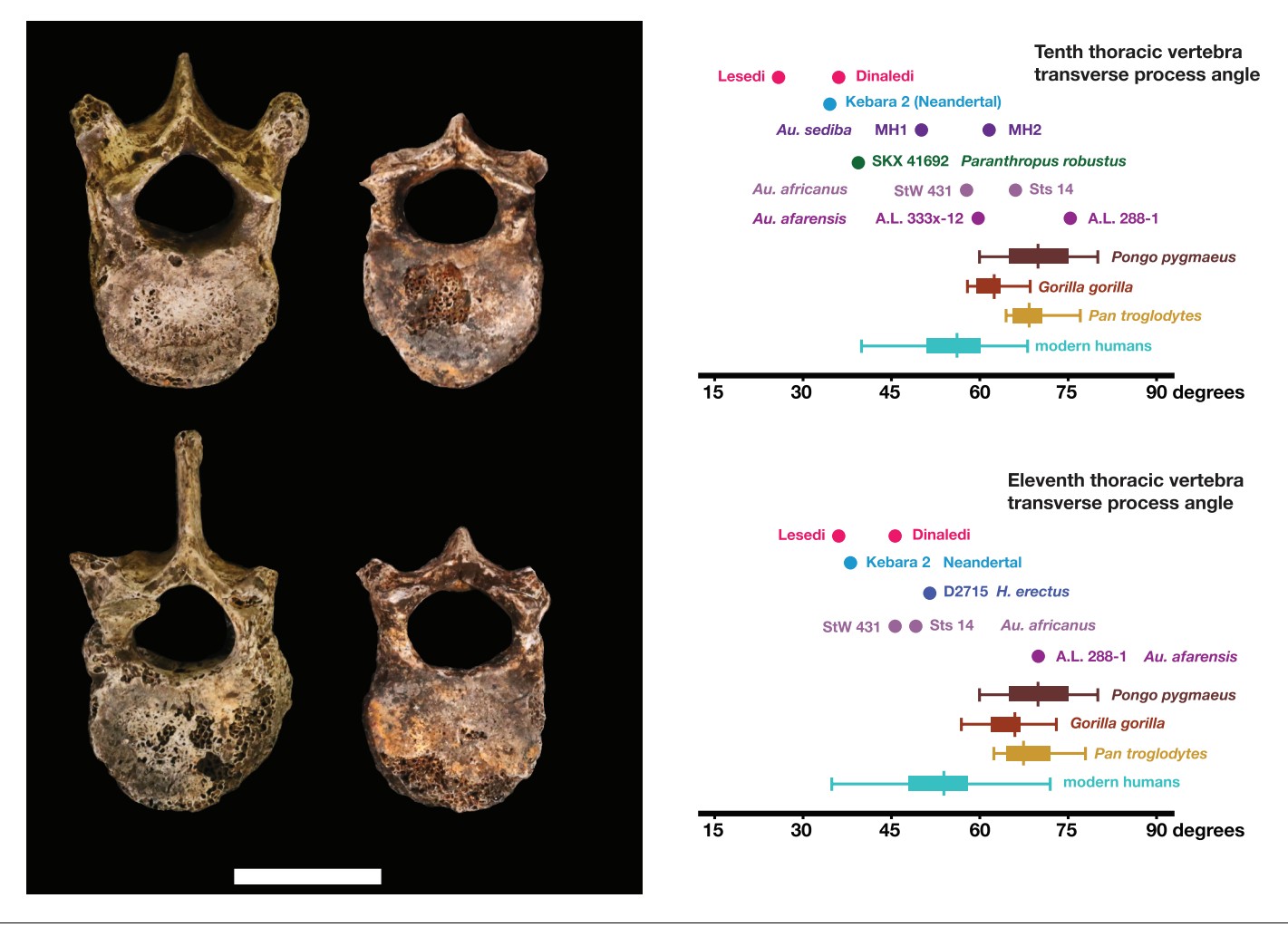

**Figure 22.** Vertebral transverse process orientation. *H. naledi* is distinctive when compared to many other hominin species in having T10 and T11 vertebral transverse processes oriented with a relatively low angle. Left: U.W. 102a-036 compared to U.W. 101–855 from the Dinaledi Chamber (top), and U.W. 102a-151 compared to U.W. 101–1733 (bottom). All of these vertebrae have transverse processes oriented more posteriorly than those of most other hominins, U.W. 102a-036 is the most extreme. Right: charts showing the comparative orientation of transverse processes in humans, living great apes, and fossil hominins. For the T10 (top), the U.W. 102a-036 value (labeled 'Lesedi') is lower than that for any other hominins, while the Dinaledi T10 is similar to the Neandertal value and extremely low compared to that for modern humans. The T11 (bottom) shows a similar but less extreme pattern.

Two partial first ribs (U.W. 101–083 and U.W. 101–621) of *H. naledi* are preserved in the Dinaledi Chamber hominin sample, but neither rib has its head nor enough of the shaft preserved to allow accurate estimation of curvature (*Williams et al., 2017*). The angulation and shape of these fragments appears comparable to those of MH2 *Au. sediba* and A.L. 288–1 *Au. afarensis*. U.W. 102a-250 is more complete than the Dinaledi first rib fragments, and is similar in morphology in the overlapping regions. This rib is slightly more curved than the Sterkfontein first rib, StW 670 (*Tawan et al., 2016*). The anatomy of U.W. 102a-250 is entirely compatible with attribution to *H. naledi*, although the bone is also similar in morphology and size to known australopith first ribs.

Thirteen additional specimens from 102a are partial ribs or rib fragments, none are identifiable to element and none present anatomical information that is useful for testing the taxonomic affiliation of the sample.

## Ossa coxae

**U.W. 102a-138** (*Figure 23*) is a fragmentary right ilium of an immature individual (as evident by the presence of triradiate cartilage, by an unfused apophysis at the anterior inferior iliac spine, and by very small overall size). The fragment is very light, with thin cortical bone, and is eroded around margins of the acetabular portion. The iliac blade is mostly missing, but the auricular surface, greater sciatic notch, acetabulosacral buttress, and anterior margin of the iliac blade are present. Despite the thin and fragile nature of this element, the surface is well-preserved.

The adult pelvic material of *H. naledi* from the Dinaledi Chamber is notable in combining an *Au. afarensis*-like degree of iliac flare, a weak and anteriorly placed iliac pillar, and a narrow tuberoacetabular sulcus on the ischium (*Berger et al., 2015*; VanSickle et al., personal communication). U.W. 102a-138 represents the most complete immature ilium fragment of *H. naledi* found to date, and its morphology is comparable to that of the juvenile U.W. 101–486 ilium fragment, and thus consistent with the morphology seen in *H. naledi*. It lacks the diagnostic characters that could differentiate it clearly from ilium fragments from other hominin species, as the iliac blade and iliac pillar are both poorly preserved. The lack of an accompanying ischial fragment precludes an evaluation of tuberoacetabular sulcus morphology in the 102a material.

## Femora

**U.W. 102a-001** is a proximal right femur, in which much of the head and neck, and the proximal subtrochanteric shaft are preserved (*Figure 24*). The head is badly eroded, especially anteriorly, and

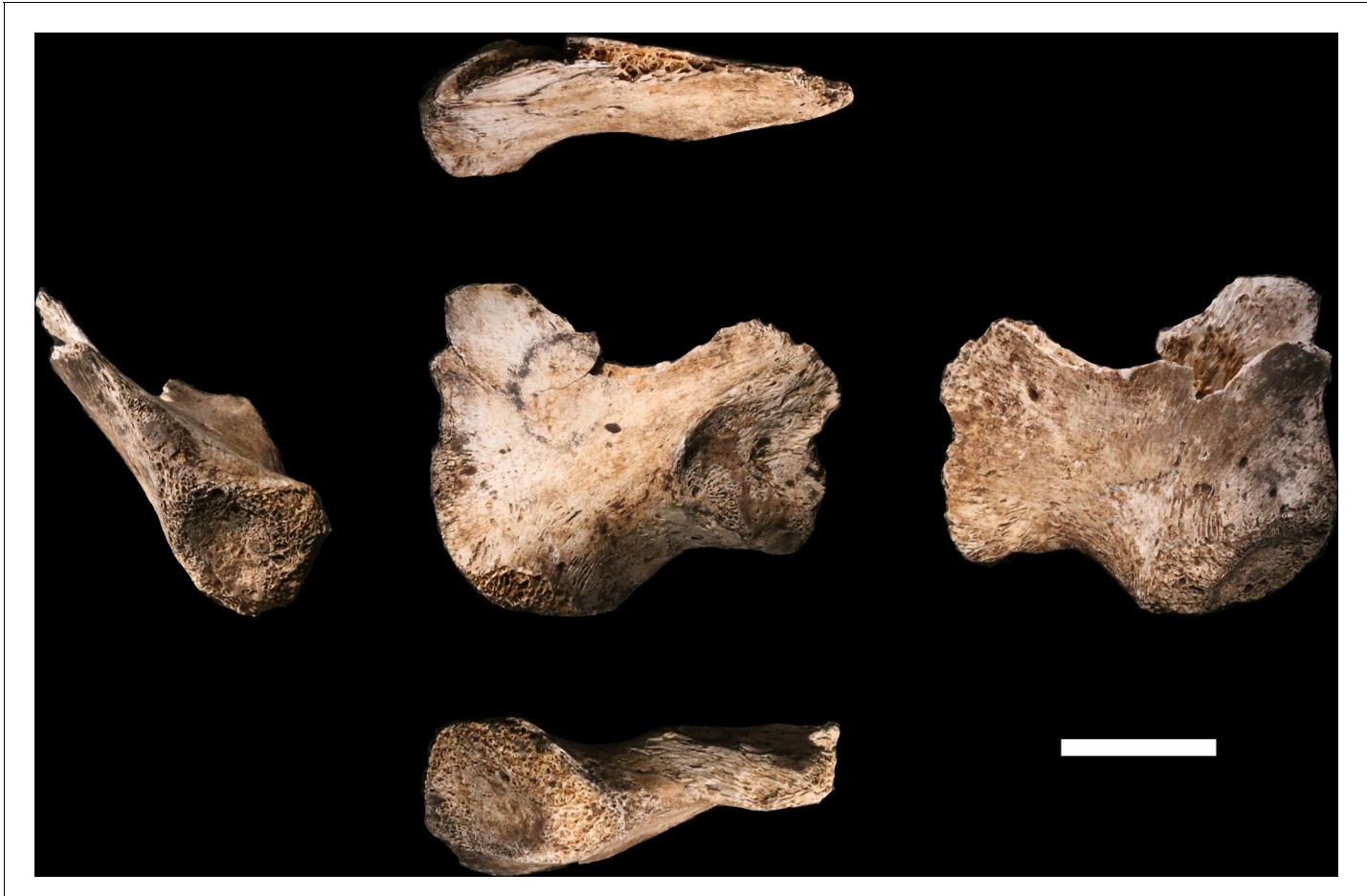

**Figure 23.** U.W. 102a-138 immature right os coxa fragment. The medial view is at the center. Clockwise from top: superior, lateral, inferior and anterior views. The unfused triradiate suture is notable.

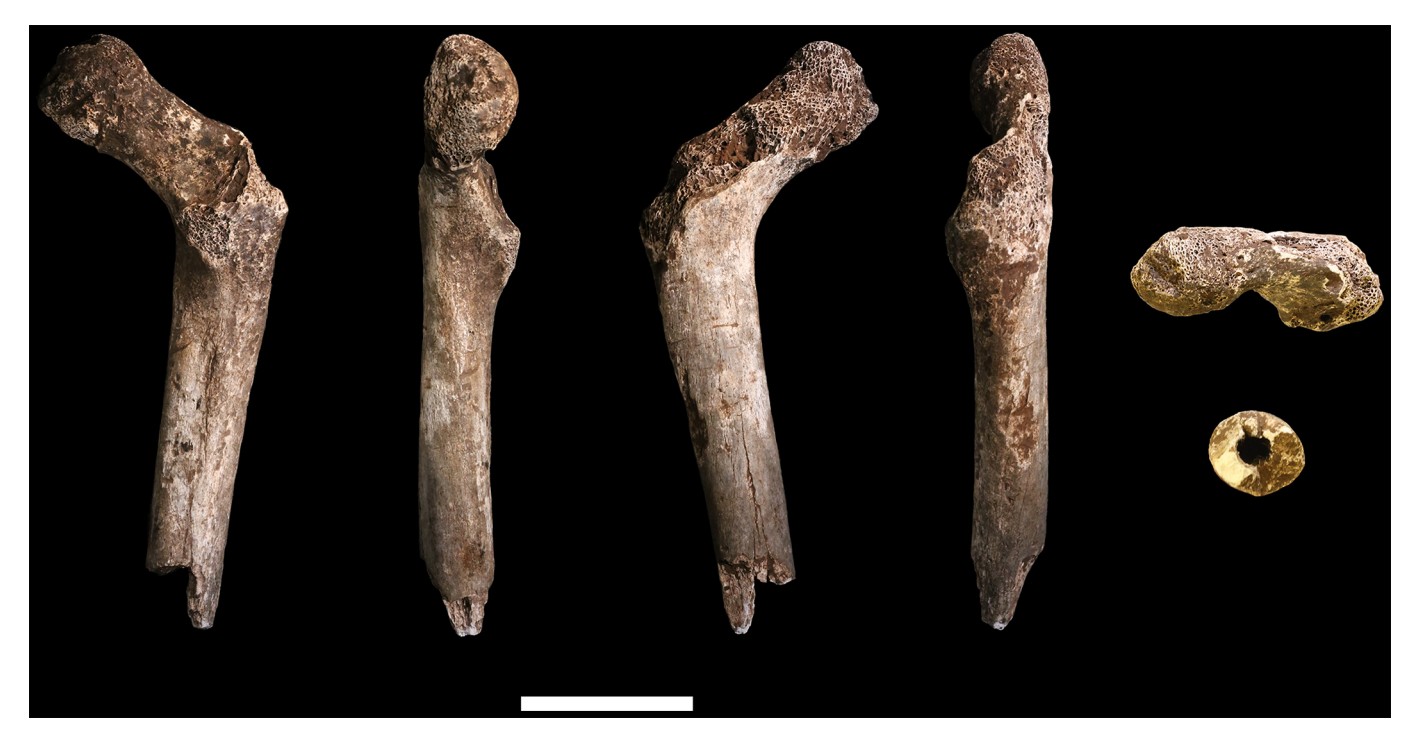

**Figure 24.** U.W. 102a-001 proximal femoral fragment. From left: posterior, medial, anterior and lateral views. Right from top: proximal and distal views. Scale bar = 5 cm.

only a few small patches of subchondral articular bone are preserved on the posterior aspect. The posterior side of the neck is fairly well preserved from the head all the way to the lesser trochanter, which is planed off, with only the base remaining. The anterior side of the neck is missing. Trabecular bone is exposed from the anterior head all the way to the lateral surface at the base of the greater trochanter. The greater trochanter is missing entirely, save for a small bit of its distal lateral surface. The surface overall is marred by areas of post-depositional damage, including a number of transverse scratches on the shaft.

**U.W. 102a-003** is a left femoral shaft fragment, from the lesser trochanter proximally to about midshaft (*Figure 25*). Only the base of the lesser trochanter remains. The head and neck are not present.

**U.W. 102a-004** is a fragment of left distal femur, preserved from roughly midshaft to the distal subchondral bone surface of the intercondylar notch (*Figure 26*). Both condyles are missing. The shaft has surficial markings similar to those present on U.W. 102a-001. This fragment is morphologically compatible with U.W. 102a-003 in shaft diameter and cross-section, and the two specimens exhibit no morphological overlap, suggesting that they may represent the same femur.

U.W. 102a-003 and U.W. 102a-004 may conjoin with each other. Both fragments are morphologically compatible in shaft diameter and cross-section, and at the broken distal end of U.W. 102a-003 and at the proximal end of U.W. 102a-004, a small part of the circumference of the shaft (approximately 10 mm in total) on the posterolateral side appears to provide a refit. However, the edges of this apparent break are abraded, reducing the certainty of the association. Joining the bones at this point, U.W. 102a-003 and U.W. 102a-004 preserve 321 mm of a femoral shaft. Using the similarly sized U.W. 101–002 to represent the missing proximal end and KNM-ER 1481 to represent the distal end (*Figure 27*), we preliminarily estimate the femoral length of this individual to be ~375 mm.

U.W. 102a-001 (right proximal) and U.W. 102a-003 (left proximal) are similar in size. They preserve an overlapping area of anatomy from just above the lesser trochanter down to around the midshaft area. However, despite their similarity in size, the two contrast in several

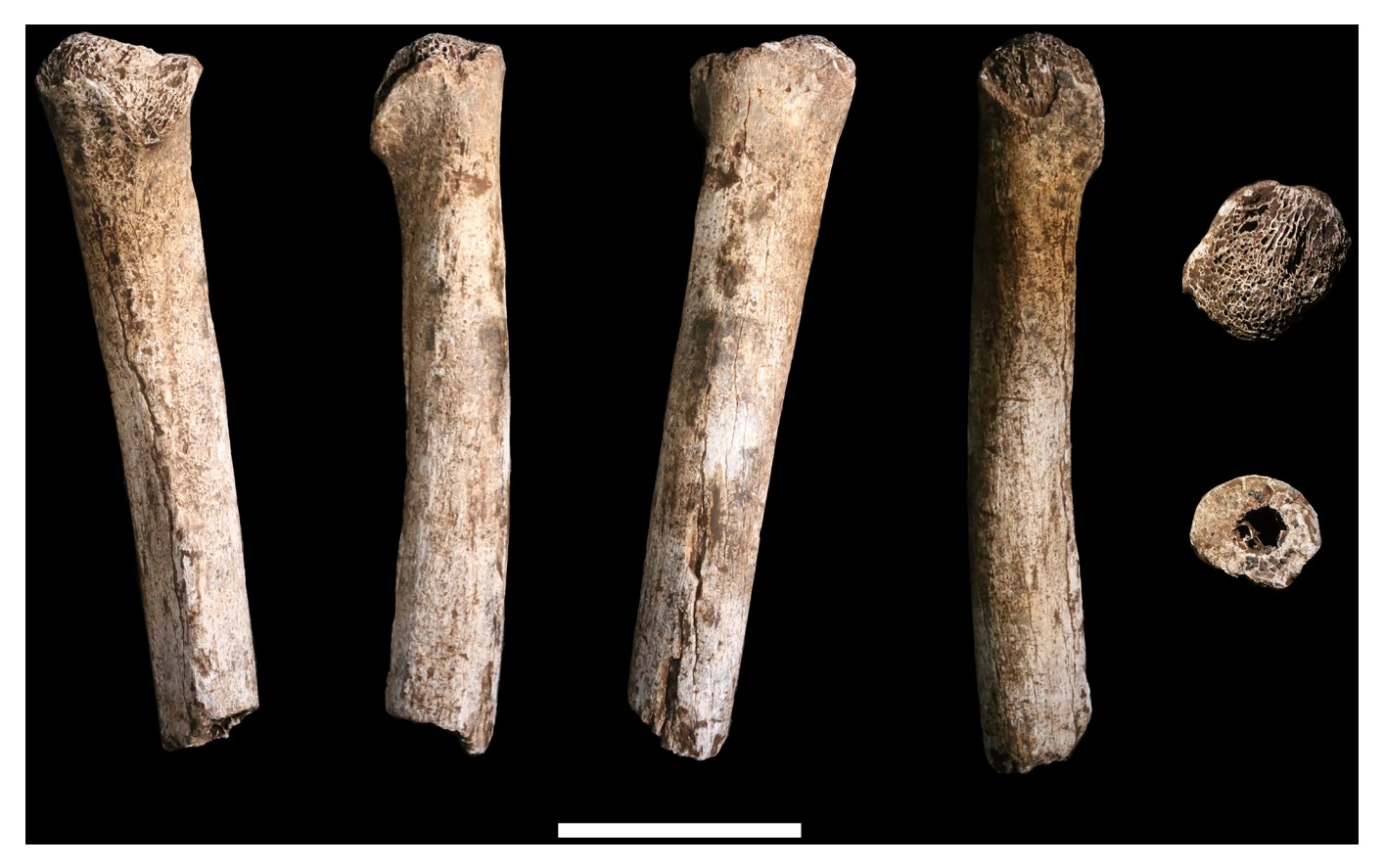

**Figure 25.** U.W. 102a-003 left proximal femur fragment. From left: posterior, medial, anterior and lateral views. Right from top: proximal and distal views. Scale bar = 5 cm.

anatomical details. U.W. 102a-001 is more platymeric in the subtrochanteric area, with a greater mediolateral (ML) breadth than U.W. 102a-003. The lesser trochanter is abraded in both specimens, but the morphology of the inferior and medial aspects of it appear different in the two bones. U.W. 102a-001 has a shallow sloping border to the lesser trochanter medially, and the inferior aspect tails off into a broad, less marked line leading to a very slight linea aspera by midshaft. By contrast, U.W. 102a-003 has a steep medial aspect to the lesser trochanter, and it tails into a sharply defined crest that broadens around 15 mm down the shaft into a rugose, double crest, which narrows by midshaft into a strong linea aspera. The insertion for *m. gluteus maximus* is prominent and rugose in both femora but in U.W. 102a-001, the rugosity extends further down the shaft. Overall, the asymmetry of these two bones would be very unusual in the left and right femora of a single individual. We accept this provisionally as evidence for a second adult individual in the 102a assemblage.

## Comparative femoral anatomy

The femoral morphology of *H. naledi* is a mosaic of features seen in *Australopithecus* species such as *Au. afarensis*, *Au. africanus* and *Au. sediba*, and features otherwise known in *Homo*, including Plio-Pleistocene *H. erectus* and fossils attributed to *Homo* sp. indet., such as KNM-ER 1472 and KNM-ER 1481 (*Figure 28*; *Marchi et al., 2017*). The femoral remains from the 102a locality share this mosaic of features, including: a marked linea aspera, with a weak pilaster in U.W. 102a-003; a strong muscle insertion for *m. gluteus maximus*; a platymeric shaft just inferior to the lesser trochanter; a long and anteroposteriorly flattened femoral neck; and an AP expanded femoral midshaft. The Dinaledi remains of *H. naledi* have markedly anteverted femoral necks. Damage to the femoral neck of the U. W. 102a-001 specimen limits the accuracy of an estimate of anteversion, but at 115 degrees, this

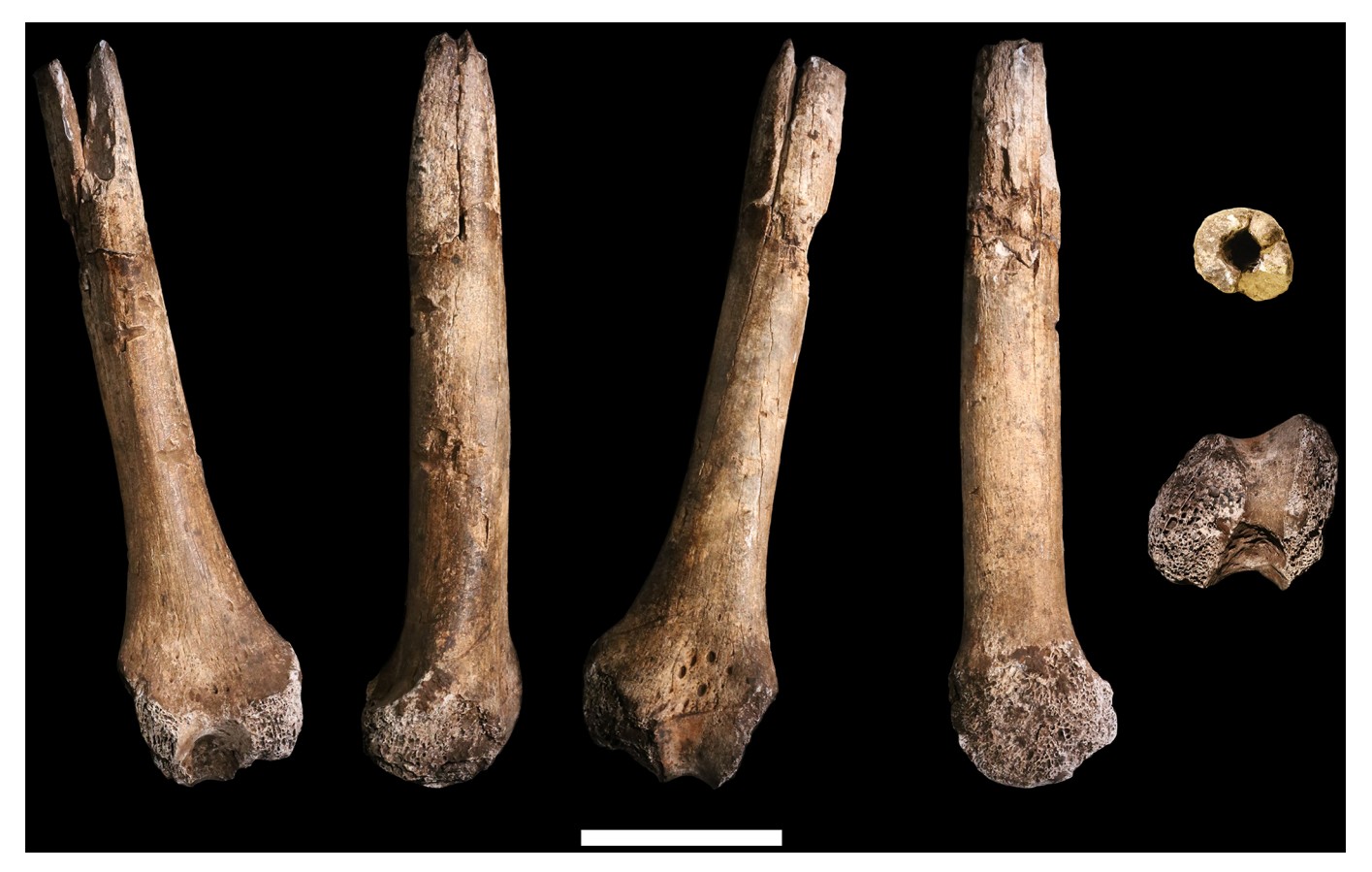

**Figure 26.** U.W. 102a-004 left distal femur fragment. From left: posterior, medial, anterior and lateral views. Right from top: proximal and distal views. Scale bar = 5 cm.

estimate is within the range for the Dinaledi specimens. The most distinctive feature of the *H. naledi* proximal femur is the presence of two mediolaterally oriented pillars on the superior femur neck, separated by a medially positioned shallow and vascularized groove where *m. obturator internus* and *gemelli* insert (*Marchi et al., 2017*). This configuration is not seen in other hominin species. The U.W. 102a-001 femoral neck is damaged superiorly, precluding a clear assessment of whether two distinct pillars were present. What is preserved posteriorly demonstrates the presence of a medially positioned vascularized groove, but not of the inferior pillar present in the Dinaledi femora. With this exception, the U.W. 102a-001, U.W. 102a-003 and U.W. 102a-004 femoral fragments are entirely consistent with the morphology known for *H. naledi*, and the combination of features in the U.W. 102a-001 proximal femur is not consistent with known fossil examples attributed to other species of *Homo* or *Australopithecus*.

## Individuals represented in 102a

The hominin material from 102a appears to represent a minimum of two adult individuals and one immature individual. The inference of two adults is based upon the morphological incongruence of the left (U.W. 102a-003) and right (U.W. 102a-001) femoral elements (discussed above). Still, no adult element is clearly duplicated in the collection. The U.W. 102a-138 ilium, along with an immature sacrum fragment and two immature long bone fragments not described here, demonstrates the presence of at least one immature individual.

The lack of duplication of elements suggests that much (but not all) of the adult material may represent a single individual skeleton, which parsimoniously would also include the LES1 cranium. We

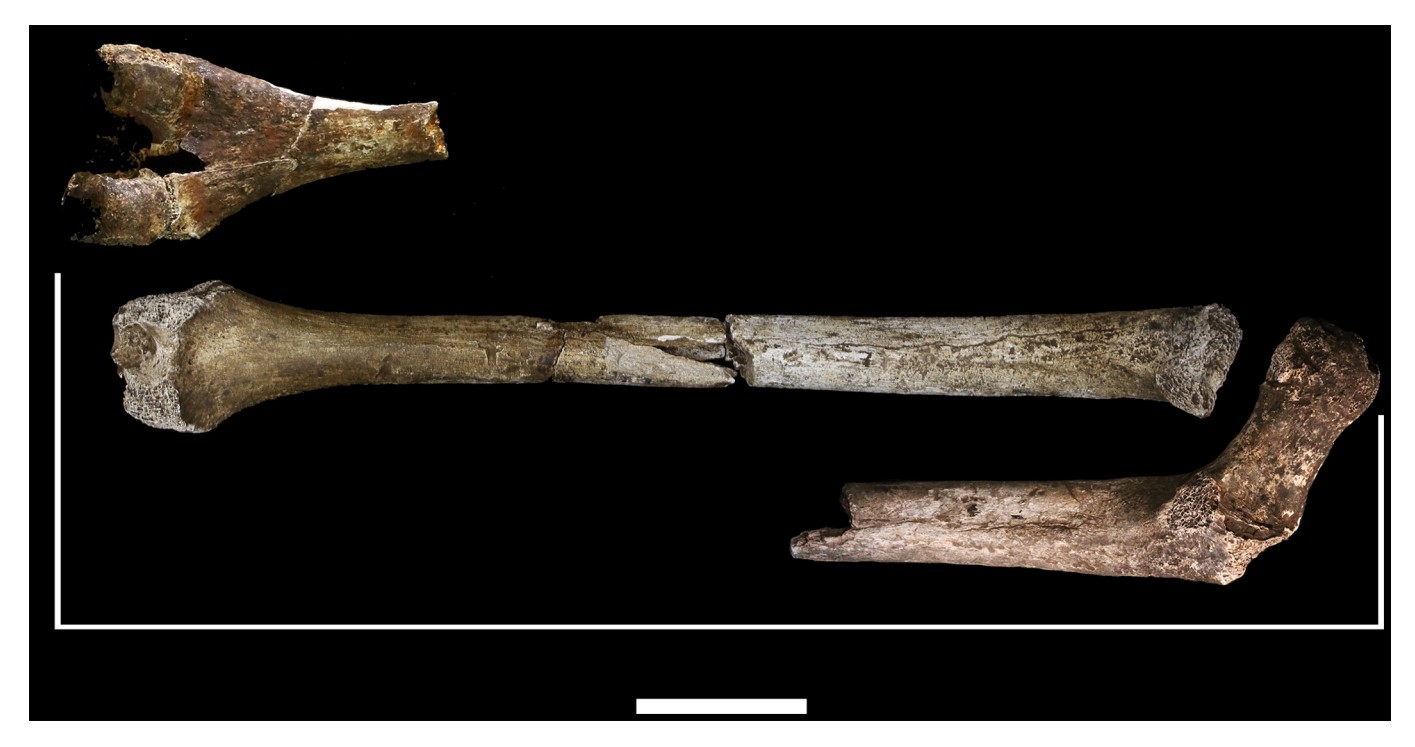

**Figure 27.** Length estimation of femur based on U.W. 102a-003 and U.W. 102a-004. Two specimens were used to estimate the missing proximal and distal ends of the femur. Top: U.W. 101–215 is a distal femur fragment that presents a similar morphology to the U.W. 102a-004 distal femur, while preserving the distal articular surface. Middle: U.W. 102a-004 and U.W. 102a-003 conjoined, in posterior view. Bottom: U.W. 102a-001 is comparable in size with U.W. 102a-003, and while the morphology of the muscle markings is different, the alignment of the lesser trochanters gives a good basis for estimating the proximal extent of the bone. The length estimate is 375 mm.

accept this hypothesis provisionally. All elements in the current 102a collection were recovered from within an excavation area less than 50 cm x 70 cm, and 40 cm deep. Two vertebrae (U.W. 102a-154a and U.W. 102a-154b) were in articulation in situ. The articular morphology and sizes of seven vertebrae suggest strongly that they represent a single individual; the remainder of these elements were recovered in close physical proximity but not in articulation. All fragments attributed to the LES1 cranium were likewise recovered from a small area. The hand and wrist material is consistent with a single right hand on the basis of articular morphology. None of the other elements lend themselves to an evaluation of articular compatibility, but they are consistent in size. We consider it unlikely that the number of elements in the current collection would be recovered from a commingled assemblage consisting of substantial parts of multiple skeletons without also introducing duplicate elements.

This raises the problem of what seems to be a mismatch of the U.W. 102a-001 and U.W. 102a-003 femora. These two are similar in size, and the difference in their shaft dimensions is not greater than that found in 95% of a large sample of paired left and right human femora. However, they are different in muscle attachments and diaphyseal morphology, to the extent that would represent unusual asymmetry in a single individual. The data do not allow us to discard the hypothesis that one of the femora represents a second adult individual in 102a, albeit an individual of similar body size. The descriptive and measurement data do not indicate which (if either) of these femora may belong to the individual represented by LES1, nor which (if any) of the other postcranial elements are associated with either femur. On the basis of the non-duplication of elements, it seems likely that if there is a second individual, this second individual is represented by only a small number of elements, possibly just the femur. The two humerus specimens, U.W. 102a-002 and U.W. 102a-257, differ slightly in shaft diameter where it can be compared, but no other morphological differences are apparent on the preserved fragments, and a slight degree of upper limb asymmetry is not

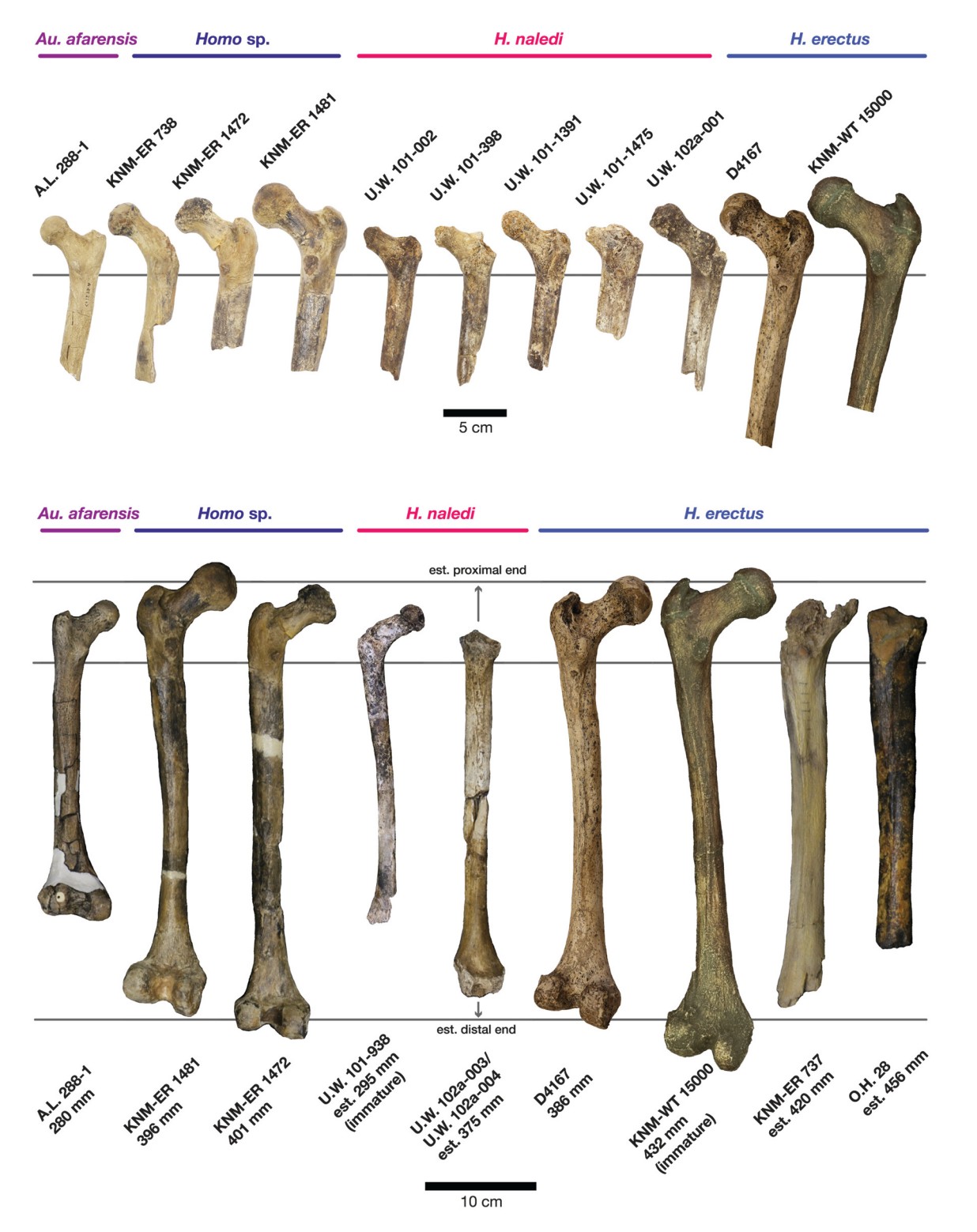

**Figure 28.** Comparison of *H. naledi* femora to those attributed to early *Homo*. Top: roximal femora attributed to *Au. afarensis*, *Homo* sp., *H. naledi*, and *H. erectus*, all shown in posterior view. The femora have been aligned by matching the inferior point on the lesser trochanter to the horizontal line on the figure, and all are shown at approximately the same shaft angle. Many of the *H. naledi* femora have notably thin shafts, although the largest shown here, U.W. 101–1475, is greater in shaft thickness than the complete KNM-ER 1480, KNM-ER 1472, or D4167 femora. The specimens here attributed to 'Homo sp.' were surface finds without associated cranial material. Their anatomy has been considered consistent with *Homo*, although
*Figure 28 continued on next page*

*Figure 28 continued*

some have suggested that KNM-ER 738 may instead be *Paranthropus*. U.W. 102a-001 is a right femoral fragment, and left femora here have been mirrored for comparison, including KNM-ER 738, KNM-ER 1481, U.W. 101–398, U.W. 101–1475, KNM-WT 15000, and A.L. 288–1. Bottom: *H. naledi* femur length compared to those of other fossil femora. U.W. 102a-003 is shown conjoined with U.W. 102a-004. The top and bottom horizontal lines correspond to the proximal and distal limits of the maximum length estimate for this femur as illustrated in *Figure 24*. Femur maximum lengths and length estimates are as reported by *McHenry (1991)*, D4167 length is from *Lordkipanidze et al. (2007)*. As above, all femora are aligned by the lesser trochanter, which is preserved in all of these shaft fragments. The *H. naledi* U.W. 102a-003 femur has a shaft diameter comparable to that of A.L. 288–1, or even slightly shorter, but at an estimated 375 mm, it is nearly the same length as the D4167 femur of *H. erectus* at 387 mm. The U.W. 101–484 tibia specimen from the Dinaledi Chamber is also long and relatively narrow, and has an estimated length greater than that of the D3901 tibia from Dmanisi (*Marchi et al., 2017*). Several very thick, large femora have been attributed to *H. erectus* from the Early Pleistocene of Africa; these are very different from U.W. 102a-003 in their size, robustness, and more platymeric proximal shafts. U.W. 101–938 is an immature femur with no fusion of the head or distal epiphysis; it represents a younger developmental age than KNM-WT 15000, the immature *H. erectus* skeleton. Even at its young age, this *H. naledi* specimen is nonetheless longer than the A.L. 288–1 femur of *Au. afarensis*. A.L. 288–1 is shown here as reconstructed by P. Schmid. For comparison to the right U.W. 102a-003 femur, left femora are shown mirrored here, these include KNM-ER 1472, U.W. 101–938, and D4167. In this figure, KNM-ER 739, KNM-ER 1472, KNM-ER 1481, KNM-WT 15000, O.H. 28, and A.L. 288–1 are represented by casts.

unusual in humans or in fossil *Homo*. We conclude that most of the 102a adult material probably represents a single skeleton, but we cannot assume that the ratios of either femur with other elements in the sample would reflect the proportions of a single individual. To be conservative, any consideration of proportions should allow for the possibility that multiple individuals are present.

## Hominin material from 102b

The material from area 102b includes 12 specimens identified as hominin. Most of this collection consists of small fragments of cranium, many of which are identifiable to element, but which preserve no diagnostic morphology that would assist in taxonomic assessment. One partial mandible and five possibly associated teeth preserve morphological characters that are useful for taxonomic attribution.

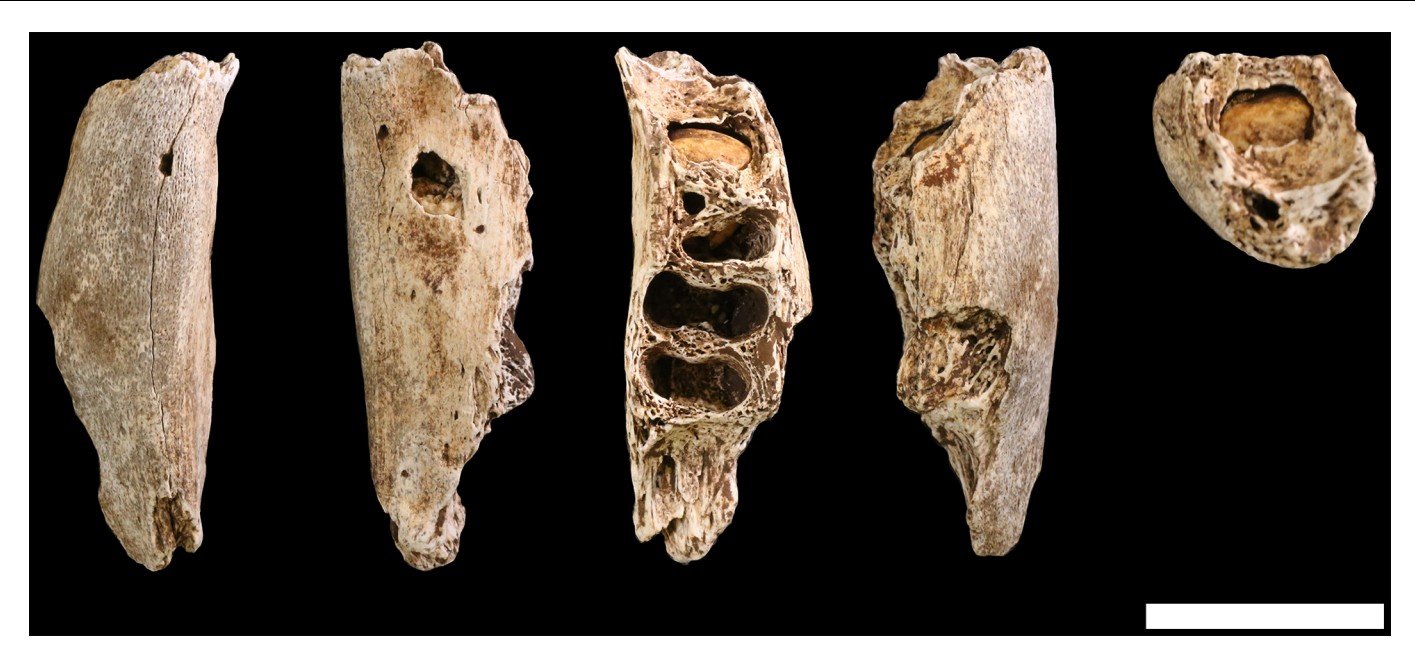

**Figure 29.** U.W. 102b-438 immature mandibular fragment. From left: basal, lingual, occlusal, buccal and anterior views. The RP$_4$ is within its crypt. Scale bar = 2 cm.

**U.W. 102b-438** is a fragment of right mandibular corpus from an immature individual (*Figure 29*). The alveoli for the $RM_1$ are present; the alveoli for the $rdm_2$ are also present and reveal the crypt with $RP_4$ crown intact within. The fragment is broken anteriorly at the crypt for $RP_3$ and posteriorly at the crypt for $RM_2$, neither permanent tooth is present. The preserved corpus height at the mid-point of $rdm_2$ is 14.9 mm; this is perhaps 1–2 mm less than the true value because of the erosion of the alveolar bone surface. Corpus breadth at the anterior edge of $M_1$ is 15.2 mm; total length of the fragment as preserved is 43.5 mm.

Several teeth excavated from 102b are compatible with the same approximate developmental stage as the U.W. 102b-438 mandibular fragment, and were recovered in close proximity to each other and to the mandible; no elements are duplicated. We hypothesize that these fragments represent the same individual, at least until the recovery of additional material makes us reassess this possible association. **U.W. 102b-437** is the complete crown and two nearly complete roots of a $ldm_2$ with moderate occlusal wear, including dentine exposure at the centra of cusps, and roots that appear to have begun to resorb at the tips. **U.W. 102b-503** is a $RP_4$ crown, nearly complete but not erupted. **U.W. 102b-511** is a $LC_1$ crown, nearly complete with no occlusal wear. **U.W. 102b-515** is a $LI_2$ that is nearly crown complete and unerupted. **U.W. 102b-178** is a broken but apparently unworn probable $RI_2$ crown that was recovered separately from these other teeth, which may also represent this individual.

The teeth all fall within the size range of equivalent elements from the Dinaledi Chamber sample. The U.W. 102b-438 $RP_4$ crown is partially obscured within its crypt and not complete, to the minimal extent it is visible at this time, it is consistent with *H. naledi* mandibular $P_4$ morphology. The U.W. 102b-503 maxillary $P_4$ is likewise consistent with comparable Dinaledi examples, although this tooth differs little among several species of *Homo*, including modern humans. Like the incisors of LES1, U.W. 102b-515 lacks prominent cervico-incisal crown curvature, has no prominent crests on its lingual surface, and is not shoveled.

The U.W. 102b-437 $ldm_2$ is buccolingually narrow and mesiodistally long, with an ovo-rectangular shape. Five well-developed cusps are present in a Y-fissure pattern, and the talonid is wider than the trigonid. Compared to the two lower $dm_2$ specimens from the Dinaledi chamber, the metaconid and protoconid are relatively small, although it is not clear whether or not this is an artifact of mesial wear. Although the mesial aspect is worn, it is clear that the mesial marginal ridge was thick; it forms the distal border of a fissure-like anterior fovea. A thick distal marginal ridge is also present and it borders a fissure-like posterior fovea. Both the mesio- and disto-buccal grooves are deep and are associated with a wide V-shaped furrow. No protostylid is present. The mesial and distal roots each have a buccal and lingual canal connected by a dentin plate. The crown morphology is nearly identical to that of the two analogous teeth from the Dinaledi chamber (U.W. 101–655 and U.W. 101–1686), and supports their taxonomic designation to *H. naledi*. Although few have been found, the lower $dm_2$s of *H. erectus s.l.* are relatively longer and more rectangular than those of *H. naledi*. In addition, the substantially wider talonid compared to the trigonid of both the 102 tooth and the *H. naledi* lower $dm_2$s differentiates these $dm_2$s from those of most other *Homo*.

The most important of the 102b teeth for morphological assessment is the $LC_1$. The U.W. 102b-511 crown preserves its occlusal morphology and replicates with better detail the same morphology observed in the LES1 mandibular canines (*Figure 30*). It has asymmetrically placed crown shoulders, with the mesial shoulder more apically placed than the distal. Further, the distal shoulder is formed by an accessory cuspule and the mesial crest is shorter and more convex than the vertically disposed distal crest. These features are identical to those found in the Dinaledi permanent mandibular canines and are among the defining features of *H. naledi*. The asymmetrical shoulders of the canine crown and distal accessory cuspule are found to some degree in a number of specimens attributed to *H. erectus*, *H. rudolfensis*, *H. habilis* and *Australopithecus*, but the small canine and incisor sizes are inconsistent with attribution to any of these other taxa. On the basis of these observations, the immature dentition from 102b is identifiable as *H. naledi*.

All of the cranial fragments recovered from 102b are compatible with an immature developmental stage, and may come from the same individual as U.W. 102b-438. The minimum number of individuals (MNI) in 102b is therefore one individual at present. Without more information about the depositional history of the 102b locality in comparison with that of 102a, we cannot say for sure whether any of the specimens collected at this locality may represent the same immature individual that is represented by the pelvic fragments in 102a. The evidence does not currently exclude the

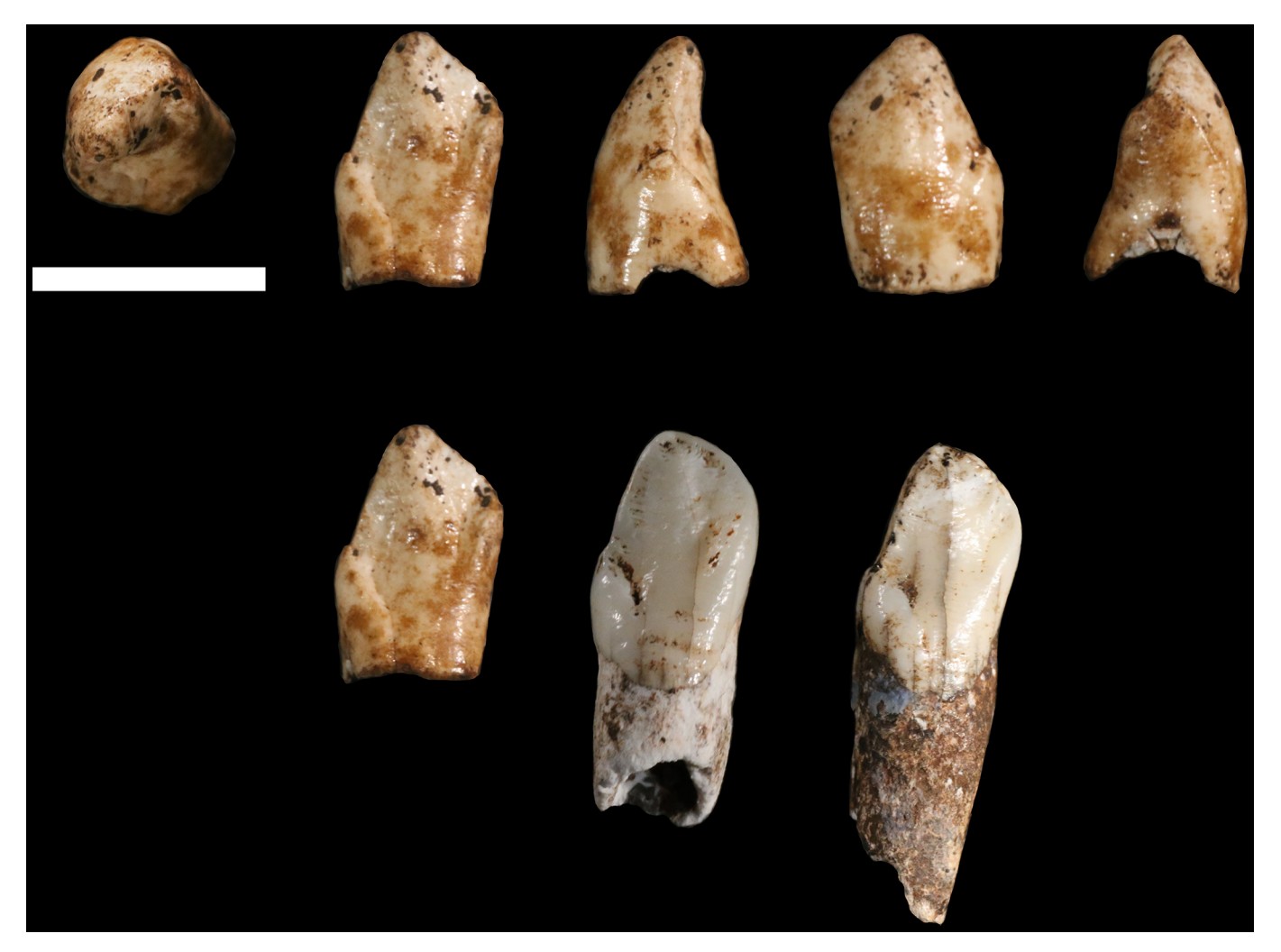

**Figure 30.** U.W. 102b-511 left mandibular canine crown from locality 102b. Top row, from left: occlusal, lingual, distal, labial and mesial views. Bottom row: U.W. 102b-511 (left), compared to U.W. 101–1126 (middle) and U.W. 101–985 (right) relatively unworn left mandibular canines from the Dinaledi Chamber. All three teeth share a distinctive morphology, which is also present in the other Dinaledi mandibular canines, that includes an asymmetrical crown, higher mesial shoulder, and distal accessory cuspule. Scale bar = 1 cm.

hypothesis that the U.W. 102a-138 ilium and other immature pelvic fragments may represent the same individual as the U.W. 102b-438 mandible. The 102a and 102b areas are separated by approximately 3 m, and there is a possibility that slumping of material might have brought the remains of a single individual into both areas. Further work to establish the life-history stage of the immature remains may help to resolve this question.

## Hominin material from 102c

The 102c deposit is a very small (~2 L) volume of sediment enclosed within a dissolution cavity in the cave wall. Only one morphologically identifiable hominin specimen has been recovered from this area. **U.W. 102c-589** is a fragment of left mandibular corpus with worn $LM_1$ and $LM_2$ in situ (*Figure 31*). These teeth have less occlusal wear than those of the LES1 mandible, but are morphologically very similar. The mandibular corpus is broken irregularly at the alveoli and the base of the mandible is not present. Anteriorly, it is broken at the mesial $M_1$ root; posteriorly, it is broken at the mesial alveolus for the $M_3$. The root of the ascending ramus is preserved and becomes independent

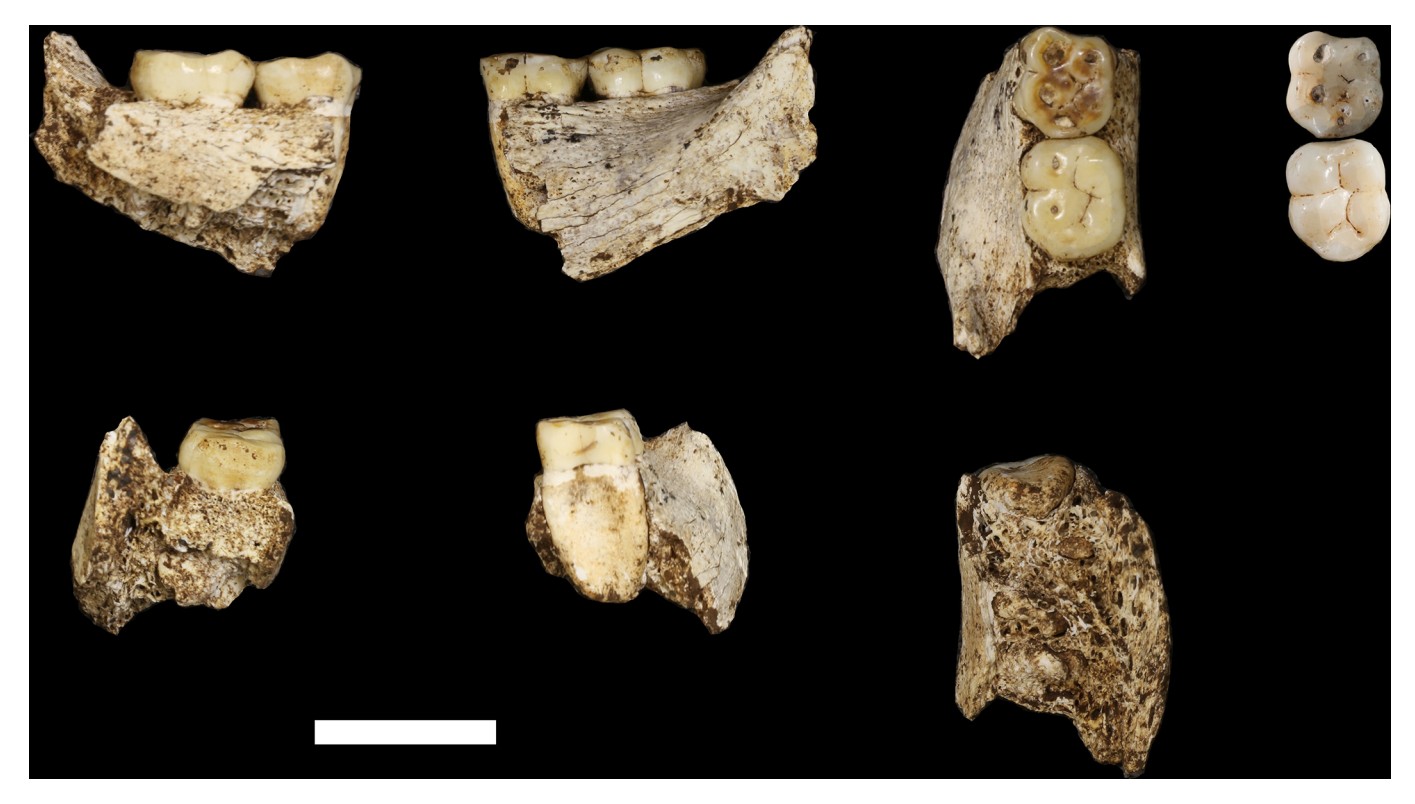

**Figure 31.** U.W. 102 c-589 mandibular fragment.  Top row, from left: lingual view, buccal view and occlusal view. Top right: two *H. naledi* teeth from the Dinaledi Chamber that present comparable occlusal morphology and wear to the U.W. 102 c-589 teeth: U.W. 101–297 $RM_1$ (top, reversed to represent left side) and U.W. 101–284 $LM_2$ (bottom). Bottom row, from left: U.W. 102 c-589 posterior view, anterior view and basal view. Scale bar = 2 cm.

at approximately the midpoint of $M_2$. Corpus breadth at $M_2$ is approximately the anatomical value at 19.8 mm; the broken corpus does not permit this measurement elsewhere.

The morphology of the mandible and tooth crowns is similar to that of the mandibular remains of *H. naledi* from the Dinaledi Chamber. The size and preserved morphology of the mandibular corpus and root of the ascending ramus are very similar to that of the DH1 holotype of *H. naledi*. The morphology and wear stage of the U.W. 102 c-589 $LM_1$ are nearly identical to those of the U.W. 101–297 $RM_1$ crown from the Dinaledi sample, and the $LM_2$ is very similar in morphology to the U.W. 101–284 $LM_2$ from Dinaledi (*Figure 31*). Neither of the U.W. 102 c-589 teeth exhibit supernumerary cusps, and both have simple crowns with a Y-5 cusp pattern without crenulation or complexity, similar to the Dinaledi teeth. The areas represented by the cusps appear similar to those in the Dinaledi molars, as do the crown heights. The relatively small size of the mandible and molars rule out attribution to *Au. africanus* or *P. robustus*, and the crown morphology rules out attribution to *H. erectus* or *H. habilis*. This mandible is entirely consistent with *H. naledi*.

The U.W. 102 c-589 mandible clearly duplicates the LES1 mandible, thereby demonstrating the presence of a second adult individual in the Lesedi Chamber as a whole. Nevertheless, it is more difficult to determine whether the U.W. 102 c-589 individual also contributed a femur to 102a, accounting for the evidence of two adult individuals in that deposit. The 102c deposit is separated by a distance of approximately 12 m from the 102a assemblage. While this distance does not rule out the occurrence of the remains of a single individual in both locations, the context does not provide any reason for an assumption that they are the same.

## Taphonomy

The general preservation of the Lesedi Chamber hominin material resembles that of the skeletal assemblage from the Dinaledi Chamber (*Dirks et al., 2015*). The Lesedi skeletal material has a surface coloration that ranges from light grey to red-brown, and internal structures or cortex at fresh break points are colored pale buff to off-white, contrasting with unbroken adjacent surfaces. The remains present no evidence of calcite crystal formation. Specific comments on preservation have been included with the specimen descriptions above, and taphonomic observations on each specimen are summarized in *Supplementary file 5*.

The Lesedi skeletal assemblage exhibits varying degrees of post-mortem damage. Most specimens are fragmented or broken to some degree, with areas of cortical bone removal or abrasion (*Supplementary file 5*). All fractures observed in the assemblage are consistent with post-mortem (dry bone) failure (*Supplementary file 5*), and there are no spiral or incomplete fractures indicative of green or wet bone (*L'Abbé et al., 2015*; *Symes et al., 2014*). Fractures in the assemblage include transverse or right-angled breaks on the shafts of long bones, showing block-comminution between major breaks and step-fractures following the longitudinal grain of the long bone, which are distinct from cracking or crazing resulting from weathering (*Symes et al., 2014*).

The Lesedi remains display no evidence of sub-aerial weathering processes (*sensu Behrensmeyer, 1978*; see also *Lyman and Fox, 1989* and *Junod and Pokines, 2014*, indicating that the dry bones were not subject to surface processes before deposition in the cave environment. Sub-aerial weathering is primarily characterized by both cracking and delamination. Within the Lesedi assemblage, there is some evidence of surface cracking, but it generally does not penetrate deep into the cortex. No indications of the secondary features of weathering, such as delamination, deep patination, bleaching or cortical exfoliation, were observed in any of the bone fragments from Lesedi, suggesting that the bones were not affected by surface exposure (*Supplementary file 5*). A comparison with human bone derived from forensic or sealed archaeological contexts (*Figure 32*) suggests that the cracking and fracture patterns observed in the Lesedi assemblage can be explained most parsimoniously by the effects of the burial environment (*Pokines, 2014*), and were exacerbated in specimens where fluctuations in moisture content caused swelling and shrinkage of the cortical structure (*Conard et al., 2008*; *Dirks et al., 2016*; *Dirks et al., 2015*).

The assemblage presents no evidence of high-energy fluvial transport, such as smoothing and rounding, polish, frosting, cortex thinning, or aperture formation (*Bassett and Manhein, 2002*; *Behrensmeyer, 1988*; *Evans, 2014*; *Nawrocki et al., 1997*). We have found no traces of carnivore or scavenger modification, with an absence of bone cylinders, tooth scores or traces of gastric corrosion (*Blumenschine et al., 1996*; *Haynes, 1983*; *Hill, 1976*; *Pickering et al., 2004*; *Thompson and Lee-Gorishti, 2007*). In addition, the profiles of damaged diaphyses show no evidence of end gnawing, scalloping or flaking (*Blumenschine et al., 1996*; *Lyman, 1994*; *Wood, 1991*). In some cases, missing areas of epiphyses display patterns of localized cortical destruction, which may be consistent with a process referred to as 'coffin wear' (*Rogers, 2005*; *Schultz, 2012*; *Schultz et al., 2003*) whereby bones come into direct contact with an underlying substrate following soft tissue decomposition; this may lead to the loss of cortex or of portions of elements that are in contact with the substrate. This sub-surface process may completely remove processes or condyles from major elements such as the femur or humerus, leaving behind flattened or sheared areas of bone (*Rogers, 2005*; *Pokines and Baker, 2014*).

As with the Dinaledi material, some of the Lesedi Chamber remains are stained by iron oxides and manganese oxy-hydroxide (*Dirks et al., 2015*; *Randolph-Quinney et al., 2016*). This staining primarily occurs as diffuse spots or mats of mineral, some of which cover and encapsulate the fractured ends of long bones, indicating a complex post-mortem history of mineral deposition. However, we have observed no mineral tide marks upon the Lesedi Chamber skeletal material which, unlike material from the Dinaledi Chamber, provides us with no information about the former position of skeletal remains relative to the sediment–air interface. Some of the Lesedi material exhibits minor pitting, striations, grooves, scratches, or gouges. Striations and grooves are consistent with abrasion marks, as defined by *D'Errico and Villa (1997)* and *Fisher (1995)*. Some of the elements have a palimpsest of taphonomic traces, with mineral deposition preceding invasive surface modification or fracturing (*Figure 32*). The pitting that is present on the Lesedi material shows that the modifying agents (abiotic or biotic) penetrated already-altered bone, suggesting that damage occurred on

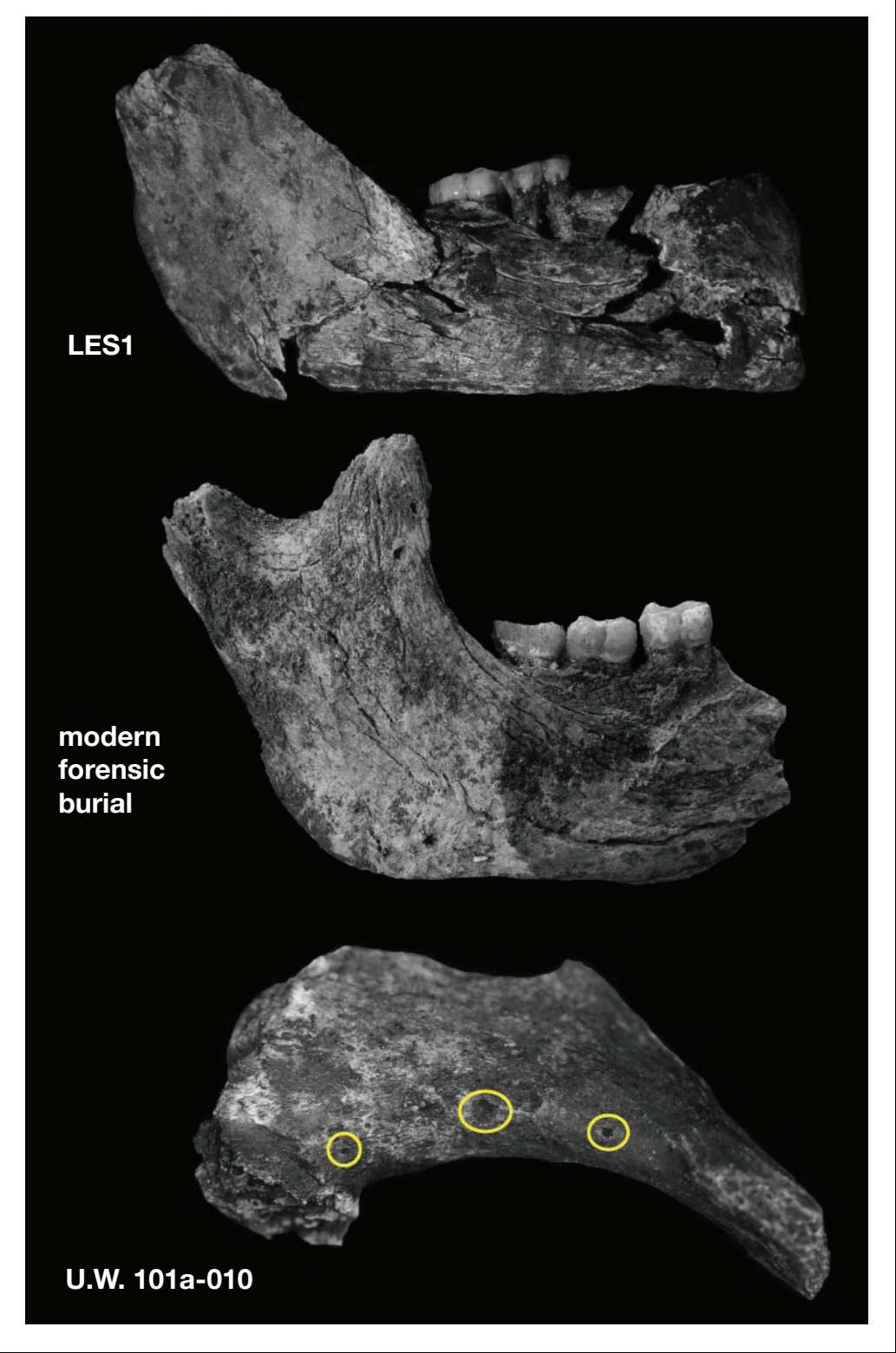

**Figure 32.** Surface cracking and pitting in Lesedi Chamber material. Top: right corpus and ramus of LES1 mandible prior to reconstruction, showing patterns of cracking consistent with the effects of sediment loading and wetting during the burial process and during skeletal decomposition. This pattern of taphonomic alteration is superficially similar to sub-aerial weathering processes, but is independent of surface exposure and occurs in both deep- and shallow-buried deposits. Middle: forensic known-history case for comparison. This specimen was

*Figure 32 continued*

recovered from a deep known-history inhumation. The fleshed body was deposited at a depth of 1.5 metres and recovered after a 30-year burial period. Note the similarities in surface texture, and superficial cracking, which follows the biomechanical stress lines (grain) of the bone. Bottom: U.W. 102a-010 acromial fragment showing surface modification with cortical pitting and punctate marks. Note that many of these marks (circled in yellow) penetrate pre-existing layers of manganese oxy-hydroxide deposited on the bone surface; the damage was therefore produced inside the Lesedi Chamber on bones that were already covered in coatings of manganese mineral.

bone surfaces that were already covered in coatings of manganese and iron oxide (*Randolph-Quinney et al., 2016*). Generally, the pattern of pitting and stripping is similar in composition and gross morphology to patterns observed in the Dinaledi Chamber assemblage, which were attributed to gastropod and other invertebrate activity (*Dirks et al., 2015, 2016*; *Randolph-Quinney et al., 2016*), although these must be studied in more detail to be certain.

Overall, the gross taphonomic signature of the hominin remains in the Lesedi Chamber appears consistent with sub-surface deposition with limited post-depositional dispersal. Particularly in the U. W. 102a locality, this is evidenced by the presence of articulated remains, the proximity of cranial and postcranial remains, and the recovery of small elements such as teeth and carpals. The bone surface condition, as well as the absence of other markers of transport and secondary modification, are generally consistent with sub-surface skeletal decomposition (*Carter and Tibbett, 2008*; *Carter et al., 2007*). At present, we find no supporting evidence for sub-aerial weathering or postmortem exposure outside of the cave environment (*Hill, 1976*; *Junod and Pokines, 2014*; *Lyman and Fox, 1989*; *Tappen and Peske, 1970*). Nor are there signs of carnivore or scavenger modification (*Pokines, 2013*), water transport (*Behrensmeyer, 1988*; *Evans, 2013*), or peri-mortem trauma such as may be expected in a natural death-trap scenario (*L'Abbé et al., 2015*; *Symes et al., 2014*).

## Faunal material

In addition to the hominin skeletal material, some faunal remains have been recovered from the Lesedi Chamber. Over 80 faunal elements or fragments were collected from U.W. 102a. U.W. 102b yielded 23 specimens of fauna, while a single rodent tooth was recovered from U.W. 102c. In-depth analyses of the faunal material are ongoing, and comprehensive descriptions are in preparation. We provide here a preliminary list of the taxa identified for reference (*Table 3*).

The Lesedi faunal assemblage includes micromammal, small to mid-size mammal, and non-mammalian remains. With respect to the micromammals (prey body mass <500 g), 5 genera of rodents and 1 genus of shrew were identified out of 28 craniodental specimens (21 MNI). There are also potentially two additional murine genera and one soricid genus present in the assemblage, although lower-level identification is not possible at this time. Interestingly, all of the non-hominin fauna are of relatively small species. The largest mammalian specimens come from dental material attributed to *Canis* aff. *C. familiaris*. The size range of this material is outside the range of modern *C. mesomelas* but the morphology is definitively not *Lycaon* (*Hartstone-Rose et al., 2010*; *Wayne, 1986*). The felid material is also small, falling in the size range of the African wildcat. None of these individuals is likely to have exceeded 10–15 kg (*Kingdon, 2015*) and the rest of the assemblage consists almost entirely of animals smaller than 3 kg, including four specimens from the family *Herpestidae*. Aside from a single lagomorph specimen, the macro-mammalian material comes exclusively from the order *Carnivora*, a situation that is unusual in the fossil record (*Werdelin and Sanders, 2010*).

We do not presently know whether some or all of these faunal remains may be contemporaneous with any of the hominin fossil material. Faunal remains have been recovered both on the surface and also from within sediments near hominin remains. However, the Lesedi Chamber is not a completely isolated environment, and sediment deposits are currently eroding from their original depositional contexts, with evidence for slumping and reworking in the chamber. Therefore, we cannot yet comment on the relative timing of deposition of the hominin and faunal material. Additional tests, including attempts to date both hominin and faunal elements directly, will help us answer these questions

**Table 3.** Mammal species recorded in the Lesedi Chamber. Several specimens from the classes Aves, Amphibia and Reptilia were also recovered, but individual counts and taxonomic identifications are pending further examination.

| Class | Order | Family | Subfamily | Genus/species | MNI | NISP |
|---|---|---|---|---|---|---|
| Mammalia | Lagomorpha | Leporidae | | | 1 | 1 |
| | Soricomorpha | Soricidae | | | 3 | 3 |
| | | | Crocidurinae | *Crocidura* | 1 | 1 |
| | Rodentia | | | | 1 | 1 |
| | | Bathyergidae | Bathyerginae | | 1 | 1 |
| | | Muridae | | | 1 | 1 |
| | | | Otomyinae | *Otomys* | 1 | 1 |
| | | | Murinae | | 5 | 7 |
| | | | | *Mus* | 1 | 2 |
| | | Nesomyidae | Mystromyinae | *Mystromys* | 1 | 1 |
| | | | Dendromurinae | | 3 | 4 |
| | | | | *Steatomys* | 3 | 6 |
| | Carnivora | Felidae | | *Felis* aff. *F. sylvestris* | 1 | 1 |
| | | | | *Felis* cf. *sylvestris* | 1 | 6 |
| | | | | *Felis* sp. | 1 | 2 |
| | | Herpestidae | | cf. *Mungos* | 1 | 1 |
| | | | | cf. Herpestidae | 1 | 3 |
| | | Canidae | | *Canis* aff. *C. familiaris* | 1 | 14 |
| | | | | *Canis* cf. *mesomelas* | 1 | 3 |
| | | | | *Canis* sp. | 1 | 21 |
| | | | | *Vulpes* cf. *chama* | 1 | 3 |
| | | | | cf. *Vulpes* | 1 | 6 |
| | | | | *Vulpes* sp. | 1 | 5 |

and to relate the faunal material to the chronological and environmental context of *H. naledi* in the Lesedi Chamber.

## Discussion

The hominin material from all three excavation contexts in the Lesedi Chamber can be clearly attributed to *H. naledi*, and no diagnostic material in this chamber represents any other hominin taxon. This confirms that a second chamber within the Rising Star cave system, in a distinct depositional context isolated from the Dinaledi Chamber, also holds the skeletal remains of *H. naledi*.

As in the Dinaledi Chamber, both adult and immature individuals are commingled in the Lesedi Chamber. When the fossils recovered from the 102a, 102b, and 102c localities are considered in aggregate, duplication of elements and age-at-death indicators combine to produce a MNI of three: two adults and one immature individual. However, the presence of material in three spatially separated localities within the chamber leads us to hypothesize that the current collection more probably represents additional individuals. The adult mandibular fragment in 102c is separated from the adult femora in 102a by more than 10 m, and the presence of this fragment in 102c, with an opening 1.3 m above the present cave floor, suggests that the sedimentary deposits may once have been much more voluminous. In light of the situation, we view it as likely that the 102c mandibular fragment represents an additional adult individual beyond what may be two individuals in 102a. The 102b deposit is separated by less distance from 102a, and the data do not yet provide a clear answer as to whether or not the juvenile remains in both areas could represent a single individual. Considering the spatial context, we hypothesize that four individuals may be represented: two adults

and one immature individual in 102a and 102b; and one adult in 102c. Further excavation may test this hypothesis.

The Lesedi Chamber sample extends our knowledge of *H. naledi* in several ways. The complete U.W. 102a-021 clavicle confirms the morphological assessment of fragmentary clavicles from the Dinaledi Chamber (*Feuerriegel et al., 2017*). The U.W. 102a-036 and U.W. 102a-151 T10 and T11 vertebrae duplicate the anatomy observed in the Dinaledi material (*Williams et al., 2017*), while additional vertebrae add information on the lower vertebral column. The U.W. 102a-250 first rib is more complete than any example from the Dinaledi Chamber, but presents similar anatomy (*Williams et al., 2017*). U.W. 102b-438 and its associated teeth present an additional immature mandibular specimen that adds to the developmental series of mandibles and immature dentitions from the Dinaledi Chamber. Comparative study of these elements is underway.

The relative completeness of the LES1 cranium allows us to examine cranial length and the nasal aperture for the first time in *H. naledi*, in part confirming the interpretation of partial crania and isolated cranial elements from the Dinaledi Chamber. This specimen shows a relatively short and tall cranium and a moderate-sized nasal aperture. This specimen expands the range of endocranial volume (ECV) known for *H. naledi,* which now extends from approximately 460 ml to approximately 610 ml. The maximum value for *H. naledi* is now somewhat above the maximum ECV observed for australopith species, including *Au. afarensis*, *Au. africanus*, *P. robustus,* and *P. boisei* (*Table 4*; *Figure 33*). No crania attributed to *H. habilis*, *H. rudolfensis*, or *H. erectus* have ECVs as small as the 460 ml estimated for DH3, but the larger *H. naledi* specimens do overlap with the smaller end of the reported values for *H. habilis* and *H. erectus*. The single specimen of *H. floresiensis* with an estimate of endocranial volume, LB1, is smaller than any specimen of *H. naledi*. The addition of the LES1 cranium now brings the range of observed ECVs in *H. naledi* into overlap with two specimens of *H. erectus* (D2700 and D4500 from Dmanisi; *Table 4*, *Figure 33*).

The hominin material from the Lesedi Chamber remains undated. The Dinaledi Chamber hominin material appears to have been deposited sometime between 236 ka and 335 ka (*Dirks et al., 2017*). The Dinaledi hominin sample is morphologically very uniform, and variations in many metric characters in that sample is less than those in local populations of modern humans (*Berger et al., 2015*). Such morphological uniformity suggests that the Dinaledi Chamber does not represent successive biologically diverse populations sampled over tens of thousands of years or longer, known as 'time-averaging', but instead may sample a single biological population. The Lesedi Chamber *H. naledi* sample adds very little to the morphological variability of the Dinaledi Chamber sample, and most measurements of cranial, dental, and postcranial elements fall within the existing range known from that chamber. The nonmetric traits observed on cranial and postcranial elements from the Lesedi Chamber are nearly all duplicated in the Dinaledi Chamber sample (*Supplementary file 1*; *Figures 34* and *35*). It is therefore reasonable to hypothesize that the Lesedi Chamber may sample the same biological population as the Dinaledi Chamber. This hypothesis will be tested with further geological information, including attempts to apply direct dating to the hominin material directly. If the two hominin assemblages represent diverse geological ages, or if either assemblage was deposited over a geologically long time interval, they would demonstrate a remarkable stasis of *H. naledi* metric and morphological characters across time.

The emphasis of this study has been to test the hypothesis that the newly discovered Lesedi skeletal material represents *H. naledi*. Based upon this diagnosis, the morphology of *H. naledi* (including the Lesedi material) across the skeleton has implications for its relationship to other Plio-Pleistocene hominins (*Figures 34* and *35*). Most of the anatomical features across the skeleton of *H. naledi* are known from multiple specimens, many of which have now been replicated in two assemblages (*Figure 35*). Phylogenetic analysis of the Dinaledi hominin assemblage has shown that the mosaic of features in the *H. naledi* cranium and dentition does not provide unambiguous or clear evidence of where the species should be placed in the phylogeny of *Homo* (*Figure 34*; *Dembo et al., 2016*). We briefly consider here how the overall pattern of morphological and temporal evidence relates to the possible affinities of *H. naledi* with other hominins.

## Modern and archaic humans

The Dinaledi Chamber assemblage of *H. naledi* represents individuals who lived at roughly the same time as skeletal remains attributed to early modern or near-modern humans in some parts of Africa (*Figure 36*; *Berger et al., 2017*). No derived features of the *H. naledi* skeleton require a close or

**Table 4.** Endocranial volume of LES1 compared to key specimens of other hominin species. Estimates from *Asfaw et al. (1999)*, *Kubo et al. (2013)*, *Holloway et al., (2014)*, *Lee and Wolpoff (2003)*, *Kimbel (2004)*; *Berger et al. (2010)*, *(2015)*, *Lordkipanidze et al. (2006)* and *Lordkipanidze et al. (2013)*.

| | Specimen | Cranial capacity in cc |
|---|---|---|
| *Au. afarensis* | Mean | 444 |
| | AL 162–28 | 400 |
| | AL 288–1 | 387 |
| | AL 333–45 | 485 |
| | AL 333–105 | 400 |
| | AL 444–2 | 550 |
| *Au. africanus* | Mean | 455 |
| | MLD 1 | 510 |
| | MLD 37/38 | 425 |
| | Sts 5 | 485 |
| | Sts 19 | 436 |
| | Sts 60 | 400 |
| | Sts 71 | 428 |
| | StW 505 | 505 |
| *Au. sediba* | MH1 | 420 |
| *H. floresiensis* | LB1 | 426 |
| *H. naledi* | Mean | 513 |
| | DH1 | 560 |
| | DH3 | 465 |
| | LES1 | 610 |
| *H. habilis* | Mean | 616 |
| | KNM-ER 1805 | 582 |
| | KNM-ER 1813 | 509 |
| | O.H. 7 | 729 |
| | O.H. 13 | 650 |
| | O.H. 16 | 638 |
| | O.H. 24 | 590 |
| *H. rudolfensis* | Mean | 789 |
| | KNM-ER 1470 | 752 |
| | KNM-ER 1590 | 825 |
| | KNM-ER 3732 | 750 |
| *H. erectus* | Mean | 917 |
| | BOU-VP-2/66 | 995 |
| | D2280 | 730 |
| | D2282 | 650 |
| | D2700 | 601 |
| | D3444 | 625 |
| | D4500 | 546 |
| | KNM-ER 3733 | 848 |
| | KNM-ER 3883 | 804 |
| | KNM-ER 42700 | 691 |

*Table 4 continued on next page*

*Table 4 continued*

| Specimen | Cranial capacity in cc |
| --- | --- |
| KNM-WT 15000 | 900 |
| O.H. 9 | 1,067 |
| O.H. 12 | 727 |
| Sangiran 2 | 813 |
| Sangiran 4 | 908 |
| Sangiran 17 | 1,004 |
| Zhoukoudian DI | 915 |
| Zhoukoudian LI | 1,025 |
| Zhoukoudian LII | 1,015 |
| Zhoukoudian LIII | 1,030 |
| Ngandong 1 | 1,172 |
| Ngandong 5 | 1,251 |
| Ngandong 6 | 1,013 |
| Ngandong 9 | 1,135 |
| Ngandong 10 | 1,231 |
| Ngandong 11 | 1,090 |
| Trinil | 940 |

exclusive relationship with modern humans. Overall, *H. naledi* resembles more primitive species of *Homo* such as *H. erectus, H. habilis,* or *H. rudolfensis* much more than it resembles archaic or modern humans (*Berger et al., 2015*; *Laird et al., 2017*; *Marchi et al., 2017*; *Feuerriegel et al., 2017*;

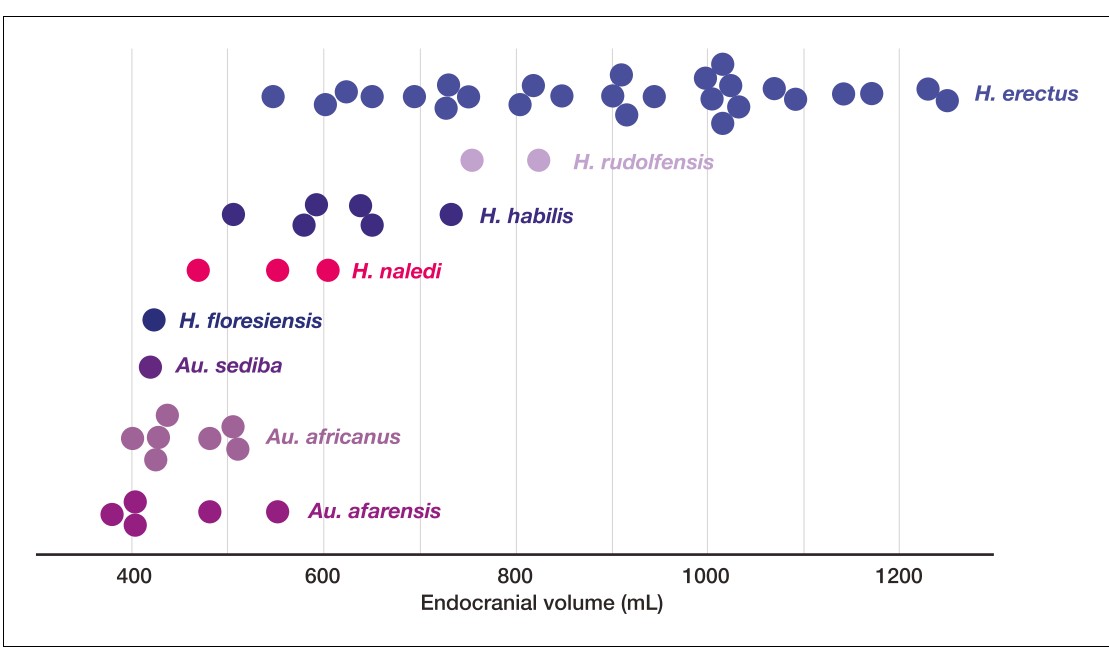

**Figure 33.** Endocranial volumes of hominin species. With the addition of LES1 to the sample, the range of endocranial volume in *H. naledi* is extended slightly beyond the range represented in the Dinaledi Chamber. This range overlaps with two specimens of *H. erectus,* and LES1 is larger than the largest *Au. africanus* or *Au. afarensis* specimens. Data and sources are listed in *Table 4*.

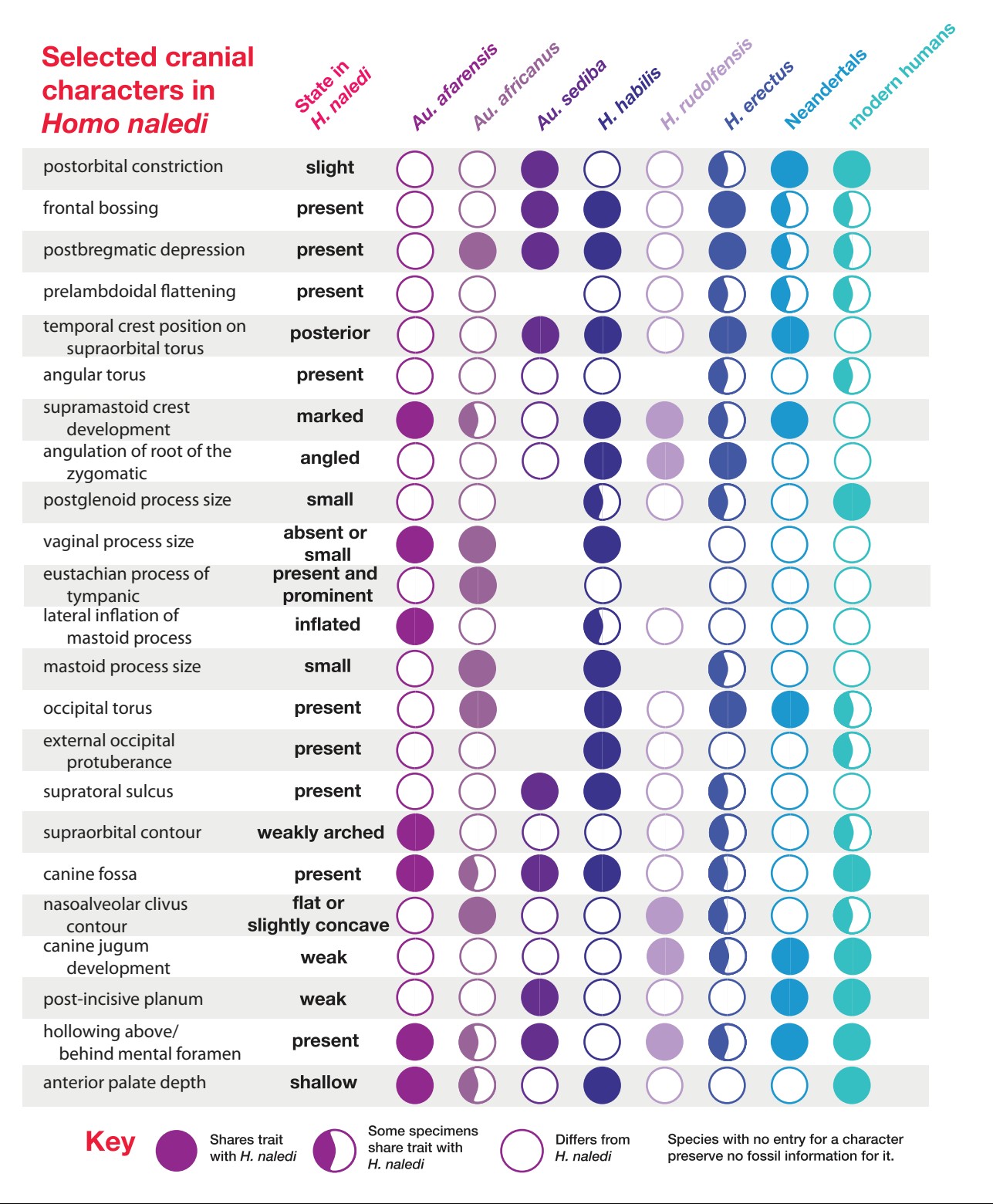

**Figure 34.** Selected cranial trait observations in *H. naledi* and other species.  A subset of observations of cranial traits reported in ***Supplementary file 1*** that vary among species attributed to *Homo*. This list omits traits that are present in only one species of *Homo* or for which nearly all species exhibit more than one state. Here, the character state observed in *H. naledi*, including both Dinaledi Chamber and Lesedi Chamber material, is reported on the left. Character states for other species are reported in terms of whether they share the same state as *H. naledi*. Some traits that are shared with *H. naledi* may be interpreted as shared derived traits, based on their absence from *Au. afarensis* and *Au. africanus*, but some species of *Homo* also

*Figure 34 continued on next page*

*Figure 34 continued*

share primitive traits with *H. naledi* that may also be found in the australopiths. The pattern of shared morphological characters among these species and *H. naledi* is complex, and there is no consistent anatomical grouping of the characters shared with any given species. *H. naledi* shares more characters with *H. erectus* and *H. habilis* than with other species, and behind them, with *Au. sediba* and archaic and modern humans. *H. erectus* is variable for a large number of traits also found in *H. naledi*. The fact that variability is noted in *H. erectus*, Neandertals and modern humans for these traits is partially a function of the large samples available for these groups. It is probable that a larger sample of other species would likewise encompass greater variability. Any phylogenetic tree of these species would reveal a high degree of homoplasy for these cranial and mandibular traits.

*Williams et al., 2016*; *Schroeder et al., 2017*). *H. naledi* does, however, possess a number of derived features that are otherwise known only from modern humans and Neandertals (*Supplementary file 1*; *Figures 34* and *35*; *Berger et al., 2015*; *Dembo et al., 2016*; *Kivell et al., 2015*; *Harcourt-Smith et al., 2015*; *Williams et al., 2016*). Some of these derived features, including features of the wrist, cannot be assessed in *H. erectus* because no fossils of the relevant bones exist for this species (*Kivell et al., 2015*). But others, including features of the cranium and dentition, raise at least the possibility that *H. naledi* may be a sister to *H. antecessor* or to a clade including *H. antecessor* with other archaic and modern humans (*Dembo et al., 2016*). It is also conceivable that, rather than indicating a recent branching of *H. naledi* from an archaic human lineage, such derived similarities may have resulted from introgressive hybridization between *H. naledi* and other hominin lineages (*Berger et al., 2017*), although testing this hypothesis will likely require genetic data.

## Homo floresiensis

The late Middle Pleistocene geological age of *H. naledi*, when considered together with its small endocranial volume, prompt comparisons with *H. floresiensis*. Skeletal remains of *H. floresiensis* are known from ~100–60-ka-old sediments at Liang Bua, Flores (*Brown et al., 2004*; *Sutikna et al., 2016*), and possibly also in Mata Menge, Flores sediments from the early Middle Pleistocene (*van den Bergh et al., 2016*; *Brumm et al., 2016*). The features shared by *H. naledi* and *H. floresiensis* are nearly all inferred to be primitive features shared with australopiths and/or in some cases with early *Homo* (*Figures 35* and *36*). For most features of the skeleton where *H. naledi* exhibits derived morphology, including aspects of the foot and hand, *H. floresiensis* exhibits morphology thought to be primitive within hominins (*Harcourt-Smith et al., 2015*; *Kivell et al., 2015*; *Tocheri et al., 2007*; *Jungers et al., 2009a*; *Orr et al., 2013*). All known *H. floresiensis* long bones are small, reflecting small stature and mass (*Jungers et al., 2009b*; *Larson et al., 2009*; *Grabowski et al., 2015*), and none approach the much larger size, particularly the taller stature, manifested in adult specimens of *H. naledi*. The mandibles of *H. floresiensis* are relatively robust, even for their small size, (*Brown and Maeda, 2009*; *Daegling et al., 2014*) and in this sense, are comparable to the robust mandibles of *H. naledi*, though the two species differ in symphyseal morphology (*Laird et al., 2017*). The dentition of *H. floresiensis* displays a unique combination of primitive and derived characters (*Kaifu et al., 2015*), which are different from the combination found in *H. naledi*. *Homo floresiensis* canines and premolars display a combination of features that are primitive for hominins (*Kaifu et al., 2015*), while *H. naledi* mandibular canines and third premolars have a uniquely derived form. The *H. floresiensis* molars exhibit derived proportions and some other features similar to those of modern humans, which may reflect the extremely shortened molar crowns, particularly of the first molars (*Kaifu et al., 2015*). By contrast, *H. naledi* has primitive proportions of the molars and occlusal morphology that is broadly primitive within *Homo*, although simplified in complexity. Shape analyses also place the cranium of *H. floresiensis* (LB1) far from any *H. naledi* cranial specimen (*Schroeder et al., 2017*). These differences in anatomy may reflect a distant phylogenetic relationship between the two species, although these data do not clearly resolve where on the *Homo* phylogeny either species should be placed. Some phylogenetic evidence suggests that the *H. floresiensis* branch preceded the node linking *H. erectus* and modern humans (*Argue et al., 2009*; *Dembo et al., 2015*). Cranial and dental evidence does not resolve whether *H. naledi* may also have branched early in the evolution of *Homo*, or whether it may instead be a sister taxon to modern and archaic humans (*Dembo et al., 2016*). No data that are available at present answer whether either or both of these lineages may have retained small EVC from an ancestry among the earliest members of *Homo*, or whether their smaller brain sizes may have evolved secondarily from larger-brained ancestors. Together, they

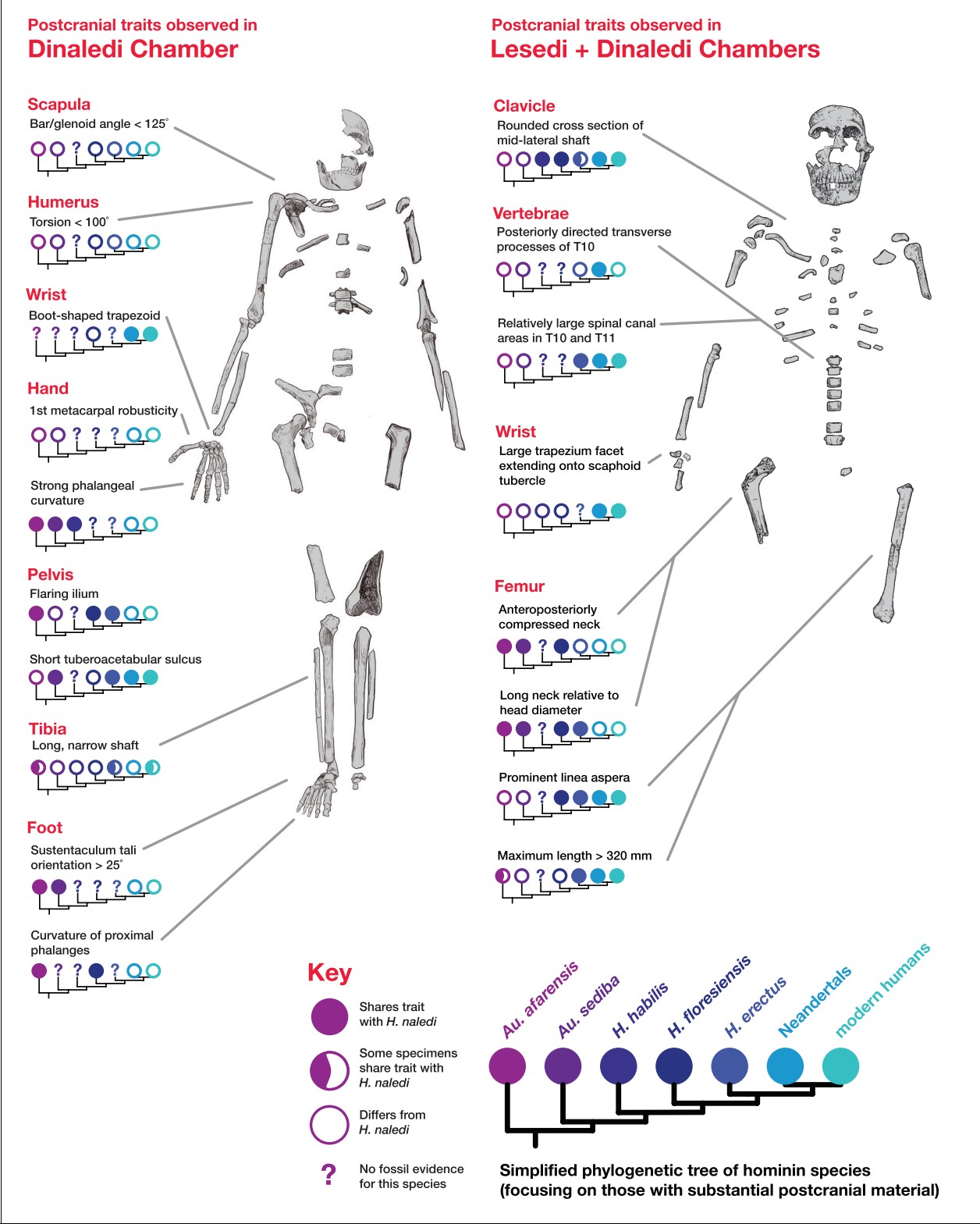

**Figure 35.** Postcranial traits in *H. naledi* compared to those in other hominin species. Here, a subset of features of the postcranial skeleton that distinguish *H. naledi* from other species are summarized in comparison to these traits in other hominin species with substantial postcranial evidence. These features include some that *H. naledi* shares with *Au. afarensis*, some that *H. naledi* shares with modern humans or Neandertals, and some that are unique or shared with even more distantly related species such as *Ardipithecus* (not shown). These traits constitute a mosaic that distinguishes *H.*

*Figure 35 continued on next page*

*Figure 35 continued*
*naledi* clearly from other species of *Homo*. Many of the traits that are notable in the Dinaledi Chamber sample are also represented in the Lesedi Chamber material.

establish that diverse hominin lineages with varying brain and body sizes existed during the Middle Pleistocene, suggesting the influence of ecological factors that promoted diversity during the Pleistocene evolution of humans and great apes (*Berger et al., 2017*; *Tocheri et al., 2011*, *2016*; *Dunn et al., 2014*).

## Homo erectus

*H. naledi* shares many derived cranial and dental characters with *H. erectus* (*Figures 34* and *36*; *Dembo et al., 2016*; *Laird et al., 2017*), and Dmanisi specimens of *H. erectus* are among the closest in multivariate shape to *H. naledi* crania (*Schroeder et al., 2017*). *H. naledi* overlaps with *H. erectus* in long bone lengths, inferred stature, and estimated body mass (*Figure 28*; *Berger et al., 2015*). To the extent that the *H. naledi* postcranial skeleton is different from material attributed to *H. erectus*, it is mostly because *H. naledi* manifests primitive traits that are not otherwise seen in *Homo*, or because *H. naledi* manifests traits not otherwise seen in hominins (*Figure 35*). *H. naledi* also shares several derived postcranial features with archaic and modern humans, but a lack of postcranial evidence for these parts of the *H. erectus* skeleton has made it impossible to determine whether *H. erectus* may also have shared these derived features (*Figure 35*). With respect to the cranium and dentition, nearly all of the nonmetric traits shared by *H. naledi* and *H. erectus* are also shared with *Au. sediba*, *H. habilis*, or both (*Supplementary file 1*; *Figure 34*). *H. naledi* and *H. erectus* share at least three derived nonmetric features of the skull that are not also found in *Au. sediba* or *H. habilis*: an angular torus, sagittal keeling, and an anteriorly projecting nasal spine. But each of these three traits can also be found in some archaic or modern humans. *H. naledi* also possesses some cranial and dental traits not seen in any specimens of *H. erectus*; these include both primitive traits shared with *Au. afarensis*, *Au. africanus* or *Au. sediba* and derived traits shared either with *H. rudolfensis* or with archaic and modern humans. Even though *H. naledi* crania are most similar in shape to the smallest *H. erectus* crania from Dmanisi (*Schroeder et al., 2017*), they are distinct from the Dmanisi sample in numerous aspects of cranial, dental, and postcranial morphology (*Berger et al., 2015*; *Laird et al., 2017*; *Marchi et al., 2017*; *Feuerriegel et al., 2017*; *Rightmire et al., 2017*), and Bayesian analysis of cranial and dental morphology provides strong evidence against the hypothesis of a sister taxon relationship for these samples (*Dembo et al., 2016*). In summary, no traits link *H. naledi* exclusively or specifically with *H. erectus*, and many traits distinguish the two. The evidence is not sufficient to say whether *H. naledi* may have evolved from an earlier population that resembled *H. erectus*, or whether earlier fossils representing the *H. naledi* lineage may already have been found and until now attributed to *H. erectus* (*Berger et al., 2017*).

## *Australopithecus sediba*, *Homo habilis*, and *Homo rudolfensis*

Fossil samples from Malapa, Olduvai Gorge, and the Lake Turkana area have been attributed to lineages often thought to represent some of the earliest species of *Homo* or their near relatives among the australopiths. *H. naledi* shares many derived cranial and mandibular features with *Au. sediba* and *H. habilis*, although most of these are also shared with *H. erectus* (*Supplementary file 1*; *Figures 34* and *35*). Compared to these other species, *H. naledi* shares many fewer cranial and dental features with *H. rudolfensis*, except for its relatively flat and squared nasoalveolar clivus. Phylogenetic analysis does not reject the hypothesis of a sister taxon relationship between *H. naledi* and *Au. sediba* or *H. habilis* (*Dembo et al., 2016*). However, in comparison to both *Au. sediba* and *H. habilis*, *H. naledi* is derived and similar to modern humans and Neandertals in several aspects of hand and wrist morphology (*Kivell et al., 2015*). It is also derived in its foot morphology in comparison to both MH2 of *Au. sediba* and the O.H. 8 foot usually attributed to *H. habilis* (*Harcourt-Smith et al., 2015*). *H. naledi* also lacks the derived configuration of the *Au. sediba* ilium (*Figure 35*). *H. naledi* further shares a number of derived cranial characters with *H. erectus* and with archaic or modern humans as discussed above (*Supplementary file 1*, *Figure 34*). Whatever the relationship of these species, their

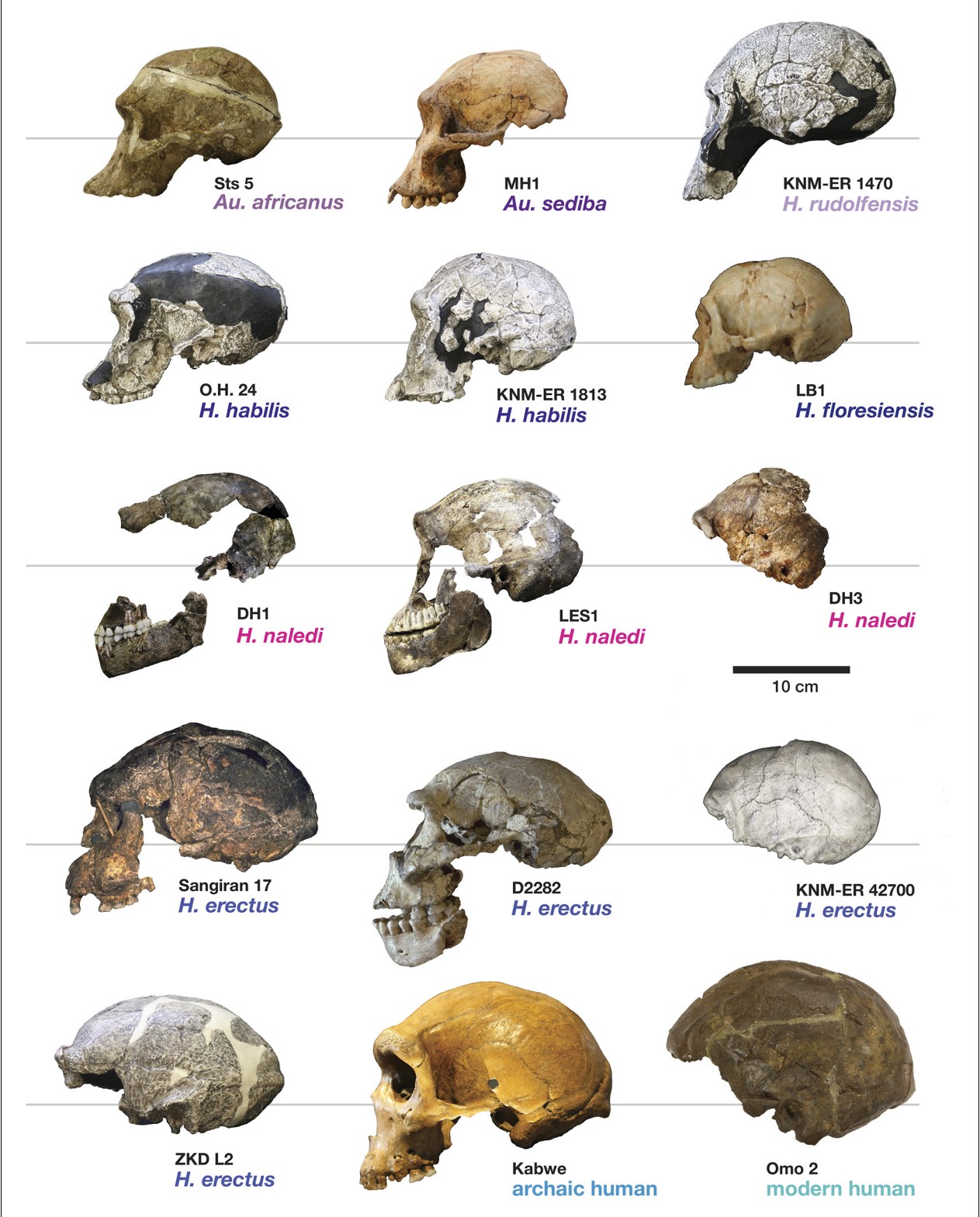

**Figure 36.** Lateral cranial comparison of *H. naledi* crania to crania of other hominin species. *H. naledi* crania, DH1, LES1, and DH3 are in the center row. All crania are oriented as near as possible to the Frankfort plane, delineated by the light gray lines in the background of the figure. Compared to other hominin genera, including *Australopithecus* and *Paranthropus*, fossil *Homo* is often recognized by cranial and dental features such as a more vertical face profile, a reduced postcanine dentition, larger endocranial volume, a higher frontal, and a true supraorbital torus. *Au. africanus* (Sts 5, top left) represents the ancestral hominin condition lacking these traits. The other crania in the top two rows vary substantially in these features. LB1 has a

*Figure 36 continued on next page*

*Figure 36 continued*

vertical face, reduced dentition, and high, rounded frontal, but has comparatively small endocranial volume. KNM-ER 1470 has a large volume, a high frontal, and a more vertical face profile, but also is inferred to have a large postcanine dentition and has no true supraorbital torus. MH1 (*Au. sediba*) has a small volume, but shares features with *Homo* that include the less sloping face profile, a supraorbital torus, and reduced postcanine dentition. O. H. 24 has a low, sloping frontal, and a concave facial profile, but a true supraorbital torus and reduced postcanine teeth. This variability among species that are interpreted as 'primitive' *Homo*, such as *H. habilis* and *H. floresiensis*, and *Homo*-like australopiths makes it difficult to delineate the genus *Homo* (**Wood and Collard, 1999**; **Dembo et al., 2016**). *H. erectus* is also highly variable. It includes several crania with endocranial volumes below 700 ml, including KNM-ER 42700 and D2282, but also many larger crania, here represented by Sangiran 17 and Zhoukoudian L2 (ZKD L2). Specimens attributed to *H. erectus* tend to share a series of traits first noted in Asian *H. erectus* samples, including a long, low cranial profile, thick cranial bone, sagittal keeling, prominent supraorbital, angular, and occipital tori, a sharply angled occiput, and a postbregmatic depression. The smallest *H. erectus* crania share most of these features, with a low cranial profile, angled occiput and postbregmatic depression visible here in D2282. But these features do vary substantially and are less evident in the immature KNM-ER 42700. The *H. naledi* crania are similar to KNM-ER 1470 in having a transversely flat clivus contour, but all are smaller, with a much smaller palate and with very different frontal morphology. Like O.H. 24 and KNM-ER 1813, the *H. naledi* crania have relatively thin cranial bone and a thin and projecting supraorbital torus. But *H. naledi* manifests a different clivus shape, a projecting nasal spine, a greater cranial height, sagittal keeling and an angular torus. The *H. naledi* crania bear little resemblance to LB1, differing in face profile, size, and their larger postcanine dentitions. Known African *Homo* specimens from the later Middle Pleistocene other than *H. naledi*, such as the Kabwe skull (pictured), contrast strongly with *H. naledi* in cranial size and morphology. The Omo 2 skull, one of the earliest known modern human crania at approximately 196,000 years (**McDougall et al., 2005**), is vastly larger and very different from any *H. naledi* specimen, despite being near the same geological age. In this figure, O.H. 24, KNM-ER 1470, LB1, KNM-ER 42700, ZKD L2, and Omo 2 are represented by casts. Images have been adjusted to a common scale by maximum cranial length, or by glabella-bregma length where maximum length is not available. Photos of Sangiran 17 and D2282 are courtesy of Milford Wolpoff.

evolution must have involved homoplasy of many features and the evidence does not yet make it clear how they are connected. If any one of these species were a possible ancestor of *H. naledi*, the branch leading to the Dinaledi sample of *H. naledi* would approach 1.5 Ma or longer in evolutionary time. With abundant opportunity for adaptive and nonadaptive evolution of the *H. naledi* lineage over this time, it would be very hard to distinguish a long branch connecting it to *Au. sediba* or *H. habilis* from a somewhat longer branch connecting it to the very base of *Homo*.

## Conclusions

The excavation of the Lesedi Chamber has added to our knowledge of the biology of *H. naledi* and has confirmed the presence of this species in a second depositional context. The skeletal material described here derives from a very small and limited excavation, and the total sediment volume of the chamber has not yet been sampled sufficiently to estimate the abundance of hominin-bearing deposits or the relationship of faunal and hominin species. Further resolution of how the material was originally deposited must await more detailed sedimentological analysis and more excavation work.

The relative completeness of the morphological evidence from *H. naledi* has not resolved its phylogenetic placement within the genus *Homo* (**Dembo et al., 2016**). The morphological evidence from the Lesedi Chamber hominin material reinforces the observations made on the basis of the Dinaledi hominin sample. In particular, the more complete LES1 cranium and the more fragmented cranial remains from multiple individuals already known for *H. naledi* share an almost identical pattern of derived features with other hominin species (**Laird et al., 2017**). The discovery of *H. naledi* within the Dinaledi Chamber documented a pattern of anatomy that had not been anticipated by anthropologists on the basis of earlier fossil discoveries. Learning how *H. naledi* connects to other fossil evidence of human origins may require more unexpected discoveries.

## Materials and methods

### Excavation

Formal excavations began in the Lesedi Chamber in May 2014, after initial surface and ex situ material had been recovered. Like the Dinaledi Chamber, the access route to the Lesedi Chamber is very restricted. Additionally, the excavation area available in the Lesedi Chamber is considerably more confined than that in the Dinaledi Chamber. As a result, protocols followed those used in the Dinaledi Chamber (**Dirks et al., 2015**), with a few exceptions. Specifically, attempts to use some of the

surface-scanning methods applied in the Dinaledi Chamber (*Kruger et al., 2016*) failed due to the limited working distances of the scanner, the convoluted surfaces, and the poor contrast of the surrounding dolomite. However, high-resolution laser scan data have been acquired for the Lesedi Chamber and will form the basis of future spatial work. Consequently, documentation of the excavations and provenience of the material was conducted using traditional archaeological methods, including written descriptions, photography and drawings. Excavations were conducted in tandem with geological mapping and with sedimentologic and taphonomic analyses of the site.

For excavations in U.W. 102a, the area was divided into 20–40 cm sections that were based on the contours of the sloped surface and the width of the tunnel at that point. Sections were excavated in 2–5 cm levels. In some areas where few specimens were being recovered, levels were expanded to 10 cm.

In the U.W. 102b area, excavations began on the surface of the sediment deposit in which the first material was recovered, and proceeded in 2–5 cm levels until the chert layer on which the sediment had accumulated was reached. When this area was exhausted, excavations moved along the chert shelf, removing sediments from the surface, to the chert, in 20 cm horizontal sections. A sediment pocket above the fossil deposit (on Chert 3) was also excavated to determine whether material had trickled down from above. Preliminary excavations were also begun on the antechamber floor below the primary fossil deposit.

Owing to the confines of the dolomite recess in which the hominin material was found, U.W.102c was not divided into sections. However, it was also excavated in 2–5 cm levels, from the top of the sediment accumulation to the chert shelf at its base.

For all three areas, all sediments were collected for every section and level separately. Each bag of back dirt was labeled, removed from the cave and dry screened to recover small elements, fragments and microfaunal remains. All sediments were retained and are currently stored at the University of the Witwatersrand's Evolutionary Studies Institute.

## Comparative samples

Taxonomic identification of the Lesedi Chamber hominin material depended on a series of systematic comparisons to other hominin species. Many of these observations were carried out on the original fossil specimens by the authors; some have relied upon observations taken from research-grade casts of fossil specimens available at the University of the Witwatersrand and elsewhere, while in some cases, observations were only available from the literature. The fossil comparative samples employed in this study are essentially the same as those described in *Berger et al. (2015)*, with the addition of the Dinaledi Chamber sample of *H. naledi* (*Berger et al., 2015*; *Kivell et al., 2015*; *Marchi et al., 2017*; *Feuerriegel et al., 2017*; *Laird et al., 2017*; *Williams et al., 2017*; *Zipfel and Berger, 2009*). The gross morphological appearance and dental dimensions of the remains immediately made it clear that they were inconsistent with *Paranthropus*, and with *Australopithecus afarensis*, *Australopithecus africanus*, or *Australopithecus garhi*, and so we focused our comparisons upon *Homo* and *Australopithecus sediba*. The composition of samples of other hominin species is as follows, largely repeated from *Berger et al. (2015)*.

*Australopithecus sediba.* The partial skeletons MH1 and MH2 from Malapa, South Africa were included in this study on the basis of examination of the original specimens by the authors.

*Homo habilis.* Samples from Olduvai Gorge, East Lake Turkana, the Omo Shungura sequence, Hadar, and Sterkfontein were considered within the hypodigm of *H. habilis* for this study. Original Olduvai Gorge and East Lake Turkana fossils were examined first-hand, whereas for the Omo and Hadar materials, we relied on our original observations on casts and originals and published reports (*Tobias, 1991*; *Boaz et al., 1977*; *Kimbel et al., 1997*). As in the initial announcement of *H. naledi* (*Berger et al., 2015*), in this paper we adopt a conservative approach that follows a more conventional hypodigm, thereby encompassing a maximum amount of variation in this taxon; for a more detailed discussion of the probable hypodigms of early *Homo* species, see *de Ruiter et al., 2017*. We therefore include the following fossils in the hypodigm of *H. habilis*: A.L. 666–1, KNM-ER 1478, KNM-ER 1501, KNM-ER 1502, KNM-ER 1805, KNM-ER 1813, KNM-ER 3735, O.H. 4, O.H. 6, O.H. 7, O.H. 8, O.H. 13, O.H. 15, O.H. 16, O.H. 21, O.H. 24, O.H. 27, O.H. 31, O.H. 35, O.H. 37, O.H. 39, O.H. 42, O.H. 44, O.H. 45, O.H. 62, OMO-L894-1, and Stw 53.

*Homo rudolfensis.* Samples from Olduvai Gorge, East Lake Turkana, and Lake Malawi were considered as part of the hypodigm of *H. rudolfensis* for this study. The East Lake Turkana fossils

available prior to 2010 were examined first-hand, while for the Olduvai and Lake Malawi fossils and for KNM-ER 60000, 62000, and 62003, we relied on original observations of fossils and casts as well as published reports (*Blumenschine et al., 2003*; *Schrenk et al., 1993*; *Leakey et al., 2012*). As above, and in the initial announcement of *H. naledi* (*Berger et al., 2015*), in this paper we adopt a conservative approach that follows a more conventional hypodigm, thereby encompassing a maximum amount of variation in this taxon; for a more detailed discussion of the probable hypodigms of early *Homo* species, see *de Ruiter et al. (2017)*. We include the following fossils in the hypodigm of *H. rudolfensis*: KNM-ER 819, KNM-ER 1470, KNM-ER 1482, KNM-ER 1483, KNM-ER 1590, KNM-ER 1801, KNM-ER 1802, KNM-ER 3732, KNM-ER 3891, KNM-ER 60000, KNM-ER 62000, KNM-ER 62003, O.H. 65, and UR 501.

*Homo erectus*. Samples from Buia, Chemeron, Daka, Dmanisi, East and West Lake Turkana, Gona, Hexian, Konso, Mojokerto, Olduvai Gorge, Sangiran, Swartkrans, Trinil, and Zhoukoudian were included in the hypodigm of *H. erectus* for the purposes of this study. South African material is of special interest in this comparison because of the geographic proximity, and because of the difficulty of clearly identifying *Homo* specimens within the large fossil sample from Swartkrans. In particular, the following specimens from Swartkrans are considered to represent *H. erectus*: SK 15, SK 18a, SK 27, SK 43, SK 45, SK 68, SK 847, SK 878, SK 2635, SKW 3114, SKX 257/258, SKX 267/2671, SKX 268, SKX 269, SKX 334, SKX 339, SKX 610, SKX 1756, SKX 2354, SKX 2355, SKX 2356, and SKX 21204. We considered 'Homo ergaster' (and also 'Homo aff. erectus' from *Wood, 1991*) to be synonyms of *Homo erectus* for this study; Turkana Basin specimens that are attributed to *H. erectus* thus include KNM-ER 730, KNM-ER 820, KNM-ER 992, KNM-ER 1808, KNM-ER 3733, KNM-ER 3883, KNM-ER 42700, KNM-WT 15000. Olduvai specimens include O.H. 9, O.H. 12 and O.H. 28. Original fossil materials from Chemeron, Lake Turkana, Swartkrans, Trinil, and Dmanisi were examined first-hand by the authors, whereas the remainder were based on casts and published reports (*Abbate et al., 1998*; *Gilbert and Asfaw, 2008*; *Wood, 1991*; *Weidenreich, 1943*; *Suwa et al., 2007*; *Antón, 2003*; *Rightmire et al., 2006*, *2017*; *Martinón-Torres et al., 2008* ; *Spoor et al., 2007*).

A large number of postcranial specimens have been collected from the Turkana Basin and appear consistent with the anatomical range otherwise found in *Homo*, and inconsistent with known samples of *Australopithecus* and *Paranthropus* from elsewhere. These include KNM-ER 1472, KNM-ER 1481, KNM-ER 3228, KNM-ER 737, KNM-ER 5881 (*Ward et al., 2015*), and others.

Specimens from the latest Lower Pleistocene and Middle Pleistocene of Europe and Africa that cannot be attributed to *H. erectus* were also included in the comparisons in this study. These include fossils that have been attributed to *H. heidelbergensis*, *H. rhodesiensis*, 'archaic *H. sapiens*' or 'evolved *H. erectus*' by a variety of other authors. Specimens include KNM-ES 11693, Arago 2, Arago 13, Arago 21, Atapuerca 1, Atapuerca 2, Atapuerca 4, Atapuerca 5, Atapuerca 6, Cave of Hearths, Ceprano, SAM-PQ-EH1, Kabwe, Mauer, Ndutu, Salé, Petralona, Reilingen-Schwetzingen, and Steinheim. We also included Neandertal samples from Krapina, Vindija, La Chapelle-aux-Saints, La Ferrassie, Monte Circeo, Saccopastore, and Feldhofer.

*Homo floresiensis*. The hypodigm of *H. floresiensis* used in this study includes specimens from Liang Bua, Flores, as described by *Brown et al. (2004)*, *Brown and Maeda (2009)*, *Tocheri et al. (2007)*, *Orr et al. (2013)*, *Jungers et al. (2009a*, *2009b)*, *Larson et al. (2009)*, *Morwood et al. (2005)*, *Falk et al., 2005*, *Kaifu et al. (2011)*, and *Kubo et al. (2013)*.

## Endocranial volume estimation

The LES1 ECV was estimated virtually using 3D surface scans of the reconstructed partial cranium. These methods were similar to those performed on the previously published Dinaledi *H. naledi* crania (*Berger et al., 2015*). The LES1 partial cranium was scanned using a NextEngine 3D Scanner and associated ScanStudio HD Pro software. Two 360-degree scans were collected using 16 divisions and 1000 dpi with the cranium in different orientations in order to capture the maximum ectocranial and endocranial surfaces. The two 360-degree scans were then merged and exported as a .ply file. The .ply file was opened in GeoMagic Studio, converted to a point cloud, and re-wrapped to minimize the number of edges in the model (created from the individual surface scans). The LES1 virtual model was then duplicated and mirrored, and the mirror-image (right portion) was aligned with the original model (left portion) using both the Manual and Global Registration functions. In doing this, the program uses an iterative process to find the greatest congruency between the region of overlap

in the two models, minimizing deviations. Convergence was detected after 20 iterations, with an average deviation of 0.47 mm and a standard deviation of 0.43 mm. Some deviation is expected given the natural anatomical asymmetry and refitting of the fragments. A deviation map illustrates that areas with the highest deviation were fairly localized (*Figure 37*). We inspected the mirror image in comparison to the preserved but not refitted fragments that represent portions of the right parietal and right temporal bones of LES1, finding that they are similar in size and curvature. The LES1 original images and mirror-image were then merged.

The endocranial surface was then isolated by manually selecting the surfaces and creating a new model (*Figure 38*). Fragment edges were deleted and small holes in the model were filled using the 'Fill by Curvature' function. Gaps in the model remained in the posterior parietal, occipital, and cranial base regions. In order to compute a volume, the model must be completely closed; thus, these regions were carefully filled using either the 'Fill by Curvature' or 'Flat Filling' functions depending on the regions. This procedure was carefully monitored and the approximated surfaces were modified as necessary to ensure that they were congruent with the endocranial surfaces present. The volume of the closed endocranial model was then computed in GeoMagic Studio.

Previous research suggests that using a model cranial base from another hominin taxon does not significantly alter virtual estimate results (*Berger et al., 2015*); however, given the uniformity in the cranial base reconstruction of the endocranial model presented here (e.g., the lack of consideration for sella turcica, etc.), the current estimate may be a slight overestimate. Holloway (personal communication) found that by manually constructing a cranial base, ECV estimates for the previously published DH1/DH2 and DH3/DH4 *H. naledi* composite endocasts decreased by 5 ml (0.9% and 1.1%, respectively).

## Taphonomy

The hominin assemblage from the Lesedi Chamber was analysed using the taphonomic protocols and methods applied to the U.W. 101 Dinaledi fossils, and detailed in *Dirks et al. (2015)*: 22–24 and 32–33. Specimens were viewed macroscopically, and fragments >50 mm diameter were imaged on all surfaces using a Canon 70D DSLR fitted with a Canon EF-S 60mm f2.8 macro lens and ring-flash. Criteria for scoring taphonomic observations are detailed in *Supplementary file 5*. Additional taphonomic analyses are underway and will be described in future publications.

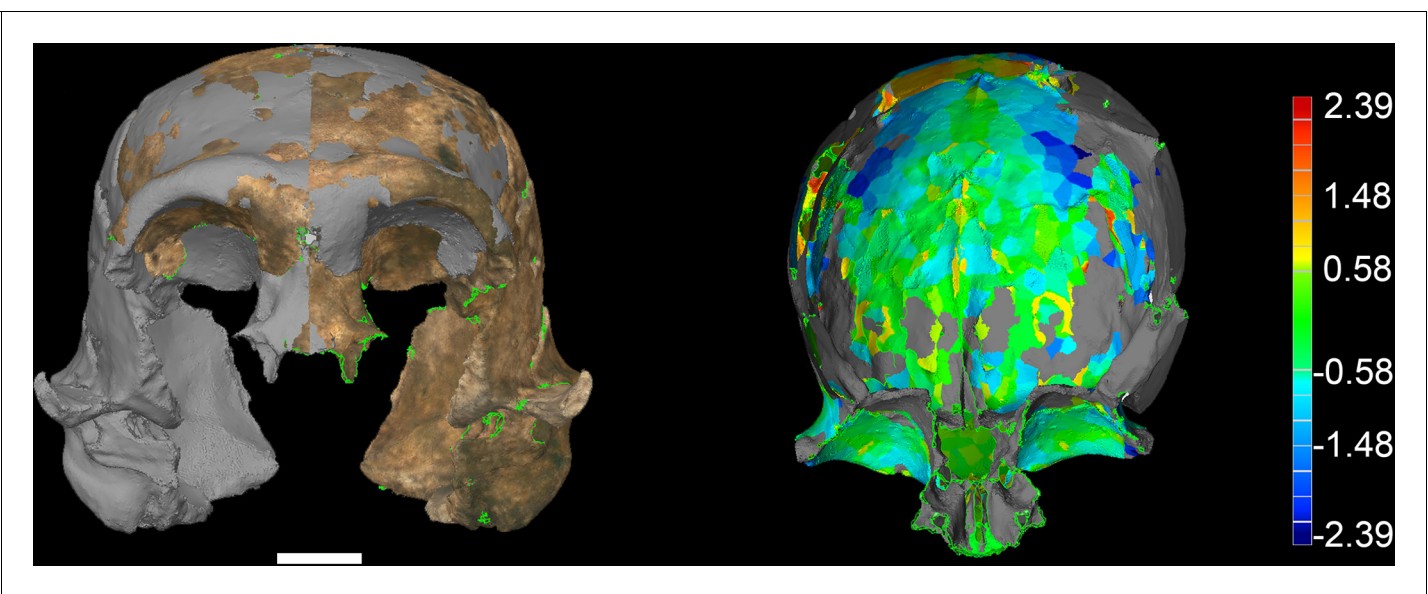

**Figure 37.** LES1 model congruency. Congruency between the original LES1 3D scan and an aligned mirror-image. Scale bar = 2 cm. (**A**) Frontal view of a 3D scan of the original LES1 specimen (brown) aligned with the mirror-image (grey), illustrating congruency between overlapping regions. (**B**) Deviation map of the internal view of the frontal region. Deviation scale bar in mm.

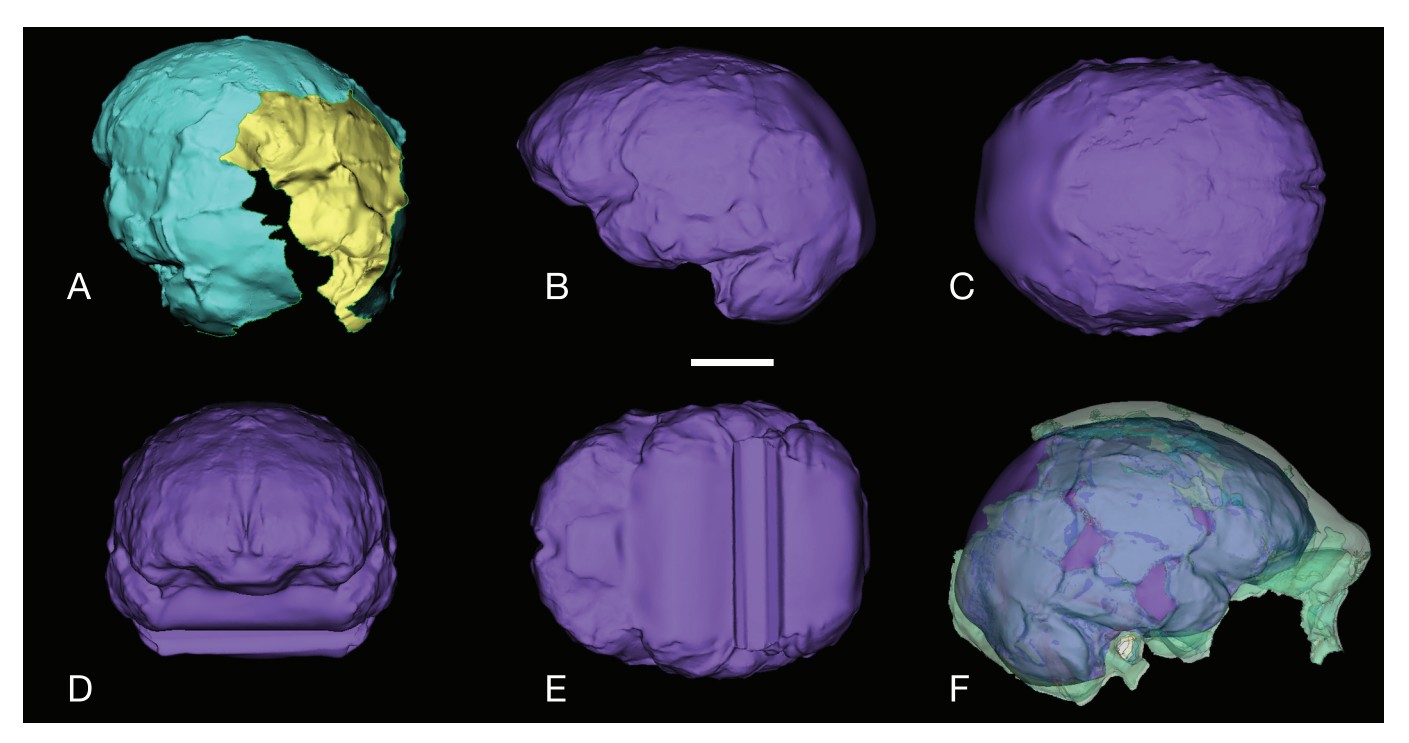

**Figure 38.** LES1 endocast reconstruction. Virtual reconstruction of LES1 endocast for endocranial volume estimation. Scale bar = 3 cm. (**A**) Oblique posterior view illustrating missing portions in the endocranial reconstruction that were filled to close the model for volume estimation. (**B-E**) Left lateral (**B**), superior (**C**), anterior (**D**), and inferior (**E**) views of the completed endocranial model. (**F**) Right lateral view of the completed endocranial model within the surrounding cranium.

### Faunal identifications

The taxonomic classification of the micromammal remains follows that of *Wilson and Reeder, 2005*, using published descriptions and images (*Avery, 2007*; *Coetzee, 1972*; *Davis, 1965*; *De Graaff, 1981*; *Meester and Setzer, 1971*; *Reppening, 1967*; *Skinner and Chimimba, 2005*), as well as comparisons with a photographic database of southern African rodents from various museum collections. As genus is the lowest taxonomic level at which most material can be identified accurately (*Reed, 2005*, *(2007)*; *Reed and Geraads, 2012*), specimens were identified to this level or higher. Macrovertebrate remains were also identified using published reference guides (*Walker, 1985*) and visual comparisons with mammalian collections at the University of the Witwatersrand. Wherever possible, specimens were attributed to species. In cases where this was not possible, the next highest taxonomic level was used. Identification of the non-mammalian remains was limited to the level of class or order, pending more detailed analyses.

### Access to material

All fossil material from the Lesedi Chamber is available for study by researchers upon application to the Evolutionary Studies Institute at the University of the Witwatersrand where the material is curated. Three-dimensional surface renderings and other digital data are available from the Morpho-Source digital repository (http://morphosource.org).

### Acknowledgements

The authors would like to thank the many funding agencies that supported various aspects of this work. In particular, the authors would like to thank the National Geographic Society, the South African National Research Foundation and the Gauteng Provincial Government for particularly

significant funding of the discovery, recovery and analysis of this material. Other funding agencies include and the Palaeontological Scientific Trust, the Texas A&M College of Liberal Arts Seed Grant Program, the Lyda Hill Foundation, the Wisconsin Alumni Research Foundation, the Vilas Trust, and the Fulbright Scholar Program. Further support was provided by ARC (DP140104282: PHGMD, ER, HHW). We wish to thank the Jacobs Family, and later the Lee R. Berger Foundation for Exploration, for access to the site, and the South African Heritage Resource Agency and Cradle of Humankind UNESCO World Heritage Site Management Authority for issuing the various permits required for this work, including the excavation permit (PermitID: 952). We would also like to thank the University of the Witwatersrand and the Evolutionary Studies Institute, as well as the South African National Centre of Excellence in PalaeoSciences, for curating the material and for hosting the authors while they were studying the material.

## Additional information

### Funding

| Funder | Grant reference number | Author |
|---|---|---|
| Wisconsin Alumni Research Foundation | | John Hawks |
| Vilas Trust | | John Hawks |
| Texas A and M College of Liberal Arts | Seed Grant Program | Darryl J de Ruiter |
| National Geographic Society | | Lee R Berger |
| National Research Foundation | | Lee R Berger |
| Gauteng Provincial Government | | Lee R Berger |
| Palaeontological Scientific Trust | | Lee R Berger |
| Lyda Hill Foundation | | Lee R Berger |
| Fulbright Scholar Program | | John Hawks |

The funders had no role in study design, data collection and interpretation, or the decision to submit the work for publication.

### Author contributions

JH, assisted in project conception and administration, developed methodology, supervised analyses, investigated morphological and contextual information, carried out statistical analyses, curated data, created visualizations, wrote the original draft and edited the paper; ME, supervised fieldwork in the Lesedi Chamber, assisted in project administration, recovered and conserved hominin fossil remains, carried out analysis of contextual and morphological information, and curated data; PS, led the conservation and reconstruction of hominin fossil remains, investigated morphological data, and created visualizations; SEC, DJdR, investigated morphological information, assisted in project conception, supervised analyses, carried out statistical analyses, and wrote parts of the manuscript; EMR, carried out fieldwork, investigated the stratigraphic and geological context, created visualizations, and wrote parts of the manuscript; HH-W, carried out fieldwork, investigated the stratigraphy and geological context, assisted in recovery of hominin material, and wrote parts of the manuscript; HMG, investigated the endocast morphology, carried out virtual reconstruction, created visualizations, and wrote parts of the manuscript; SAW, investigated vertebral and costal material, carried out statistical analyses, and wrote part of the manuscript; LKD, SEB, JKB, MMS, investigated dental material, carried out formal analyses, and wrote part of the manuscript; EMF, investigated upper limb material, carried out formal analyses, and wrote part of the manuscript; PR-Q, investigated taphonomic aspects of the sample, carried out formal analyses, created visualizations, and wrote part of the manuscript; TLK, MWT, investigated hand and wrist material, carried out formal analyses, and wrote part of the manuscript; MFL, investigated cranial material, carried out formal analyses, and wrote part of the manuscript; GT, investigated costal material, carried out formal analyses, and wrote part of the

manuscript; JMD, CSW, investigated lower limb material, carried out formal analyses, and wrote part of the manuscript; MRM, investigated vertebral material, carried out formal analyses, and wrote part of the manuscript; CV, investigated pelvic material, carried out formal analyses, and wrote part of the manuscript; TLC, BK, investigated faunal material, carried out formal analyses, and wrote part of the manuscript; AK, analyzed spatial data and support for fieldwork; ST, NH, RH, HM, BP, MR, DvR, MT, carried out fieldwork in the Lesedi Chamber, investigated contextual data and assisted in design of excavation methodology; AG, carried out fieldwork in the Lesedi Chamber, recovered and recorded hominin fossil material; PB, carried out fieldwork in the Lesedi Chamber, investigated contextual data and assisted in design of excavation methodology; PHGMD, assisted with the conception of the project, developed methodology, investigated stratigraphic and contextual information, supervised contextual analyses, assisted in creating visualizations and wrote part of the manuscript; LRB, led the conception and administration of the project, developed field and laboratory methodology, carried out formal analyses, investigated morphological and contextual information, wrote parts of the manuscript

**Author ORCIDs**
John Hawks, http://orcid.org/0000-0003-3187-3755
Paul HGM Dirks, http://orcid.org/0000-0002-1582-1405
Lee R Berger, http://orcid.org/0000-0002-0367-7629

## Additional files

**Supplementary files**
• Supplementary file 1. Traits of the LES1 cranium in comparison to *H. naledi* and other hominin species.

• Supplementary file 2. Cranial and mandibular measurements.

• Supplementary file 3. Postcranial measurements.

• Supplementary file 4. Canonical variates analysis of carpal morphology.

• Supplementary file 5. Taphonomic observations by specimen from the Lesedi Chamber.

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
