## [Decision Letter]

Thank you for submitting your article "New fossil remains of *Homo naledi* from the Lesedi Chamber, South Africa" for consideration by eLife. Your article has been reviewed by three peer reviewers, and the evaluation has been overseen by a Reviewing Editor and Ian Baldwin as the Senior Editor. One of the three reviewers, Chris Stringer, has agreed to share his name. The other reviewers remain anonymous.

The reviewers have discussed the reviews with one another and the Reviewing Editor has drafted this decision to help you prepare a revised submission.

Following the initial report of the new hominin taxon *Homo naledi* from the Dinaledi Chamber of the Rising Star cave system in South Africa, published in *eLife* in 2015, this paper describes and details a second set of *Homo naledi* fossils from an additional chamber of the Rising Star system, Lesedi. The discovery is extraordinary. While the reviews of the submitted manuscript are largely complimentary, there are two substantial issues that need to be addressed prior to publication (see the reviews for specific details):

1) The overall view from the reviewers is that well-designed comparative figure(s), illustrating how *Homo naledi* morphology differs from that of other hominins, would greatly aid readers as they examined the new data and results as they consider the descriptions and conclusions. This analysis should specifically include smaller-bodied hominins from Africa, as well as *Homo floresiensis*. To some extent, addressing this issue may be aided by clear citation to previous morphological descriptions and comparisons from your group's recent publications; however, the present manuscript should be as interpretable and valuable on its own as possible. That is, since new material is being presented and discussed in the larger hominin fossil context, it would be useful and is important to have a richer visual comparative guide as part of this manuscript.

2) Taphonomic observations of the individual hominin specimens should be provided as part of this paper on a bone-by-bone basis, even if there is some overlap with the other paper under consideration.

In addition to the two largest issues mentioned above, I am choosing to take the unusual step (for *eLife*) of providing the full reviews as part of the decision letters for each of the co-submitted papers, for two reasons. First is the interconnectedness among the multiple papers submitted, which is reflected in the reviews (as some reviewers participated in the review of multiple papers). Seeing the more direct reader reactions to each paper in the context of others should be helpful (and should be taken into account) as you consider the larger revision process for the entire group of papers. The comments provided by the reviewers identify issues additional to the two listed above that must be addressed. Note that one reviewer also provided specific comments (using tracked changes) in the attached document.

We will likely ask reviewers to evaluate the revised manuscript before considering it for acceptance.

*Reviewer #1:*

This is a fairly straightforward paper describing the new *H. naledi* remains from another chamber in the Rising Star cave system. I do think that it would be helpful if they included taphonomic observations of the individual hominin specimens. I know that the overall taphonomy of the hominin bones are discussed in an accompanying paper by Elliott et al., but perhaps it would make sense for the authors to include it here in more detail on a bone-by-bone basis. It is hard to tell from pictures, but it does appear that some of the specimens may be weathered, for example, the mandibular fragment shown in Figure 21 and many limb elements appear to have damaged or missing epiphyses. There hasn't been much done on weathering patterns in caves, but generally speaking, weathering is associated with exposure. In a protected environment like a cave chamber, I would not expect much or any weathering at all. The authors attribute the breakages observed to sediment loading (was the cave filled with sediments at one point?), inadvertent human activity, and matrix effects (from the accompanying paper by Elliott et al.). But I am not sure that the damage seen on the epiphyses (Figure 15–Figure 18) is consistent with those factors. With weathering on the bone, it would suggest some exposure to sun, wind, and other abrasion agents. I think that taphonomy of the hominin specimens should be something that is addressed in some detail as it impacts interpretations of how the bones got into the cave system in the first place.

*Reviewer #2:*

Overall: This is an interesting paper providing straightforward descriptions of the new hominin fossils from the Lesedi Chamber from the Rising Star system that expand the hypodigm of the interesting *Homo naledi* assemblage. The new cranium (along with the partial skeleton, hopefully associated) represents an extraordinary finding. I look forward to see this paper published to see the reaction of the scientific community, as well as the criticisms from other colleagues.

The main problem with this paper, however, is that in its current form there is not a single comparative figure where one can actually see how *H. naledi* differs from other fossils. Hawks et al., are forcing the readers to believe all the statements about how different this taxon is from previously known hominins. Well-designed comparative figures pointing out the distinctive morphologies of *H. naledi* would strengthen the arguments of this study and make it a great addition to the literature.

In relation to the above, why there are not qualitatively and quantitatively comparisons with other small *Homo* from Africa (KNM-ER 42700, KNM-ER 5881) and especially *H. floresiensis*? Aside from the wrist, it is like the hobbit does not exist at all to these authors.

Endocranial size:

There are several individuals from Dmanisi, and KNM-ER 42700, which show that some *Homo erectus* exhibit an almost equally small endocranial volume as *Homo naledi*. Although it is true that none is smaller than 460 ml (DH3), this needs to be discussed more. Again, and importantly: LB1 exhibits an endocranial volume of 417 cm^3^ and there is no mention to it, nor the implications of it. On the other hand, the wrist bones of *H. floresiensis* are compared to those of *H. naledi*.

Figures:

Could the authors remove all of the extra black background and focus on the fossils? It would be a much better use of the figures if we could actually see the fossil morphologies discussed in the papers. Also, please provide figures of all the described material (e.g., ulna, hand bones).

Other minor comments to consider:

- Regarding the pillars on the femoral neck 'This configuration is not seen in other hominin species.' The authors do not show a clear figure of this (not in the past papers, either). It seems like this must be figured here. I would actually say that, to make a compelling case, they need to show this morphology in *H. naledi* alongside a large sample of other hominins, particularly Pleistocene *Homo* femora.

- The discussion of number of individuals confusing. Lines 510-512 propose a single adult present, but then lines 651-653 say two adults.

- Line 681, The authors should define specifically what is the hypodigm that they refer as to '*Homo erectus* sensu lato'

- Please include scale in Figure 1.

- Which 'dimensions' are represented in the axes of Figure 6?

- Figure 10 (top) shows some Lesedi Mc4's metrics. Since claims are made that shaft and head robusticity is distinct from modern humans, it would be good to have pictures of the actual specimens to compare. Also, please notice that for the relative midshaft breadth the Lesedi Mc4 specimens fall well within the modern human range.

Also, the formatting of the boxplot axes is odd (overlaying the title of the axes). The font size of the x-axis is too small.

- Figure 10 (bottom), could the authors provide the% of variance explain in each axis? Also, the y-axis is stretched relative to the x-axis.

*Reviewer #3:*

In my view this paper should be suitable for publication after minor editing.

The references for all four papers need work for consistency of formatting, journal abbreviations, accents etc.

"The molar size gradient in the mandible is M1 < M2 < M3, and in the maxilla is M1 < M3 < M2, both identical to that observed in the Dinaledi Chamber sample of *H. naledi*." Wasn't M1 < M2 < M3 the pattern in the Dinaledi maxillary dentitions too?

"We refer to this putative skeletal individual as "Neo ", which is the Setswana word meaning "a gift"." How does this assist the analysis? Names like this are better left for the media and popular coverage?

"The hominin material from the Lesedi Chamber remains undated as this would require physical destruction of the specimens prior to their description." Techniques using laser ablation are minimally destructive and could have been attempted by now?

*Reviewer #3 Minor Comments:*

I have added suggested edits to a text version of the paper, which I attach.

[Further changes were requested before acceptance.]

Thank you for resubmitting your work entitled "New fossil remains of *Homo naledi* from the Lesedi Chamber, South Africa" for further consideration at eLife. Your revised article has been evaluated by a Reviewing Editor and five peer reviewers representing the intersection of the two previously submitted manuscripts from which this current version was created.

The reviewers were complimentary about the progress made on the manuscript in this revised version, but they note several outstanding larger issues, as well as a relatively long list of smaller or technical items, that must be addressed.

Major Issues:

1) The manuscript provides insufficient support for the hypotheses that there has been no post-depositional dispersal and that the faunal elements entered the chamber after the hominins. Either (A) sufficient, convincing evidence needs to be provided, or (B) this section needs to be removed along with the conclusions related to deposition. There are multiple detailed reviewer comments provided below that pertain to this topic, which all together should help to illustrate the strong consensus view among reviewers on these points.

At this point, we strongly suggest that the authors take approach (B), with a simplified presentation of these data and a straightforward statement that issues regarding the timing and method of deposition and post-depositional dispersal are the subjects of ongoing research to be fully addressed later.

2) The reviewers felt that the revisions to address the previous decision letter's request for comparative figures illustrating that the morphology of *H. naledi* is distinct from previously recognized hominin taxa, were not sufficient. Since the conclusions in this manuscript are drawing support from the new Lesedi material and not simply or only referencing prior published comparisons, the overall reviewer and editorial consensus is that additional visual comparisons (i.e., that more fully compare the discussed Lesedi material to the original *H. naledi* material, and both to the morphology of other hominin taxa), are necessary.

Detailed Comments:

1) Regarding the chamber itself and the deposits within it, it is still not quite clear if the deposits that contain skeletal material represent the only deposits in this chamber that do contain such material, or if they are just the ones that were investigated (or, in fact, just the ones that were investigated that do contain such material). Part of this confusion arises from the fact that the authors never really say why they decided to pick those three particular parts of the chamber as they did, nor how much estimated deposit (hominin-bearing or otherwise) might remain. I am sure this will be of great interest to most readers, as the authors seem to imply that more skeletal material from these same individuals may be recovered in the future.

2) Lines 894-897: "sub-surface deposition with limited post-depositional dispersal" May be the case for U.W. 102a, but the preservation of only small cranial fragments and partial mandible from 102b, and solely a mandibular fragment from 102c suggests there was post depositional dispersal. Unless the authors are claiming the cranial fragments were originally deposited as isolated fragments, and then not dispersed afterwards. If an entire cranium decomposed, most cranial bones should be present unless there was significant post-depositional dispersal or significant pre-depositional dispersal (which is not discussed here).

3) Lines 930-941: The authors propose the hypothesis that the faunal remains entered the chamber well after the deposition of the hominins, but do not present convincing or conclusive evidence that they are NOT contemporaneous, nor that the hominids material was not deposited in the same manner as the other faunal remains (contemporaneous or not). I know that this was transferred from the Lesedi geology paper, but the authors do not present "evidence for slumping and reworking of the sediments above the hominin layers" and claim there is a "lack of clear stratigraphic association", but do not present information on the stratigraphic association (or stratigraphy period) to determine if the interpretation is supported by evidence. In short, what precludes the hominid material from being deposited by the same method (and therefore associated) as the other faunal remains? Similarly, if the immature hominin remains in 102b could be the result of slumping from around 102a (in Discussion), why couldn't the same situation be ascribed to the other fauna, arguing for the exact same pattern of deposition (and possible association). I would suggest that the entire paragraph be removed except for the first sentence, which could be added to the first paragraph of the "Faunal material" and include these issues of association with the "ongoing" research to be addressed later. Otherwise, as is, the authors are presenting interpretations without presenting the data/information behind said interpretations.

4) The authors still do not provide evidence for their hypotheses that there has been no post-depositional dispersal and that the faunal elements entered after the hominins. If the taphonomic factors are the same for all 3 chambers, which is what they suggest, then one would expect that hominin element representation would be fairly similar, but instead, there are articulated and associated specimens in 102a, but only isolated cranial fragments from 102b and c. The fact that there aren't more elements must suggest that there was some post-depositional disturbance. Otherwise, where are they? The authors suggest that the hominin deposit was much more extensive at one point (960-961), the natural question to that is, what happened? What post-depositional process occurred to substantially decrease the hominin material? I won't go into detail regarding my thoughts on the issue of contemporaneity of the hominin and faunal specimens, but I think that the authors need to provide the evidence or remove the section.

5) Line 935-941: These are all valuable observations, but they lead to an impression that the authors are pushing a depositional scenario that removes the timing of hominin deposition from the faunal remains without sufficient evidence to do so. I recommend either softening these assertions or giving additional weight to other possibilities. For example, they mention that there are "differences in preservation" of faunal and hominin remains but do not specify what those are. The fact that all faunal remains are all carnivores, indeed, suggests that perhaps small carnivores entered the cave system, became disoriented, and ended up in the chamber because they were following the scent of decomposing tissue. Without comparisons via direct radiometric or relative ages/geochemical signatures on the faunal and hominin remains, this section must remain more speculative than is made out to be. There are some very simple (albeit minimally destructive) tests that could be carried out that would definitively resolve this.

6) My main concern in the previous version of this work was the lack of comparisons with other hominins (both analytical and graphical). Besides taphonomic details, in the revised version of their manuscript, Hawks et al provide extensive tables comparing cranial, mandibular and dental characters of the Lesedi fossils with those previously attributed to *H. naledi* as well as other hominin species. The authors now also incorporated a comparative figure (their Figure 10) the LES1 skull with selected hominins in lateral view. Although I applaud these incorporations, I still miss comparisons of the rest of the fossils. Claims are made that *H. naledi* (including the Lesedi material) is clearly distinct from previously recognized hominin taxa, and that these distinctive morphologies are spread throughout the body. However, there are not comparative figures showing the distinct morphology alongside other hominins. To be fair, the existing postcranial material of early *Homo* is very scant (and some not attributed to any specific taxon), and it would be very difficult to make the case that *H. naledi* is really distinct based on its postcrania. Thus, I would appreciate it if the authors made the effort to recognize this limitation (at the moment there is no way to really tell whether or not *H. naledi* is just a late *H. habilis*). Based on the metrics reported by the authors, only two dimensions of the maxillary M1 seems distinct from other early *Homo*, and this is based on small samples of the latter.

7) Only the combination of 18 variables of two wrist bones together in a single multi-group discriminant analysis shows that Lesedi and Dinaledi are distinct (Figure 16). However, I wonder if these fossils were left ungrouped in this analysis or whether were they included as their own OTU. If it is the latter case, the null hypothesis is that this analysis will show how distinct these specimens are, rather than to which other taxa they resemble. More importantly, even for the case of the wrist bones (supposedly very distinct), they are not morphological comparisons with other fossils (the same applies for all the other postcranial and dental material). Why are the authors hesitant to show morphological comparisons with other hominin materials? Again, the readers should be able to evaluate the distinctiveness of the new fossils by themselves. I understand that full morphometric comparisons can follow in future studies, but I believe this paper is not complete without more graphical comparisons.

8) I think that the authors have sufficiently addressed my comment regarding the appearance of weathering on the bones. I am very glad that they included the detailed table on the taphonomy of the hominids; however, as is, it does raise questions about how they scored some of the taphonomic variables. For example, how did they score heavy vs. patchy staining, what is excellent, good, fair, and poor preservation? Also, the table needs to be reformatted. It is very difficult to read it in its current format.

9) Line 170: Use of "in situ" implies that there is good evidence the skeletal material has not been moved substantially from its place of deposition. It seems like reworking and spreading out of material is fairly common inside this chamber (for example, reference to the "apron" of material), so I am unsure "in situ" is the appropriate term without some qualification. My preference would be to describe the assemblage as minimally reworked, as evidenced by some articulated elements and refitting fragments/individuals, but to avoid use of "in situ" unless it is very clearly the original place of deposition.

10) Location of chamber: All the information about the various parts of the cave system (knee-breaker, pinch and punch, etc.) should be either left out or included on the Figure 2 diagram.

11) Based on various reviewer comments, a small clarification is needed for consistency between the abstract and later in the text, on the minimum number of individuals present. When hypothesizing that there may be at least four individuals, first restate that the confirmed minimum is three.

---

## [Author Response]

Editorial comments:

*Comparative analyses and figures.*

We have reorganized the descriptions to emphasize the comparative anatomy as a separate section where appropriate with each anatomical element. The key similarities and differences between *H. naledi* and other species are now highlighted in each of these sections, with specific reference to fossil specimens in other species. For the LES1 cranium, this section is newly added and summarizes the key comparisons with other species with reference to the more descriptive and detailed analyses of the Dinaledi hominin sample published by Laird et al. (2017) and Schroeder et al. (2017). We have added one large comparative figure that presents visual comparisons between LES1 and key specimens of other species of *Homo*. We have also included many references to the descriptive papers that provide more detail and comparative figures including Dinaledi fossil remains of *H. naledi*.

*Taphonomy.*

We have ensured that each anatomical description includes a brief statement about conditions of its preservation, surface appearance, and any other details that affect the interpretation of the anatomy. In addition, we have added the new Table 6, which presents observational data pertinent to taphonomy for each bone specimen. This should conform to the instruction for a bone-by-bone treatment of taphonomic observations. The text has brought the section on general taphonomy from the manuscript by Elliott et al. This has been augmented to discuss in some detail whether the surface condition of the bones is consistent with subaerial weathering or moisture cycles within a buried environment, as requested by reviewers. We have added a new Figure 29 that shows a direct visual comparison of surface cracking on a Lesedi Chamber specimen paired with a comparable specimen from a controlled burial.

*Context.*

At the suggestion of the editors, we have taken three additional sections from the paper by Elliott et al. and added them to this paper with appropriate edits in response to reviewer comments. These include the sections describing the name of the Lesedi Chamber, the location of the chamber within the Rising Star cave system with detail about the three excavation areas, and the list of faunal species recovered in the chamber. These aspects are the essential ones for understanding the location of the hominin remains and any possible geological associations. The paper by Elliott et al. includes substantial detail concerning the sediment facies, formation of the deposits and other matters that do not directly impinge on the identification or taxonomic assessment of the hominin remains, and we leave those for a resubmission of that work to another journal. The author lists of the two papers have been merged with appropriate weight on contributions.

Reviewer comments:

Weathering: The surface condition of the bones have been described in some detail both bone-by-bone and in a general taphonomy section. The conditions under which cracking and some abrasion have occurred are compatible with a long underground burial, and surface modifications were made after deposition of manganese on the surfaces.

*H. floresiensis:* We have added text with explicit comparisons to LB1 and other *H. floresiensis* remains where appropriate throughout the paper. We have also updated the endocranial volume as reflected by the newer estimations by Kubo et al. 2013. Qualitative comparisons with other hominin species are presented in Table 2 and should now be clear in text for the postcranial skeleton.

Black backgrounds: We appreciate the reviewer’s comment concerning background color. When deciding on photo backgrounds, we have to weigh many issues including the range of end-users of these open-access photos, for whom a dark background is commonly requested for use in lectures, courses and comparisons with other fossil hominins on black backgrounds. We have worked to make each image as legible and true to the color of the specimen as possible. We are also providing freely downloadable surface models of the fossils for readers who want a better ability to inspect the orientations and outlines of specimens.

Ulna and metacarpal figures have been added.

Femoral neck: This morphology has been pictured by Marchi et al. (2017) in two figures, which were probably not yet available at the time of the review. As these have now been published, we have cited that paper here.

Number of adults: We have clarified this in text.

Hand and wrist figures: We have changed these plots in several ways to enhance their readability. The text now reports more detail about the canonical variates analysis, and Table 5 now presents all correlations between variables and CV1 and CV2.

Molar size gradient: We have expanded this paragraph to clarify that the maxillary molar size gradient, which was not described in earlier publications, is not the same as the mandibular molar size gradient in *H. naledi*.

Laser ablation: The accompanying paper by Dirks et al. expresses our protocols for geological sampling and destructive sampling for dating purposes. While laser ablation for U-series dating can allow a rapid assessment of whether material has substantial antiquity, the method is limited to determining a minimum age based on uranium uptake. In the case of the Dinaledi tooth samples, the direct U-Th minimum ages are much lower than the dates as estimated by ESR. Proceeding with only the laser ablation U-Th approach would have yielded very misleading results. In our case we do not pursue any destructive sampling of material without an integrated plan for combination of methods, including biomolecule sampling in addition to stable isotope and radioisotope sampling. We hope that this process maximizes information and confidence in the end results, and it is also consistent with our understanding of the heritage regulations that apply to this work.

[Further changes were requested before acceptance.]

We have addressed the two editorial directives as follows:

a. We followed the advice to simplify all aspects related to deposition and site formation to a bare minimum, so that we are presenting data with no unsupported conclusions about these matters. To be clear, we completely agree with the reviewers that post-depositional dispersal of some material has happened, and that we do not know the timing of entry of faunal material, and we did not intend to give any other impression.

b. We have included 6 new figures (7, 10, 11, 12, 28, and 36) that feature comparative photographs of Lesedi Chamber fossil material, examples from the Dinaledi Chamber, and selected fossil specimens from other sites, as appropriate. These illustrate the cranium, mandible, dentition, and femur, which join the existing comparative figures that present data on tooth size, vertebral morphology and hand and wrist morphology. We have also included 2 new figures (34 and 35) that present visual summaries of the anatomical evidence for similarity and difference with other species in a phylogenetic context for cranial and mandibular, and postcranial features, respectively. One more new figure (33) presents a visual comparison of endocranial volume data, and Figure 13 has been changed to add comparative data on maxillary and mandibular canine size in *H. naledi* and other species. We would love to add even more specimens in some of the comparative figures, but high-quality images or casts of many fossils are not available to include. We think the new figures strike an appropriate balance of illustrating well-known fossils, for which many researchers have casts in their reference or teaching collections, in combination with fossils from both the Dinaledi and Lesedi Chamber assemblages. We have prioritized including comparative images of Dmanisi specimens as these have previously been compared to the cranial and mandibular anatomy of *H. naledi*, and these specimens are not part of most researchers’ cast collections. In addition to these new figures, we have added 4 paragraphs to the conclusion discussing how combined morphological and temporal evidence address the relationship of *H. naledi* to (1) archaic and modern humans, (2) *H. floresiensis*, (3) *H. erectus*, and (4) *Au. sediba, H. habilis,* and *H. rudolfensis*.

We have also taken the opportunity to move large tables into Supplementary Files, which we think will make them easier to read. Each of these has been edited and formatted for readability.

Our changes responding to each of the reviewer comments are numbered here as in the decision letter (rather than re-iterating the full comment).

1. Why did we dig where we did? We appreciate this comment because many of the other questions are easily answered when the very limited scope of the excavation is understood. We have revised our discussion on pages 5 and 6, of the location of fossils in the chamber, to make clear that the total excavation to date involves an extremely small and limited sampling of the chamber’s contents, and our investigation of the three areas followed discovery of fossils on the surface, not any systematic sampling. A clear description of the very small excavation volumes (< 200 L in total, only <20 L for 102b and 2 L for 102c) probably answer other reviewer questions or place them better into context.

2. The question about postdepositional dispersal of material is well taken. We have addressed this in the text by (1) clearly stating our working hypothesis that extensive sediment removal has happened during the history of the chamber, as well as reinforcing that postdepositional processes may have dispersed material, (2) removing any parts that appear to assume anything about the formation or subsequent history of the deposits.

3. Faunal material: We agree with the reviewer here who notes that we do not know whether the fauna are contemporary with the hominins, and that is exactly what we hope to convey. Our impression from the reviewer comments is that including alternate hypotheses for entry may confuse matters by making it appear that we are pushing for non-association. Accordingly, we have edited the text to remove any speculative statements so that we do not seem to be pushing any hypothesis and are strictly limiting to the data.

4. Why is there a discrepancy in element representation between 102a, 102b and 102c? We think that this question is now answered by our description of the excavation areas, where we are clear that the 102b and 102c excavations each comprise only a few liters of sediment. The 102c deposit, for example, could not contain very much material.

5. This comment also concerns the timing of faunal deposition. We think that the edits to this section, which now only say that we do not know whether the faunal remains and hominins are contemporary, will address this comment.

6. Comparative figures. We have added the 8 new figures and comparative text to Discussion as noted above. Figure 35 helps to address some of the concerns with lack of postcranial evidence in other species. The figure clearly notes all instances where data are missing from other species, and delineates the Dinaledi Chamber evidence from the evidence that is present in both chambers.

7. Wrist bones discriminant function. This comment seems to have misunderstood the text. The analysis of the scaphoid and capitate morphology is a canonical variates analysis (CVA) and discriminant function analysis is one special case of canonical correlation, we have not carried out a discriminant function analysis here. It is not our aim to assign the fossil specimens to groups based on their anatomy, but to graphically show how the fossil specimens compare to the extant taxa and each other, specifically in terms of their scaphoid and capitate anatomy. We have examined the results of discriminant function analysis and they are fundamentally identical; that is, whether *Homo naledi* (or the other fossil specimens) is included as its own *a priori* group or as *a posteriori* unknown specimens, the CVA results are essentially identical. Just to avoid any confusion, we have added a title to that CVA plot in addition to the description of the CVA analysis in text and the figure caption. We have moved the details of the analysis from a table into [Supplementary-material SD4-data].

8. Taphonomy table. We have revised this table so that it is more legible on the page. We have moved this to [Supplementary-material SD5-data]. This supplementary file also now includes a new table with scoring criteria for each observation and detailed references for all observations made on the preservation of the bone material.

9. *In situ*. This term causes a lot of confusion as it is used differently by archaeologists, taphonomists, and geologists. We have removed it except where it refers to the instance of articulated remains.

10. Description of chamber: We have omitted all nonessential details concerning the cave system and names of locations other than the Lesedi and Dinaledi Chambers.

11. MNI: We have edited the text in accord with the suggestion here to reflect both the “confirmed minimum” and what we view as the more likely alternative in light of the spatial context.